# Ocean carbon sink assessment via temperature and salinity data assimilation into a global ocean biogeochemistry model

Frauke Bunsen[1], Judith Hauck[1], Sinhué Torres-Valdés[1], and Lars Nerger[1]

[1]Alfred-Wegener-Institut, Helmholtz Zentrum für Polar- und Meeresforschung, Bremerhaven, Germany

**Correspondence:** Frauke Bunsen (frauke.bunsen@awi.de)

**Abstract.** Global ocean biogeochemistry models are frequently used to derive a comprehensive estimate of the global ocean carbon uptake. These models are designed to represent the most important processes of the ocean carbon cycle, but the idealized process representation and uncertainties in the initialization of model variables lead to errors in their predictions. Here, observations of ocean physics (temperature and salinity) are assimilated into the ocean biogeochemistry model FESOM2.1-REcoM3 over the period 2010-2020 to study the effect on the air-sea $CO_2$ flux and other biogeochemical variables. The assimilation nearly halves the model-observation differences in sea surface temperature and salinity, with modest effects on the modeled ecosystem and $CO_2$ fluxes. The main effects of the assimilation on the air-sea $CO_2$ flux occur on small scales in highly dynamic regions, which pose challenges to ocean models. Its largest imprint is in the Southern Ocean during winter. South of $50\,°$S, winter $CO_2$ outgassing is reduced and thus the regional $CO_2$ uptake increases by $0.18\,\mathrm{Pg\,C\,yr^{-1}}$ through the assimilation. Other particularly strong regional effects on the air-sea $CO_2$ flux are located in the area of the North Atlantic Current. Yet, the effect on the global ocean carbon uptake is a comparatively small increase by $0.05\,\mathrm{Pg\,C\,yr^{-1}}$ induced by the assimilation, yielding a global mean uptake of $2.78\,\mathrm{Pg\,C\,yr^{-1}}$ for the period 2010-2020.

## 1 Introduction

The ocean plays a pivotal role in regulating the global carbon budget and thereby mitigating the impacts of anthropogenic carbon dioxide ($CO_2$) emissions on the Earth's climate. Since the 1960s, the ocean has absorbed consistently around 25% of anthropogenic $CO_2$ emissions annually (Friedlingstein et al., 2023) and has cumulatively taken up 26–34% of fossil and land-use change $CO_2$ emissions since the onset of the industrial revolution (Crisp et al., 2022). However, quantification of air-sea $CO_2$ flux still remains challenging. Air-sea CO2 flux is usually inferred from the gradient of partial pressure (pCO$_2$) or fugacity (fCO$_2$) of $CO_2$ across the air-sea interface (Wanninkhof, 2014). Yet, during 2010-2020, which constitutes the best-sampled decade in terms of surface ocean pCO$_2$ observations so far, observations covered merely 3% of the monthly global ocean (as calculated from the $1°$x$1°$-gridded SOCAT product; Bakker et al., 2016). While the North Atlantic and North Pacific are comparably well observed, data remain scarce in vast regions, such as the Indian Ocean, South Pacific and areas south

of 30°S during austral winter, where less than 1% of SOCAT grid cells have been sampled. Although these observations are thought to be representative of a larger area (Jones et al., 2012; Hauck et al., 2020), challenges in deriving a comprehensive global estimate of the global ocean $CO_2$ uptake arise due to substantial spatial and temporal $pCO_2$ variations and potential biases induced by the irregular sampling pattern (Denvil-Sommer et al., 2021; Gloege et al., 2021; Hauck et al., 2023b). Particularly in the Southern Ocean, the uncertainty is considerable (Gerber et al., 2009; Gloege et al., 2021), where estimates of the mean flux range from -0.37 to $-1.25\,\mathrm{Pg\,C\,yr}^{-1}$ for the period 2010-2018 (data provided by Hauck et al., 2023b).

In the Global Carbon Budget, estimates of the ocean carbon sink were initially derived from hindcast simulations of global ocean biogeochemistry models (GOBMs) (Le Quéré et al., 2009; Wanninkhof et al., 2013; Hauck et al., 2020). More recently, air-sea $CO_2$ flux estimates were added based on regression and machine learning techniques, interpolating $pCO_2$ observations to achieve global coverage through advanced statistical methods (referred to as $pCO_2$ products; Rödenbeck et al., 2015). Furthermore, atmospheric transport models that ingest atmospheric $CO_2$ measurements were employed to estimate the ocean carbon uptake (referred to as atmospheric inversions; Peylin et al., 2013). Although the different estimation methods have provided valuable and robust insights into large-scale patterns of oceanic carbon uptake (Gruber et al., 2009), discrepancies have emerged. Assessments based on $pCO_2$-products tend to yield larger estimates of the ocean carbon sink, with stronger trends towards more uptake, compared to estimates based on models (Friedlingstein et al., 2023; Terhaar et al., 2022). The larger estimates are supported by ocean interior observations (Müller et al., 2023), atmospheric oxygen data and atmospheric inversions (Friedlingstein et al., 2023). For the years 2010-2020, $pCO_2$ products included in the Global Carbon Project suggest a mean oceanic sink of $3.0 \pm 0.4\,\mathrm{Pg\,C\,yr}^{-1}$, while the mean of Global Carbon Project GOBMs is $2.5 \pm 0.4\,\mathrm{Pg\,C\,yr}^{-1}$ (data provided by Friedlingstein et al., 2023). Trends over the same time period are $0.7\,\mathrm{Pg\,C\,yr}^{-1}\,\mathrm{dec}^{-1}$ and $0.3\,\mathrm{Pg\,C\,yr}^{-1}\,\mathrm{dec}^{-1}$, respectively.

Machine learning estimates perform well when trained with sufficient data (Gloege et al., 2021). Their performance is less reliable in data-sparse areas. Particularly in the Southern Ocean, many $pCO_2$-products show diverging results from one another and are likely biased towards more ocean uptake (Hauck et al., 2023b). However even in parts of the North Pacific, which is undersampled in the 2010s, some $pCO_2$ products show spurious decadal trends (Mayot et al., 2024). Models provide process-driven estimates of the $CO_2$ flux across the entire global ocean, drawing from the theory of ocean dynamics, biological and chemical processes (Hauck et al., 2020; Fennel et al., 2022). Despite the growing confidence in our mechanistic understanding of the ocean carbon cycle (Crisp et al., 2022), models are also subject to uncertainty. This uncertainty stems from uncertainties in model parametrization, model spin-up and initial conditions, unresolved sub-gridscale processes and uncertainties in the atmospheric forcing (Hauck et al., 2020; Terhaar et al., 2024).

Data assimilation (DA) can be employed to address the emerging discrepancies between $pCO_2$-products and models (Carroll et al., 2020). Several studies assimilating ocean surface $pCO_2$ have focused on specific regions (e.g., a baseline state of air-sea $CO_2$ fluxes in the Southern Ocean; Verdy and Mazloff, 2017), short time periods (e.g., optimized biogeochemical initial fields for the period 2009-2011 in Brix et al., 2015) or the climatological mean state (e.g., corrections of large-scale $pCO_2$ model biases in While et al., 2012). These studies capture well the assimilated $pCO_2$ observations, while obeying physical laws and biogeochemical (BGC) equations. Data assimilation can also be used to provide a better understanding of various

components of the ocean carbon cycle, such as the transport of anthropogenic $CO_2$ in the ocean (e.g., a reconstruction of anthropogenic carbon storage since 1770 in Gerber et al., 2009), regional and interannual variability of the air-sea $CO_2$ flux (e.g., global reanalysis in Ford and Barciela, 2017; Carroll et al., 2020; Valsala and Maksyutov, 2010), the biological carbon pump (e.g., carbon export at a nutrient-rich and nutrient-poor site and estimation of BGC parameters related to air-sea $CO_2$ fluxes in Sursham, 2018; Hemmings et al., 2008, respectively) and specific ecosystems (e.g., the North West European Shelf ecosystem in Ciavatta et al., 2016, 2018). So far, however, there is no data assimilation product that provides a long-term, annually updated estimate of global ocean $CO_2$ uptake.

While previous studies indicate that the available BGC observations, when assimilated in isolation, are too sparse to constrain the modeled carbon cycle (Verdy and Mazloff, 2017; Spring et al., 2021), the assimilation of physical variables is expected to have a significant indirect effect on the modeled air-sea $CO_2$ fluxes (Bernardello et al., 2024). This is because the uptake of atmospheric $CO_2$ depends in large parts on the physical carbon transport between the surface, the mixed layer and the deep ocean in the form of dissolved inorganic carbon (DIC) through mixing, upwelling and subduction (Doney et al., 2004). According to current knowledge, ocean physics is the dominant driver of interannual variability of the global air-sea $CO_2$ flux and also responsible for stagnation and acceleration of the $CO_2$ uptake on decadal scales (Doney et al., 2009; Keppler and Landschützer, 2019; Mayot et al., 2023; Liao et al., 2020; DeVries et al., 2017). Related to the strong control that physics exert on the interannual variability of air-sea $CO_2$ fluxes, it was shown in one idealized study that assimilating ocean physics at the initial state of a model simulation has a stronger and more positive impact on the modeled carbon cycle on interannual time-scales than assimilating the BGC initial state (Fransner et al., 2020). However, the relative importance of uncertainties in physical and biogeochemical fields generally remains an open research question (e.g. Séférian et al., 2014; Li et al., 2016; Lebehot et al., 2019). Therefore, we here use ensemble-based data assimilation of physical observations into a global ocean general circulation model coupled to a biogeochemistry model aiming to improve the modeled air-sea $CO_2$ flux for the years 2010-2020. For this, we continuously assimilate temperature and salinity observations from remote-sensing at the surface and from in-situ profile measurements for eleven years and update the modeled temperature, salinity, horizontal velocities and sea surface height, using an ensemble Kalman filter variant (Nerger et al., 2012).

Several difficulties are associated with physics DA into GOBMs. A common issue is erroneous equatorial upwelling leading to unrealistically high biological productivity in the tropics (Park et al., 2018; Gasparin et al., 2021; Raghukumar et al., 2015). Furthermore, any coupled ecosystem model is adapted to its associated physical model with its strengths and weaknesses through carefully selected parameter values and a spin-up to near-equilibrium. Accordingly, the modeled carbon cycle may be sensitive to deviations from the physical state that is typical for this model (Kriest et al., 2020; Spring et al., 2021). Potentially, this leads to biases in the carbon cycle through physics DA. Such effects highlight where physical model errors are compensated for by BGC parameters, and thereby DA may reveal critical areas for potentially unrealistic BGC model behavior in projections in a changing climate (Löptien and Dietze, 2019). The question therefore arises to what extent an ecosystem model coupled to a data-assimilated physical model also represents a more realistic biogeochemistry, and which mechanisms drive the response of the $CO_2$ flux in physics DA approaches. One possible driver is the physical transport of DIC and alkalinity because velocities and diffusion are changed by the DA, affecting in particular the upwelling of carbon-rich waters and subduction, which is

important to capture the ocean storage of anthropogenic carbon (Davila et al., 2022). Furthermore, physics DA may change pCO$_2$ directly through its temperature-dependence, an effect emphasized by Verdy and Mazloff (2017). Additionally, the modeled biological pump might be altered, for example through the temperature-dependency of phytoplankton growth or through effects of stratification on nutrient availability.

In this study, we describe the response of the model's air-sea CO$_2$ flux to physics DA and identify the underlying mechanistic drivers. To this end, we differentiate between the thermally, DIC- and alkalinity induced components and changes in lateral and vertical transport through mixing and advection. We focus, firstly, on the global air-sea CO$_2$ flux. Secondly, we investigate the Southern Ocean given the relevant impact of DA in Southern Ocean winter in our study. Thirdly, we present regions in the North Atlantic given observational coverage and relevant local processes there.

## 2 Methods

### 2.1 Model FESOM2.1-REcoM3

The oceanic model component, FESOM2.1, computes the advection and diffusion of passive biogeochemical tracers. The model is based on hydrostatic primitive equations under the Boussinesq approximation and utilizes a finite-volume discretization approach with surface triangles projected vertically to form prisms. Salinity (S), temperature (T), and biogeochemical tracers are located at the vertices of triangles (nodes), while the horizontal velocities are centered at the triangles (elements). The model allows for a variable mesh resolution (see Section 2.2) and incorporates parametrizations for diffusion and eddy-stirring along isoneutral surfaces, for which parametrized mixing is scaled by mesh resolution (Danilov et al., 2017). Vertical mixing is parametrized through the KPP scheme and the mixing depth is specified through a 'boundary layer' (the layer of active mixing, which may have a vertical structure because the mixing of all properties across the layer is not instantaneous, as opposed to the mixed layer which is defined by already well-mixed properties, Large et al., 1994), with an additional vertical mixing scheme used in the Southern Ocean (Monin–Obukhov parametrization, Timmermann and Beckmann, 2004). The surface salinity (SSS) is restored towards the World Ocean Atlas climatology through a fictional surface flux with $v_{\text{SSS}} = 50\,\text{m}/300\,\text{days}$ according to Equation 1 and as in Gürses et al. (2023):

$$(\text{SSS}_{\text{clim}} - \text{SSS}_{\text{model}}) * v_{\text{SSS}} * (h_{\text{surf}})^{-1} \tag{1}$$

with surface-layer thickness $h_{\text{surf}}$. A detailed description of FESOM2.1 and a model assessment are provided by Danilov et al. (2017) and Scholz et al. (2019, 2022).

The ocean biogeochemistry component, the Regulated Ecosystem Model version 3 (REcoM3), describes processes in the ocean carbon cycle and represents oceanic carbon in the form of dissolved inorganic carbon, dissolved organic carbon, plankton and detritus (Gürses et al., 2023). REcoM3 contains 28 BGC tracers listed in Appendix Table A1. There are two phytoplankton groups: diatoms and small phytoplankton with implicit representation of calcifiers; two zooplankton groups: mixed and polar macro zooplankton (Karakuş et al., 2021); and two classes of detritus. REcoM3 includes variable intracellular stoichiometry with ratios of C:N:Chl:CaCO$_3$ for the small phytoplankton and C:N:Chl:Si for diatoms, which is propagated to

zooplankton and detritus (Schartau et al., 2007; Hohn, 2008). The publicly available Routines To Model The Ocean Carbonate System (mocsy2.0, Orr and Epitalon, 2015) are used to compute $pCO_2$ and air-sea $CO_2$ flux, employing the gas-exchange parametrization of Wanninkhof (2014). Alkalinity is restored by a fictional surface flux of $10\,\mathrm{m\,yr}^{-1}$ (as in Hauck et al., 2013; Schourup-Kristensen et al., 2014; Gürses et al., 2023). The current model version FESOM2.1–REcoM3 was assessed by Gürses

et al. (2023) and previous versions were evaluated and applied in global and regional studies of the ocean carbon cycle and planktonic ecosystems (Hauck et al., 2013; Schourup-Kristensen et al., 2014; Hauck et al., 2020; Karakuş et al., 2021).

## 2.2  Simulation set-up

The model setup for both simulations closely follows Gürses et al. (2023). The mesh resolution is nominally 1 degree, ranging between 120 km and 20 km with enhanced resolution in the equatorial belt and north of $50\,^\circ$N (126858 surface nodes). It has

47 vertical layers with thickness ranging from 5 m at the surface to 250 m in the deep ocean, as described by Scholz et al. (2019, CORE mesh). The model time step is 45 minutes. For atmospheric forcing, JRA55-do v.1.5.0 is used, a reanalysis product tailored for driving ocean-sea-ice models (Tsujino et al., 2018). The atmospheric $CO_2$ mixing ratio values were taken from the Global Carbon Budget (Joos and Spahni, 2008; Ballantyne et al., 2012; Friedlingstein et al., 2023). We use model restart fields from Gürses et al. (2023) where the model was spun-up by repeating the year-1961 JRA forcing for 189 years

with preindustrial atmospheric $CO_2$ conditions, followed by a period from 1800 to 1957 with increasing atmospheric $CO_2$. Subsequently, simulations were continued with historical JRA forcing from 1958 to 2009. During the assimilation window (2010-2020), we conduct two ensemble simulations to study the impact of data assimilation (DA): one without DA (referred to as FREE) and another identical setup applying DA (referred to as ASML). For each simulation, the ensemble mean for the following variables is written as output: temperature, salinity, velocity, boundary-layer depth, surface $pCO_2$, DIC, alkalinity,

nutrients, chlorophyll, net primary production and biological export through sinking of detritus at 190 m. For the year 2020, additional output is available for individual ensemble members, mixed-layer depth, physical sources or sinks of DIC and alkalinity through horizontal and vertical advection and diffusion, and biological net sources or sinks of DIC and alkalinity through combined processes: For DIC, the net biological term is the sum of photosynthesis, respiration, remineralization of dissolved organic carbon, and formation and dissolution of calcite (Gürses et al., 2023, Equation A6). For alkalinity, the net

biological term is the sum of nitrogen assimilation and remineralization, and formation and dissolution of calcite (Gürses et al., 2023, Equation A7).

## 2.3  Data Assimilation

### 2.3.1  Assimilated observations

The assimilated observations are sea surface temperature (SST), sea surface salinity and profiles of temperature and salinity.

The assimilated SST observations are from the Operational Sea Surface Temperature and Ice Analysis (OSTIA) data set (CMEMS Marine Data Store; Good et al., 2020; Donlon et al., 2012; Stark et al., 2007). OSTIA provides daily gap-free maps of SST at a horizontal resolution of $0.05^\circ \times 0.05^\circ$, compiled from in-situ and satellite data from infrared and microwave

radiometers. The OSTIA observations were averaged to the FESOM2.1 model grid because their spatial resolution is higher than the nominal resolution of the model grid. An observation error standard deviation of $0.8\,^\circ$C is prescribed for the DA following Nerger et al. (2020). Observations are excluded in the DA process if the difference between the model and observation exceeds three times the observation error standard deviation, thus $2.4\,^\circ$C, and at grid points with sea ice in the model, as in Tang et al. (2020) and Mu et al. (2022). This exclusion keeps the model stable despite large differences between model and observations at these sites, in particular as water temperature and salinity develop differently under sea ice than under the influence of the atmosphere (Tang et al., 2020). Instead, a 'gentler' correction is made by assimilating neighboring points. After the initial phase, about 7% of SST observations are excluded because of the $2.4\,^\circ$C-threshold. Nevertheless, the data assimilation still has a strong effect in areas where these large model-observation discrepancies are typically found (North Atlantic, Japan and Southern Ocean).

The assimilated SSS data is taken from the European Space Agency (ESA) Sea Surface Salinity Climate Change Initiative (CCI) v03.21 data set (Boutin et al., 2021). ESA-CCI contains daily data at a spatial resolution of 50 km, albeit not capturing temporal variability below weekly. The ESA-CCI observations are averaged to the FESOM2.1 model grid. We prescribe a constant observation error standard deviation of 0.5 psu following Nerger et al. (2024). Like for the SST data, SSS observations are excluded at locations where sea ice is present in the model.

The assimilated temperature and salinity profiles are taken from the EN.4.2.2 data set (Good et al., 2013). The EN4 dataset contains quality-controlled profiles from various in-situ ocean profiling instruments. To assimilate the profiles, the observations are assigned to the respective model layers (depth range) in the vertical. In the horizontal, the model values are computed as the average of the grid points of the triangle enclosing the observation. The observation error standard deviation is set to $0.8\,^\circ$C for temperature and to 0.5 psu for salinity without excluding observations, as in Tang et al. (2020).

### 2.3.2 Assimilation method and implementation

For the assimilation, we use the Localized Error Subspace Transform Kalman Filter (LESTKF, Nerger et al., 2012). The LESTKF sequentially updates the model forecast, incorporating observations when and where available. The model state and error covariance are represented by an ensemble simulation. Thereby, the assimilation of temperature and salinity affects the state of the physical model in its whole, including the horizontal velocities and sea-surface height. A review of the LESTKF and other filters frequently used in geophysics can be found in Vetra-Carvalho et al. (2018). The assimilation is implemented using the Parallel Data Assimilation Framework (PDAF version 2.1), a software environment for data assimilation. PDAF is an open source project and provides fully implemented DA algorithms (Nerger et al., 2020, pdaf.awi.de). The current implementation builds on the works of Mu et al. (2022) who used DA of ocean temperature and salinity for sea-ice forecasts with FESOM2.0 coupled to an atmospheric model, and Tang et al. (2020) who studied the dynamic impact of oceanic DA into FESOM1.4 onto a coupled atmospheric component.

With localization of the LESTKF, the observation error is increased for an increasing horizontal distance between an observation and a model grid point, which weighs down the influence of a more distant observation. This avoids that the model is influenced by observations at distant locations through spurious ensemble estimated correlations. We use a localization radius

of 200 km and choose a 5th-order polynomial weighting function that mimics a Gaussian function (Gaspari and Cohn, 1999). We apply daily analysis steps at $0\,\mathrm{UTC}$ model time, assimilating all available observations for the day. The DA process only directly updates the physical model variables temperature, salinity, horizontal velocities and sea surface height. After each assimilation step, corrections are applied to the analysis state to ensure the consistency of model physics: Salinity is set to a minimum value of zero and temperature to a minimum value of $-2\,°\mathrm{C}$, if the value is otherwise below. The increment of sea surface height (SSH) is limited to two standard deviations of the ensemble. While in the simulation the correction was necessary for about 10% of SSH updates and $10^{-5}\%$ of temperature values, the correction of salinity was never required. The analysis step is followed by an ensemble forecast of 1 day.

The ensemble size is 40, a compromise to balance computational resources while ensuring a sufficiently large ensemble with enough variability even in the deep ocean. The ensemble is generated through an initial perturbation of sea surface height, horizontal and vertical velocities, temperature, salinity and sea-ice concentration based on the implementation of Tang et al. (2020). This initial ensemble perturbation is generated by second-order exact sampling (Pham, 2001) from a model trajectory of FESOM2.1. With this method, the leading Empirical Orthogonal Functions (EOFs) of a model trajectory are used to generate an ensemble perturbation that contains the leading patterns of model variability. A time-scale must be chosen for the variability that is represented by the ensemble. Here, we chose variability on a weekly time-scale (Tang et al., 2020).

To maintain ensemble spread, we apply a perturbed atmospheric forcing with an autoregressive perturbation $(\mathrm{perturb}_{\mathrm{e,n}})$ at every model time step (n) to each ensemble member (e), with:

$$\mathrm{perturb}_{\mathrm{e,n+1}} = (1 - \mathrm{arc}) * \mathrm{perturb}_{\mathrm{e,n}} + \mathrm{arc} * s * \mathrm{rand}_{\mathrm{e}} \qquad (2)$$

where $\mathrm{rand}_{\mathrm{e}}$ is a stochastic element, again generated by second-order exact sampling from a 72-days-long trajectory of atmospheric forcing fields that captures patterns of day-to-day atmospheric variability. The autoregression coefficient $(\mathrm{arc})$ can be used to tune how quickly the perturbation changes and is set to the inverse number of model steps per day. $s$ is a scaling factor for each perturbed atmospheric forcing field. For specific humidity, downwelling longwave radiation and air temperature $s = 10$ is used. The perturbation of winds is set to the smaller value $s = 2$ because the air-sea $CO_2$ flux in the model is particularly sensitive to perturbations of the wind fields. Due to the functioning of the Kalman filter (which updates the model error covariance in each analysis step to reflect the new reduced uncertainty), the ensemble spread decays at each analysis step. As the method relies on a sufficiently large ensemble spread, an inflation of the ensemble covariance is applied (Pham et al., 1998). Thereby, the ensemble covariance matrix is amplified by a factor of $1/\rho$ before entering the updating step. This so-called forgetting factor downweighs that past observations have reduced the model uncertainty (see e.g. Nerger et al., 2005). The forgetting factor is tuned to maintain model uncertainty, where $\rho = 1$ means no inflation and smaller values mean larger inflation. Here, we use a time-varying forgetting factor between $\rho = 0.95$ and $\rho = 1$. The strongest inflation ($\rho = 0.95$) is applied during the first two weeks of the DA process. This is when the DA increments are largest because the model state estimates are furthest from the observations. During the following 75 days $\rho$ is increased to 0.99. From month 17 onward, the forgetting factor is set to either 0.99 or 1.0 depending on the ensemble standard deviation of temperature.

The ensemble standard deviation of the local instantaneous air-sea $CO_2$ fluxes that results from the perturbation of physical fields is larger than that of the global $CO_2$ flux, with a mean standard deviation of $0.32\,\mathrm{mmol\,m^{-2}\,day^{-1}}$ for monthly means of local fluxes compared to a standard deviation of $0.0068\,\mathrm{mmol\,m^{-2}\,day^{-1}}$ $(0.01\,\mathrm{Pg\,C\,yr^{-1}})$ for the annual global flux in FREE in the year 2020. The largest ensemble standard deviation (Fig. A1a) is generated in the Southern Ocean, the North Atlantic and the North Pacific, which corresponds to regions of high uncertainty in existing $CO_2$ flux estimates (Pérez et al., 2024; Hauck

et al., 2023a; Mayot et al., 2024). However, the modeled standard deviation should not be understood as the true uncertainty of the model, but as a value dependent on tuning (Evensen, 2003).

## 2.4    Data analysis

We present $CO_2$ flux estimates for the period 2010-2020, that are compared to the 'Regional Carbon Cycle Assessment and Processes 2' (RECCAP2) global air-sea $CO_2$ flux estimates (DeVries et al., 2023). The RECCAP2 $pCO_2$ products account for

oceanic outgassing of river carbon into the atmosphere. To make them comparable with our estimate stemming from a model without river carbon input, we apply a river flux adjustment (Friedlingstein et al., 2023; Regnier et al., 2022) to the RECCAP2 $pCO_2$ products. Thus, we quantify the anthropogenic perturbation of the ocean carbon sink (as $S_{OCEAN}$ in the Global Carbon Budget Friedlingstein et al., 2023; Hauck et al., 2020), and not the contemporary net air-sea $CO_2$ flux with outgassing of river carbon (as in the original RECCAP2 $pCO_2$ products).

To study the effect of DA on the $CO_2$ flux, we define regions where the time-mean air-sea $CO_2$ flux difference $\mathrm{ASML-FREE}$ ($\Delta F_{CO_2}$) is pronounced based on the biome definition of Fay and McKinley (2014). Originally, these are, going polewards from the subtropics in each hemisphere, the Subtropical Seasonally Stratified Biome (STSS), the Subpolar Seasonally Stratified Biome (SPSS) and the Sea-Ice Biome (ICE). In the Southern Ocean (denoted by subscript $_{SO}$) within the $\mathrm{STSS_{SO}}$, we differentiate between the area where $\Delta F_{CO_2}$ is positive (the assimilation leads to a flux change directed out of the ocean) referred

to as region '$\mathrm{STSS_{SO}}+$' and the area where $\Delta F_{CO_2}$ is negative, referred to as region '$\mathrm{STSS_{SO}}-$'. All Southern Ocean regions are outlined in Fig. 5a. In the North Atlantic (denoted by subscript $_{NA}$), we consider four coherent regions within the $\mathrm{STSS_{NA}}$ and $\mathrm{SPSS_{NA}}$ outlined in Fig. 7a. The regions 'Central $\mathrm{STSS_{NA}}-$' and 'Western $\mathrm{STSS_{NA}}+$' are located in the North Atlantic $\mathrm{STSS_{NA}}$ biome and are defined by $\Delta F_{CO_2}$ less than $-1\,\mathrm{mmol\,C\,m^{-2}\,day^{-1}}$ and $\Delta F_{CO_2}$ greater than $1\,\mathrm{mmol\,C\,m^{-2}\,day^1}$, respectively. The regions 'Newfoundland Basin$_{NA}+$' and 'Coastal $\mathrm{SPSS_{NA}}-$' are part of the $\mathrm{SPSS_{NA}}$. The former is located

east of Newfoundland and south of Greenland, and is defined by $\Delta F_{CO_2}$ greater than $3\,\mathrm{mmol\,C\,m^{-2}\,day^{-1}}$; and the latter is located off the North American coast and defined by $\Delta F_{CO_2}$ less than $-1\,\mathrm{mmol\,C\,m^{-2}\,day^{-1}}$. The Central $\mathrm{STSS_{NA}}-$ and Western $\mathrm{STSS_{NA}}+$ lie on the warm side of the North Atlantic Current (NAC), and the Newfoundland Basin$_{NA}+$ and Coastal $\mathrm{SPSS_{NA}}-$ lie on the cold side of the NAC, which is evident from the modeled surface velocity field (Fig. A2a).

     Within these regions, we identify the time of the year when the DA affects air-sea $CO_2$ flux and calculate the difference

$\mathrm{ASML-FREE}$ for physical and biogeochemical fields. In order to assess the dynamic DA effects on surface $pCO_2$, it is useful to distinguish between different variables that constitute the change in $pCO_2$. Oceanic $pCO_2$ varies mainly with temperature, DIC and alkalinity. Thus, we decompose changes in $pCO_2$ into their contributions from changes in SST, surface DIC and

surface alkalinity (Alk). For that, we apply the following approximations of Sarmiento and Gruber (2006) and Takahashi et al. (1993):

$$\Delta\mathrm{pCO_{2,DIC}} = \frac{\mathrm{pCO_2}}{\mathrm{DIC}} \cdot \gamma_{\mathrm{DIC}} \cdot \Delta\mathrm{DIC} \tag{3}$$


$$\Delta\mathrm{pCO_{2,Alk}} = \frac{\mathrm{pCO_2}}{\mathrm{Alk}} \cdot \gamma_{\mathrm{Alk}} \cdot \Delta\mathrm{Alk} \tag{4}$$

$$\Delta\mathrm{pCO_{2,SST}} = \mathrm{pCO_2} \cdot \exp(0.0423\,{}^{\circ}\mathrm{C}^{-1} \cdot \Delta\mathrm{SST}) \tag{5}$$

Here, differences between ASML and FREE are denoted by $\Delta$; else, the average of ASML and FREE is used for the computation. The sensitivities $\gamma_{\mathrm{DIC}}$ and $\gamma_{\mathrm{Alk}}$ describe how pCO$_2$ varies with changes in one variable while keeping all other variables constant. For the sensitivities, we use an approximation derived from seawater carbonate chemistry following Sarmiento and Gruber (2006):

$$\gamma_{\mathrm{DIC}} = \frac{3 \cdot \mathrm{Alk} \cdot \mathrm{DIC} - 2 \cdot \mathrm{DIC}^2}{(2 \cdot \mathrm{DIC} - \mathrm{Alk})(\mathrm{Alk} - \mathrm{DIC})} \tag{6}$$


$$\gamma_{\mathrm{Alk}} = \frac{-\mathrm{Alk}^2}{(2 \cdot \mathrm{DIC} - \mathrm{Alk})(\mathrm{Alk} - \mathrm{DIC})} \tag{7}$$

Based on the range of valid values for $\gamma_{\mathrm{DIC}}$ and $\gamma_{\mathrm{Alk}}$ according to the explicit formulation by Egleston et al. (2010), values are excluded above 18 and below -19, respectively. This affects parts of the Southern Ocean SPSS$_{\mathrm{SO}}$ and ICE$_{\mathrm{SO}}$ biome (see white areas in Fig. 6b and c). Finally, the effect on the air-sea CO$_2$ flux relates directly to the pCO$_2$-difference at each grid
point, as detailed in Orr et al. (2017, Equations 6-15):

$$\Delta\mathrm{F}_{\mathrm{CO_2}} = \alpha \cdot k_w \cdot \Delta\mathrm{pCO_2} \tag{8}$$

where $\alpha$ is the solubility of CO$_2$ in seawater and $k_w$ is the gas-transfer velocity.

To evaluate the impact of DA on ocean physics, we compare the simulated SST and SSS to the assimilated observations (Section 2.3.1). For temperature and salinity at depth, we use the EN4-OA product (Good et al., 2013, updated to version 4.2.2).
EN4-OA is an objective analysis ingesting the assimilated EN4 profile data, interpolated to global coverage on 42 depth levels. Furthermore, we compare the sea-ice concentration with remote sensing observations from OSI-SAF 2010-2020 (EUMETSAT, 2022), the mixed-layer depth in the year 2020 with the profile-observation based climatology of de Boyer Montégut et al. (2004, updated version 2023) and the horizontal near-surface velocities 2010-2020 with the drifter-based climatology of Laurindo et al. (2017).

To evaluate the impact of the DA on biogeochemistry, we compare model outputs with observational datasets of surface $pCO_2$, DIC, alkalinity and surface chlorophyll. To evaluate surface $pCO_2$, we use observations from the Surface Ocean $CO_2$ Atlas (SOCAT Version 2023, Bakker et al., 2023, 2016), which are provided as a monthly gridded and quality-controlled compilation. To assess DIC and alkalinity, we compare the modeled surface fields to the GLODAPv2.2023 bottle data (Lauvset et al., 2024b). At depth, we compare the model output to the GLODAPv2 DIC and alkalinity climatology (Lauvset et al., 2016), which is based on observations from the period 1972-2013 and normalized to 2002. To evaluate global surface chlorophyll, we use observations from ESA-CCI, which is a multi-sensor satellite ocean-color chlorophyll-a dataset with monthly global coverage (Sathyendranath et al., 2021). In addition, for the Southern Ocean, we use the mean of three satellite products (Johnson et al., 2013) that were processed with more suitable algorithms for southern high latitudes. For each observation type (OBS), we define the improvement as:

$$\text{improvement}_{\text{OBS}} = |\text{FREE} - \text{OBS}| - |\text{ASML} - \text{OBS}| \tag{9}$$

## 3 Results

### 3.1 Effect of DA on ocean physics

Before we investigate the $CO_2$ flux, we first evaluate the effect of DA on the modeled physics. In particular, we compare the model output of both simulations with the assimilated observations to verify that the assimilation brings them into better agreement with the observations. The assimilation improves the agreement with the assimilated SST observations. On a global average, the SST in FREE is $0.14\,°C$ colder than the observations, which is the result of an extensive cold bias in the tropics and subtropics and a warm bias in the Southern Ocean south of $40\,°S$ (Fig. 1a; mean state of SST in Fig. A3a). In addition, there are regional SST differences $\text{FREE} - \text{OBS}$ in particular near strong currents and in eddy-rich regions, such as the NAC, Kuroshio, and the Southern Subtropical Front. These SST differences are estimated to lead to a solubility-driven global air-sea flux difference of $-0.06\,\text{Pg}\,\text{C}\,\text{yr}^{-1}$ (Equations 5 and 8). The assimilation increases SST in the tropics and subtropics and reduces SST south of $40\,°S$, with particularly large effects in the Southern Ocean and in the North Atlantic (difference $\text{ASML} - \text{FREE}$ in Fig. 1b). Thereby, the global mean model-observation difference is reduced from $-0.14\,°C$ to $-0.12\,°C$, and from $0.59\,°C$ to $0.32\,°C$ in absolute terms. This assimilation-induced change in SST is estimated to drive a direct solubility-driven effect on the global-air sea $CO_2$ flux of $-0.14\,\text{Pg}\,\text{C}\,\text{yr}^{-1}$ (Equations 5 and 8). Yet, this global attribution is subject to high uncertainty due to the non-linear dependency of $pCO_2$ on temperature, and because regionally large effects with opposite signs lead to uncertainty in the global mean.

The assimilation also improves the agreement with the assimilated SSS observations. Additional experiments with and without salinity restoring towards climatology show that the best agreement with the SSS-CCI observations is achieved by simultaneously using assimilation and restoring. A benefit of the additional use of restoring is the global coverage of the SSS climatology. FREE shows a global SSS bias (0.49 psu, Fig. 1d). The assimilation leads to a global surface freshening (Fig. 1e).

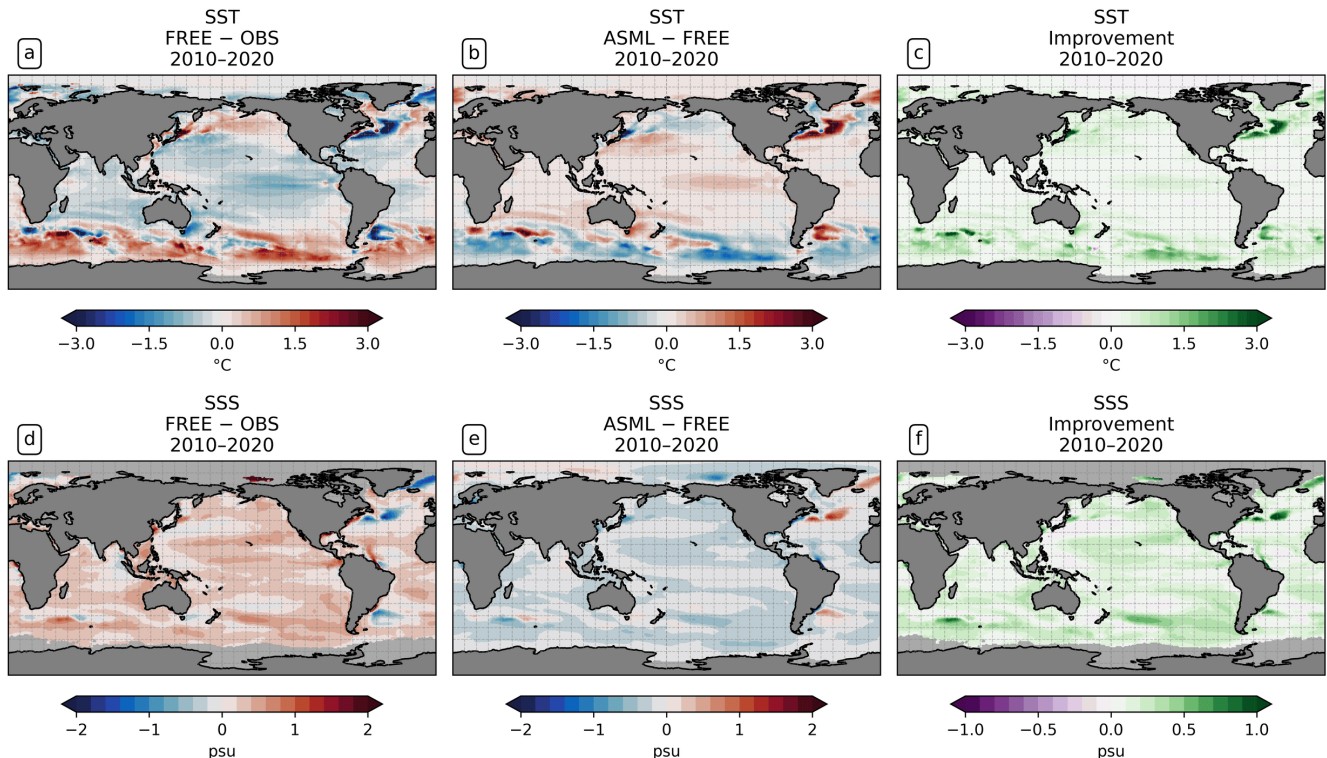

**Figure 1.** Effect of data assimilation on sea surface temperature (SST) and sea surface salinity (SSS). All panels show the mean over the period 2010-2020. (a) The model-observation difference in SST (FREE - OSTIA). (b) The difference ASML - FREE. (c) The improvement of monthly averaged model SST relative to OSTIA, where positive denotes that the assimilation brings the model closer to observations (Eq. (9)). (d - f) The same for SSS, computed with SSS from ESA-CCI.

There are only a few regions where SSS in FREE is fresher than the observations and where the DA consequently increases the salinity, as for example in parts of the North Atlantic. The assimilation improves the model-observation agreement in 91% of the observed ocean area, particularly in the North Atlantic Central $STSS_{NA}-$ and in the Southern Ocean $STSS_{SO}$ (Fig. 1f). Tests with the assimilation of temperature alone show negative side-effects of temperature assimilation on SSS in some locations (not shown). In the final set-up with combined assimilation, negative effects on SSS are found in 9% of the observed area. Globally, the mean absolute difference is reduced from 0.32 to 0.17 psu relative to the SSS observations. The direct solubility-driven effect of salinity differences on the global air-sea $CO_2$ flux is estimated to be negligible.

The assimilation leads to a better agreement with subsurface temperature and salinity data from the EN4-OA product in the upper 1000 m. In the upper 100-200 m of the ocean, the model-observation difference in temperature follows the surface signal (compare Fig. 1a and Fig. 2a), and the difference is reduced by the assimilation (Fig. 2b and c). At intermediate depth (roughly 200-500 m), a subsurface warm bias exists in FREE in the southern hemisphere at mid-latitudes (Fig. 2; mean state in Fig. A4a). This bias affects the South Pacific, South Atlantic and southern Indian Ocean (not shown). The bias might be

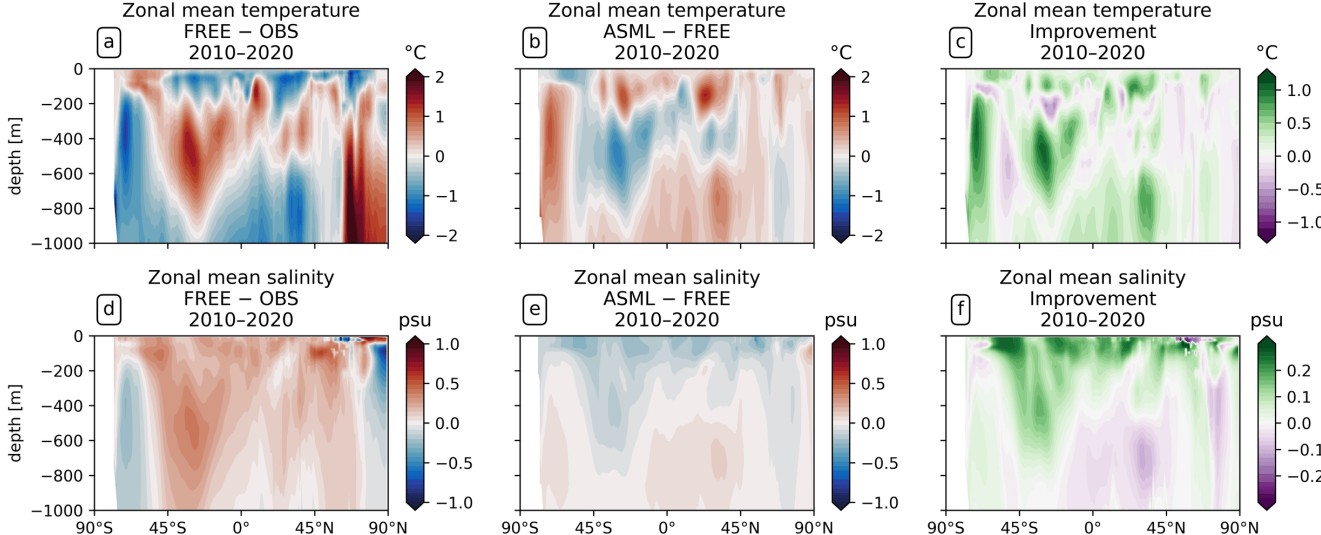

**Figure 2.** Effect of data assimilation on zonally averaged temperature and salinity in the upper 1000 m. All panels show the mean over the period 2010-2020. (a) The model-observation difference in temperature (FREE - EN4-OA). (b) The difference ASML - FREE. (c) The improvement of monthly averaged temperature relative to EN4-OA. (d - f) The same for salinity.

connected to the model's surface warm bias in the formation region of Antarctic intermediate water (Fig. 1a). Further model-observation differences exist at greater depth than 500 m, where the model's temperature is colder than the observations at almost all latitudes, but warmer than the observations north of $60\,°N$. At most latitudes and depths, the effect of the assimilation is to reduce the model observation-differences (Fig. 2c).

The model is more saline than the observations from the surface down to a depth of about 1000 m for most latitudes (Fig. 2d). This shows that the model-observation difference in this depth range follows the surface signal. The exceptions to this are at high latitudes below 200 m, where FREE is fresher than the observations. At all other latitudes, the assimilation acts towards a freshening, with the strongest effect near the surface (Fig. 2e). This improves the agreement with observations particularly near the surface (Fig. 2d). However, the improvement is smaller at depth and becomes even negative for some latitudes in greater depth. This might be due to the limited amount of assimilated in-situ salinity profiles.

The effect of the assimilation on temperature and salinity is most pronounced in the upper 1000 m and, below that, mostly decreases with depth (not shown). After the second year of assimilation, the mean absolute difference between ASML and FREE stabilizes in the range $0.35 - 0.36\,°C$ for SST and $0.20 - 0.25\,\text{psu}$ for SSS, while the effect of DA on subsurface temperature and salinity keeps increasing throughout the years 2010-2020.

Sea ice reacts dynamically to the changed ocean physical state. In the Southern Ocean, FREE is characterized by a lower sea-ice concentration compared to OSI-SAF observations. The sea-ice extent, here defined as the area where the sea-ice concentration is more than 15%, reaches a maximum in September. The maximum extent is smaller in FREE than OSI-SAF, which is demonstrated by the 15%-line surrounding that area for FREE and OSI-SAF (Fig. 3a; mean state in Fig. A5), and by the

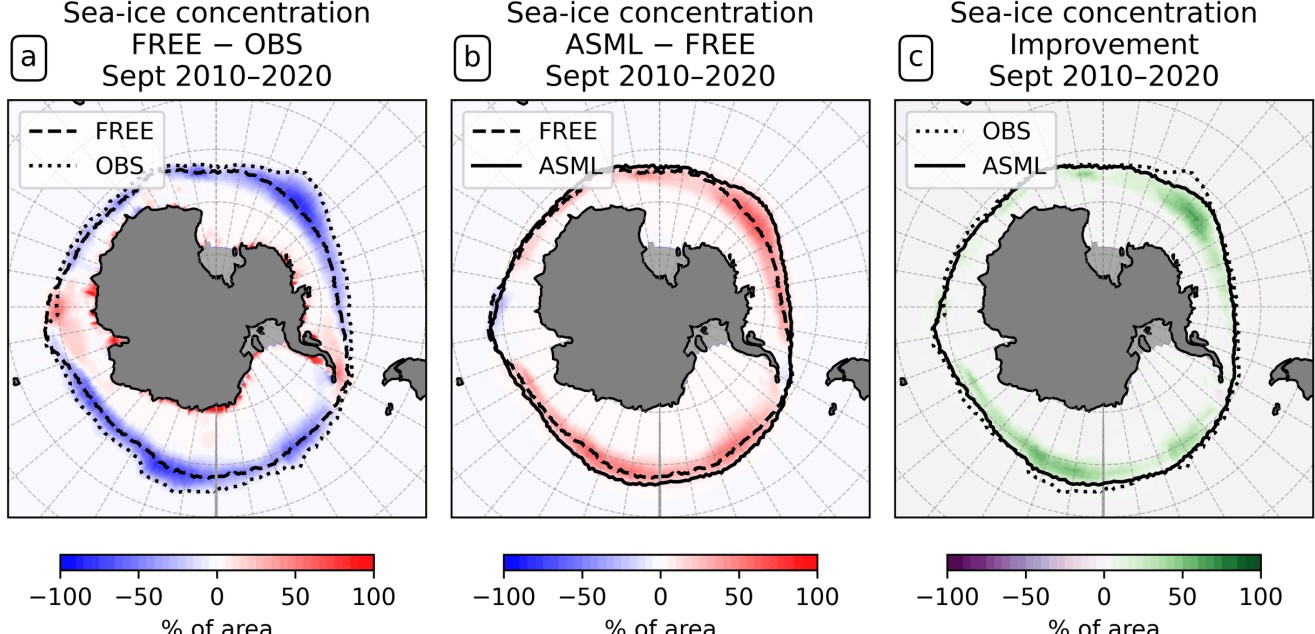

**Figure 3.** Effect of data assimilation on Antarctic sea-ice concentration in September. All panels show differences in the sea-ice concentration averaged for the month September over the period 2010-2020. The 15%-line for FREE, ASML and OSI-SAF observations is shown as a dashed, continuous or dotted line in panels a or b, respectively. (a) The difference between FREE and OSI-SAF observations. (b) The difference ASML − FREE. (c) The improvement of September mean sea-ice concentration.

sea-ice concentration difference for the month September (Fig. 3b). Through DA, a higher Antarctic sea-ice concentration is obtained. This improves the agreement with OSI-SAF (Fig. 3c). During all other seasons, the assimilation leads to a higher sea-ice concentration in the Antarctic, a larger sea-ice extent and a better agreement with OSI-SAF as well (only September is shown). In the Arctic, the differences between FREE, ASML and OSI-SAF are regionally different (not shown).

The boundary-layer depth and mixed-layer depth are mostly reduced through DA. In particular, deep water formation events characterised by a mixed-layer depth of more than 1000 m occur less frequently in ASML (not shown). This improves the agreement with the profile-observation based mixed-layer climatology of de Boyer Montégut et al. (2004), reducing the mean absolute difference to the climatology from 27 m to 19 m (comparison of mixer-layer depth in Fig. A6). In addition, the absolute difference of near-surface horizontal velocities to the drifter-observation based climatology of Laurindo et al. (2017)

is reduced by about 10% through DA (comparison of surface velocities in Fig. A7). The biological productivity near the equator is stable in ASML and FREE, indicating that FESOM2.1-REcoM3 does not suffer from the erroneous upwelling known from previous DA studies (Park et al., 2018). The meridional overturning, however, shows spurious structures, which may point to hidden assimilation artifacts on vertical velocities. Throughout the assimilation period, spurious, spatially limited and often deep overturning structures emerge, evolve through several months or years, and disappear in the tropical Indian,

Pacific and Atlantic basin (not shown). Thereby, the surface overturning cell sometimes breaks apart where it should extend over the equator, exposing the bottom cell to the surface (Fig. A8b). Transport in the North Atlantic at $26.5\,°$N, an indicator for the strength of the Atlantic Meridional Overturning Circulation, is between 8-9 Sv in FREE. In ASML, during the first two years of assimilation, transport at $26.5\,°$N decreases to below 3 Sv and, during the following years, recovers to 7-8 Sv (2016-2020). One possible cause is the effect of data assimilation on the eddy parameterisation (Gent and Mcwilliams, 1990).

The parameterised eddy activity is relevant for the dynamics in the deep ocean, and corrupting it may have a negative impact on the large-scale oceanic circulation, as described in Sidorenko (2004, Chapter 5.5 onwards) for a previous version of the ocean model FESOM.

In summary, the ASML temperature and salinity fields from the surface to several hundred meters below, and mixed-layer depth are in good agreement with observations, and the agreement of horizontal near-surface velocities with observations is

improved. This can be interpreted as an indication that the velocity field in the upper part of the ocean is also well represented. Although the spurious effects on deep ocean circulation should be further addressed in future work, we are confident that the DA provides an improved physical state in the upper ocean, which serves as an improved basis to estimate the air-sea $CO_2$ flux.

### 3.2  Effect of DA on global $CO_2$ flux

The ocean absorbs $2.78\,\mathrm{Pg\,C\,yr^{-1}}$ in ASML and $2.83\,\mathrm{Pg\,C\,yr^{-1}}$ in FREE during 2010-2020 (Fig. 4b), thus the assimilation

decreases the global mean oceanic $CO_2$ uptake by $0.05\,\mathrm{Pg\,C\,yr^{-1}}$. The temporal evolution of the annual global $CO_2$ flux is similar in ASML and FREE (Fig. 4a). The first assimilation year, 2010, stands out because it is one of the very few years during which the assimilation increases the oceanic $CO_2$ uptake. This slightly reduces the trend in $CO_2$ uptake 2010-2020 from $-0.40 \pm 0.09\,\mathrm{Pg\,C\,yr^{-1}\,dec^{-1}}$ in FREE to $-0.38 \pm 0.11\,\mathrm{Pg\,C\,yr^{-1}\,dec^{-1}}$ in ASML (negative: into the ocean). The trend, thereby, remains within its confidence interval. Furthermore, the assimilation reduces the interannual variability of the global

mean oceanic uptake slightly, demonstrated by a standard deviation of detrended annual means of $0.11\,\mathrm{Pg\,C\,yr^{-1}}$ in FREE and $0.08\,\mathrm{Pg\,C\,yr^{-1}}$ in ASML (not significantly different according to F-test). Through DA, the ensemble standard deviation of the global $CO_2$ flux is reduced from $1.0 \times 10^{-2}\,\mathrm{Pg\,C\,yr^{-1}}$ in FREE to $0.7 \times 10^{-2}\,\mathrm{Pg\,C\,yr^{-1}}$ in ASML in the year 2020.

The strongest time-mean air-sea $CO_2$ flux is found at mid and high latitudes (Fig. 4c). The large-scale pattern of the $CO_2$ flux is generally very similar in FREE and in ASML (FREE not shown). The largest local changes through DA, both towards

stronger or weaker $CO_2$ fluxes, occur in the North Atlantic in the area of the NAC and in the coastal North Pacific (Fig. 4d). The most prominent large-scale effect though, is in the Southern Ocean (Fig. 4e and f). South of $50\,°$S, the area-integrated $CO_2$ uptake increases by $0.18\,\mathrm{Pg\,C\,yr^{-1}}$ through the assimilation. In contrast, the uptake decreases by $0.07\,\mathrm{Pg\,C\,yr^{-1}}$ between 40-50$\,°$S. With the exception of the Southern Ocean, $CO_2$ uptake decreases in all world oceans by a small amount (Fig. 4d).

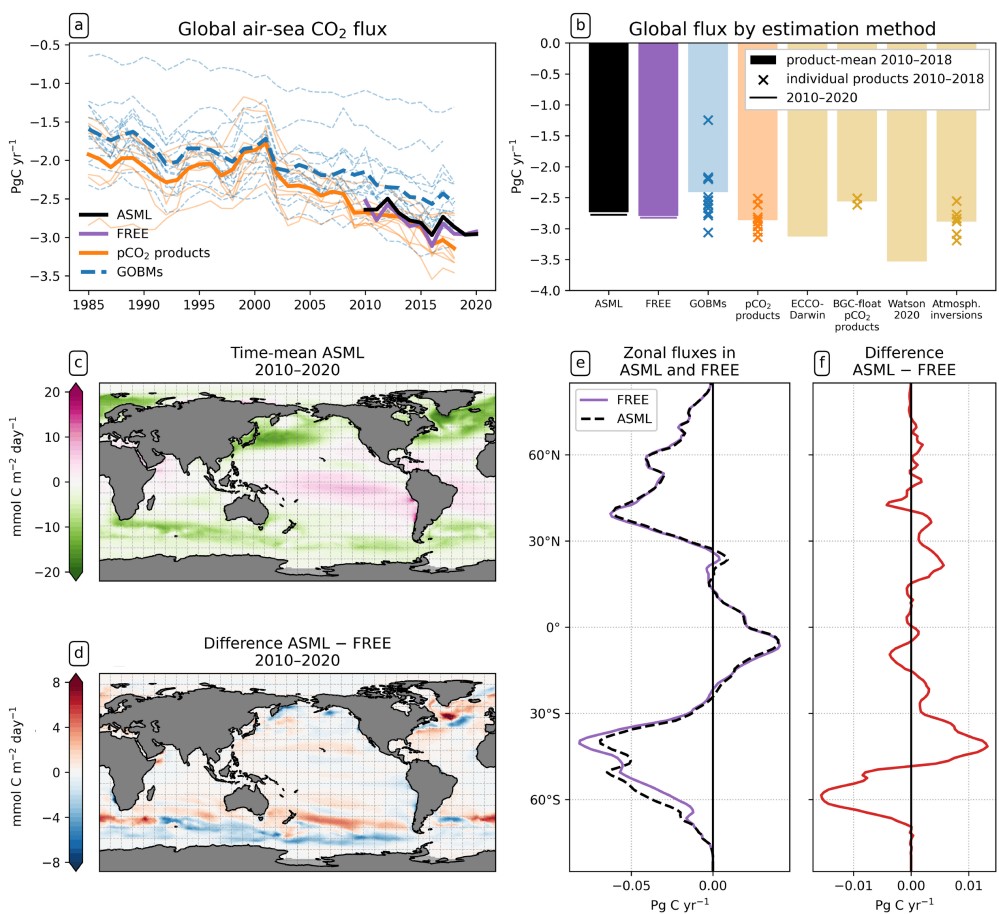

**Figure 4.** Effect of data assimilation on the air-sea $CO_2$ flux (negative: into the ocean). (a) Annual time-series of global flux in $\mathrm{Pg\,C\,yr^{-1}}$ in FESOM2.1-REcoM3 with ASML (black) and FREE (violet); and RECCAP2 estimates (DeVries et al., 2023) with $pCO_2$-products (orange) and GOBMs (blue) and their respective means (bold lines). Here, the river flux adjustment ($-0.65\,\mathrm{Pg\,C\,yr^{-1}}$) was applied to the $pCO_2$ products. (b) Time-mean global flux 2010-2018 in ASML (black), FREE (violet); and RECCAP estimates grouped by method (DeVries et al., 2023). Crosses represent individual estimates (e.g. individual GOBMs) and bars represent the method mean (e.g. mean of twelve GOBMs). Here, the river flux term was applied to all estimates except the models following the Global Carbon Budget methodology (Friedlingstein et al., 2023). For FESOM2.1-REcoM3, additionally the time-mean 2010-2020 is shown (horizontal lines). (c) Spatial distribution of $CO_2$ flux averaged over the period 2010-2020 in ASML. (d) Spatial distribution of $CO_2$ flux difference $\mathrm{ASML - FREE}$ averaged over the period 2010-2020 (e) Zonal averages of $CO_2$ flux 2010-2020 in ASML and FREE, and their difference in (f).

### 3.3 Effect of DA on regional CO$_2$ fluxes and their drivers

#### 3.3.1 Southern Ocean

In the Southern Ocean, the ocean takes up CO$_2$ in the annual average (Fig. 5a), with regionally heterogeneous effects of DA (Fig. 5b). While the effect of DA on surface pCO$_2$ and the air-sea CO$_2$ flux can almost entirely be explained by the combined variation of DIC and alkalinity at most latitudes north of $40\,°$S, the thermal effect also needs to be considered in the Southern Ocean (zonal mean pCO$_2$-effects in Fig. A9a). In the following, we examine how the assimilation influences the air-sea CO$_2$ flux across individual regions in the Southern Ocean.

**STSS$_{SO}$**   In the northernmost biome of the Southern Ocean, the subtropical seasonally stratified biome (STSS$_{SO}$), the mean oceanic CO$_2$ uptake is comparably high (Fig. 5a). The uptake is largest in austral winter and spring (June to November, Fig. 5c and d). The part of the STSS$_{SO}$ characterized by a positive CO$_2$ flux difference $ASML - FREE$ (positive difference: reduced uptake through assimilation), which we call the STSS$_{SO}$+, roughly forms an outer northerly ring around the STSS$_{SO}$ biome (hatched area in Fig. 5a and b). The reduction of CO$_2$ uptake in the STSS$_{SO}$+ is greatest in winter and spring from July to October (Fig. 5g).

The increase in pCO$_2$ in the STSS$_{SO}$+ is partly driven by lowered alkalinity and partly by increased surface DIC (Fig. 6b and c). These, as well as the colder SST and fresher SSS in the STSS$_{SO}$+ (Fig. 1b and e) are indications for a year-round stronger influence of subantarctic waters. This is evident from typical water properties in the subantarctic and subtropical Southern Ocean. In the subantarctic, surface DIC is higher, surface alkalinity is lower, temperature is colder and salinity is lower (maps of SST, SSS, DIC and alkalinity in Fig. A10). In the fragmented area of the STSS$_{SO}$+, different factors contribute to regional changes of the surface DIC and alkalinity budget in ASML (sources minus sinks of DIC and alkalinity in Fig. A11). Depending on location, an increased upward transport of DIC through mixing, an increase of DIC through a reduced biological sink of DIC in spring, or a decrease of alkalinity through changes in horizontal and vertical advection dominates. The seasonality of the effect of DA on the air-sea CO$_2$ flux in the STSS$_{SO}$+ (Fig. 5c and g) is determined by seasonal temperature differences between ASML and FREE (Fig. 6d and f). During summer, SST is slightly reduced (Fig. 6f), which lowers pCO$_2$ (Fig. 6a). This counteracts the effects of DIC and alkalinity on pCO$_2$ (Fig. 6b and c) and thus dampens the overall DA-effect on the air-sea CO$_2$ flux during summer.

The part of the STSS$_{SO}$ characterized by a negative CO$_2$ flux difference $ASML - FREE$, which we call the STSS$_{SO}$−, is a fragmented region and roughly consists of segments of an inner southerly ring (non-hatched area in Fig. 5a and b). Here, the increase of CO$_2$ uptake through DA is largest in summer and autumn (November to April, Fig. 5h). The reduction of pCO$_2$ is driven by increased alkalinity, and partly also by lower surface DIC (Fig. 6b and c, non-hatched area). These, together with higher SST in ASML than FREE in the STSS$_{SO}$− regions (Fig. 1b), indicate a higher presence of subtropical waters (see characteristics of subtropical waters in Fig. A10). Where there is lower DIC in the STSS$_{SO}$− in ASML (Fig. 6b), this can mostly be explained by an increased biological sink of DIC, with the addition of sharply defined local changes in horizontal advection of DIC and alkalinity (Fig. A11). Additionally, seasonal temperature effects occur. During winter, SST is higher in

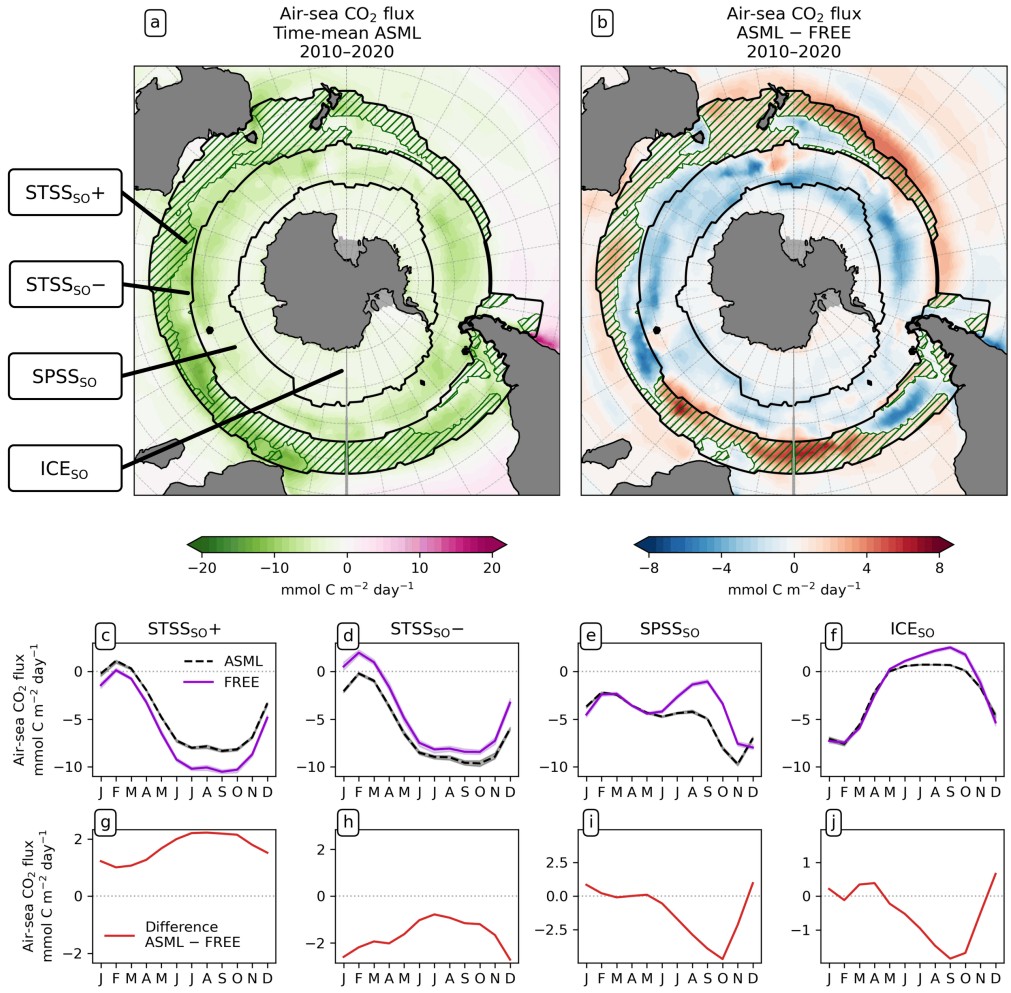

**Figure 5.** Effect of data assimilation on Southern Ocean $CO_2$ flux (negative: into the ocean) and its seasonality averaged over the period 2010-2020. Additionally, lines in a and b denote the regions, and the green hatching denotes the $STSS_{SO}+$. (a) Map of mean $CO_2$ flux in ASML. (b) Map of $CO_2$ flux difference $ASML - FREE$. (c - f) Seasonal cycle of air-sea $CO_2$ flux by region. Shading indicates the range of ensemble members in the year 2020. (g - j) Seasonal air-sea $CO_2$ flux difference $ASML - FREE$ by region. Note the different scales.

ASML than in FREE (Fig. 6e and g). This increases $pCO_2$ in the $STSS_{SO}$− (Fig. 6a), counteracting the effects of lower DIC and higher alkalinity on $pCO_2$ and dampening the overall DA-effect during winter.

The contrasting effects in the $STSS_{SO}$ indicate a horizontal shift of water masses within the $STSS_{SO}$ biome. In the center of the $STSS_{SO}$, the Subantarctic Front is located, which is associated with the Antarctic Circumpolar Current (ACC) and characterized by a strong gradient in SST, SSS and various other tracers (Chapman et al., 2020). Because SST and SSS are directly influenced and improved by the assimilation, the position of this front is also expected to change as a result of the assimilation, leading to a horizontal relocation of waters separated by the front. With the relocation of the front, dynamic shifts in regional characteristics occur, such as the amount of DIC and alkalinity transported vertically through mixing, and biological sources and sinks of DIC and alkalinity.

**SPSS$_{SO}$**  Further south, in the subpolar seasonally stratified biome (SPSS$_{SO}$), the ocean absorbs $CO_2$ all year-round (Fig. 5a). The oceanic uptake is increased through the assimilation, shown by a negative flux difference $\mathrm{ASML - FREE}$ in Fig. 5b. The largest difference between ASML and FREE is seen in spring from September to October (Fig. 5i). Due to the seasonally varying effect of DA, the seasonal cycle of the $CO_2$ flux in the SPSS$_{SO}$ is altered. In ASML, the $CO_2$ uptake is weakest in February, gets stronger in autumn (MAM), stagnates in winter (JJA) and resumes to grow in spring (SON), reaching peak uptake in November (Fig. 5e). In FREE, the $CO_2$ uptake weakens in winter, is weakest in September and gets stronger afterwards, reaching peak uptake in December.

In the SPSS$_{SO}$, the increased $CO_2$ uptake and lower surface $pCO_2$ during winter and spring is driven by a combination of colder temperatures and lower DIC (Fig. 6a and b), which outweighs the opposite effect of a decrease in alkalinity on $pCO_2$ (Fig. 6c, relative importance of thermal effect in Fig. A12a). Surface DIC is generally high due to upward transport of carbon-rich deep water (e.g. Hauck et al., 2023a). The reason for lower surface DIC in ASML is that the upward transport through mixing is reduced (Fig. A11) through a more stable stratification, which is also evident from a reduced density in the upper 300 m and an increased density below that (Fig. 6h). Thereby, the densities in the SPSS$_{SO}$ agree better with densities calculated from EN4-OA. Boundary layer and mixed layer in winter and spring are shallower and thereby in better agreement with the observation-based climatology (Fig. A6). Vertical mixing within the boundary layer affects the vertical profiles of DIC and alkalinity, towards lower DIC in ASML above 100 m and higher DIC below (Fig. 6i). The vertical profile of DIC in ASML is closer to GLODAP DIC observations, albeit some differences to GLODAP still exist. Besides the fact that the differences in stratification and boundary-layer depth affect the vertical DIC profile, they also imply less available surface nutrients in ASML. Probably due to a combination of lower nutrient availability and colder surface temperature, ASML features lower NPP, lower chlorophyll concentrations and a lower phytoplankton biomass in the SPSS$_{SO}$ (not shown). Thereby, the modeled biogeochemical cycle adjusts to the lower transport of nutrients to the surface by transferring less organic material to depth, ultimately acting to compensate about 60% of the difference in physical transport of DIC (Fig. A13a) and adding to the reduction in surface alkalinity (Fig. A13b). Within the SPSS$_{SO}$ (roughly south of $50\,°S$), differences between FREE and ASML in terms of the temperature effect on $pCO_2$, vertical transport of DIC and alkalinity and biological sources and sinks are larger than at any other latitude (Fig. A13).

**ICE$_{SO}$**   In the seasonally ice-covered biome (ICE$_{SO}$) surrounding the Antarctic continent, the time-mean $CO_2$ flux is smaller than in other biomes (Fig. 5a). In this region, the ocean absorbs $CO_2$ during summer and there is a smaller outgassing during winter (Fig. 5f), as the region is mostly ice-covered in winter (see sea-ice concentration in September in Fig. 3). In the northern part of the ICE$_{SO}$ biome, close to the SPSS$_{SO}$, the effect of the assimilation is similar to the effect within the SPSS$_{SO}$ itself (Fig. 5b). Here, the assimilation acts to increase ocean $CO_2$ uptake or to weaken $CO_2$ outgassing during winter and spring (Fig. 5i and j). Thereby, interestingly, the assimilation hinders outgassing of $CO_2$ from May to November in ASML in the ICE$_{SO}$ biome (Fig. 5f; comparison of winter outgassing with other estimates in Fig. A14). The reduced outgassing and decreased $pCO_2$ during winter and spring is driven by similar processes as within the SPSS$_{SO}$. Again, lower surface DIC and colder temperatures (Fig. 6a and b) outweigh the opposite effect of a decrease in alkalinity on $pCO_2$ (Fig. 6c). As in the SPSS$_{SO}$, the reason for the decrease in $pCO_2$, is reduced surface DIC and increased DIC below 100 m as a result of less upward transport of DIC through mixing (Fig. A11) in a more stable stratification due to surface freshening (Fig. 1e). In addition, as the surface temperature is lower in ASML (Fig. 1b), the winter sea-ice concentration is higher (Fig. 3b), which prevents winter outgassing of $CO_2$. In the southern part of the ICE$_{SO}$ biome, near the Antarctic continent, the effect of the DA on the $CO_2$ flux is small.

In summary, in the Southern Ocean, the main effects of the DA on the $CO_2$ flux are, firstly an increase of the uptake in the SPSS$_{SO}$ caused by surface cooling and by a more stable stratification and thus less upward transport of naturally carbon-rich water through mixing, and secondly an overall lower $CO_2$ uptake in the STSS$_{SO}$ as a consequence from a spatial redistribution of fluxes near the Subantarctic Front.

### 3.3.2   North Atlantic

In the North Atlantic, the assimilation has noticeable effects on the $CO_2$ flux in the area of the North Atlantic Current, where the ocean absorbs $CO_2$ in the annual average (Fig. 7a). During summer however, the ocean releases $CO_2$ while the sea surface warms (Fig. 7c-f). In the Central STSS$_{NA}$−, the effect of the DA is to prevent outgassing during summer (Fig. 7c and g). In the Western STSS$_{NA}$+ and in the Newfoundland Basin$_{NA}$+, the ocean $CO_2$ uptake is decreased during winter (Fig. 7d, e, h and j). The regionally different dynamics of the effects of the assimilation that drive these differences in the air-sea $CO_2$ flux in the North Atlantic are investigated next.

**Central STSS$_{NA}$−**   In the Central STSS$_{NA}$−, the effect of the DA is overall towards a more negative flux of $CO_2$ from May to November (Fig. 7g). Thus, spring and autumn $CO_2$ uptake are increased and summer outgassing is prevented in ASML (Fig. 7c). The reason for decreased surface $pCO_2$ is higher alkalinity in ASML (Fig. 8c). In this region, the alkalinity effect, which reduces $pCO_2$, outweighs the opposing effects of DIC and SST on $pCO_2$ (Fig. 8a and b). A higher alkalinity could point to the presence of waters of subtropical origin transported northward with the NAC (Völker et al., 2002). Other fingerprints of waters transported by the NAC are a warm SST particularly in winter, a higher salinity and higher DIC than that of North Atlantic subpolar waters (maps of SST, SSS, DIC, alkalinity in Fig. A15; Völker et al., 2002). The assimilation causes a change in these properties, towards a higher SST, higher salinity and higher DIC in the Central STSS$_{NA}$−. Simultaneously, ASML represents a deeper boundary layer in this region (Fig. 8d). While changes in the North Atlantic mixed-layer depth

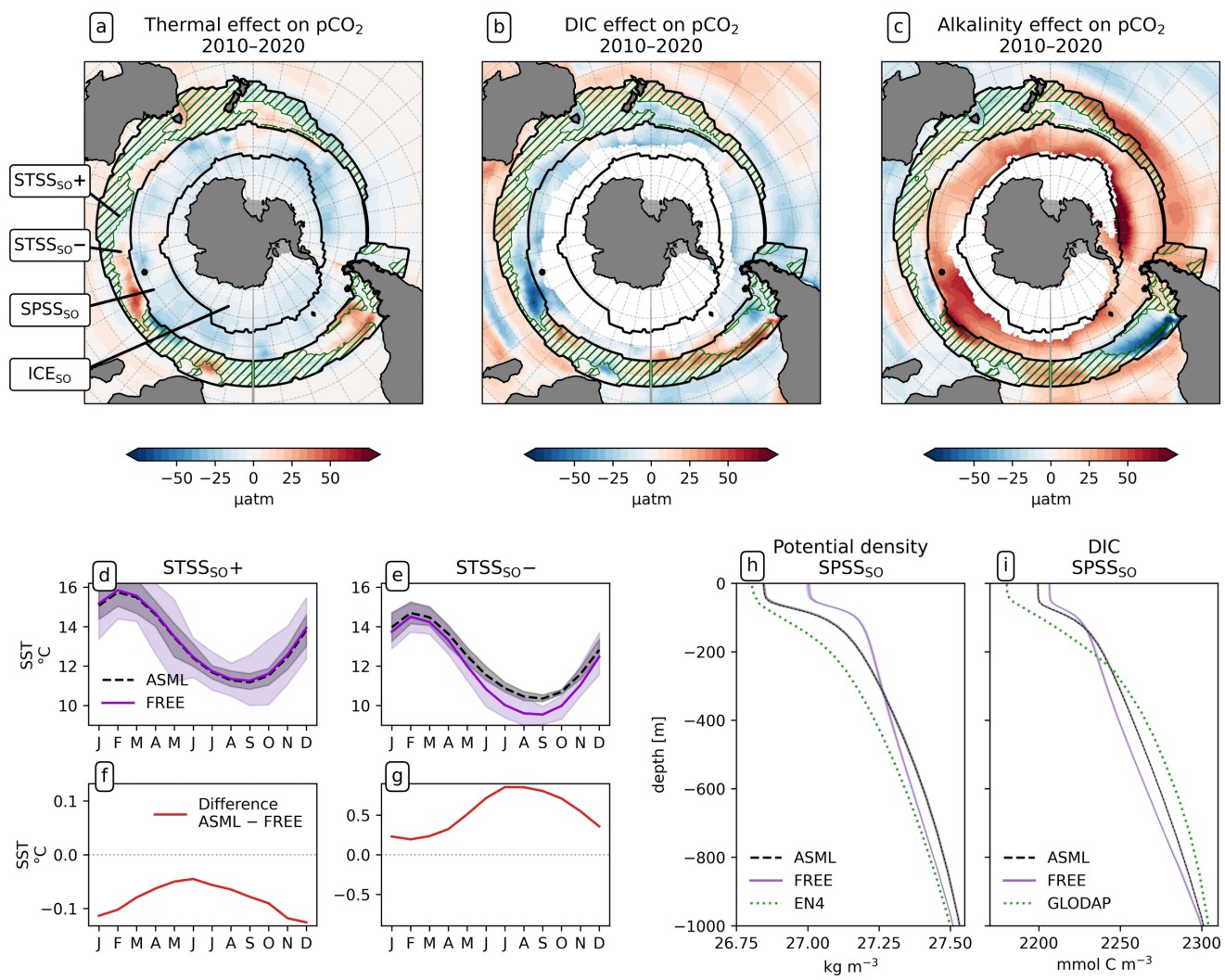

**Figure 6.** Drivers of the effects of data assimilation on pCO$_2$ in the Southern Ocean. Panels a, b and c show the effects of SST, DIC and alkalinity differences ASML − FREE simulations on surface pCO$_2$. Additionally, hatching inside the STSS$_{SO}$ indicates where net pCO$_2$ is increased through the assimilation (STSS$_{SO}$+). (d and e) Seasonal cycle of SST averaged over the regions STSS$_{SO}$+ and STSS$_{SO}$−, and (f and g) the difference ASML − FREE for each region. (h) Potential density profiles for the SPSS$_{SO}$, with FREE (violet line) and ASML (dashed black line) based on daily T and S, and with EN4-OA (dotted green line) based on monthly T and S. (i) DIC profiles for the SPSS$_{SO}$, showing FREE (violet line), ASML (dashed black line) from 2010-2020 and climatological DIC from GLODAP. Shading in d, e, h and i indicates the range of ensemble members in the year 2020.

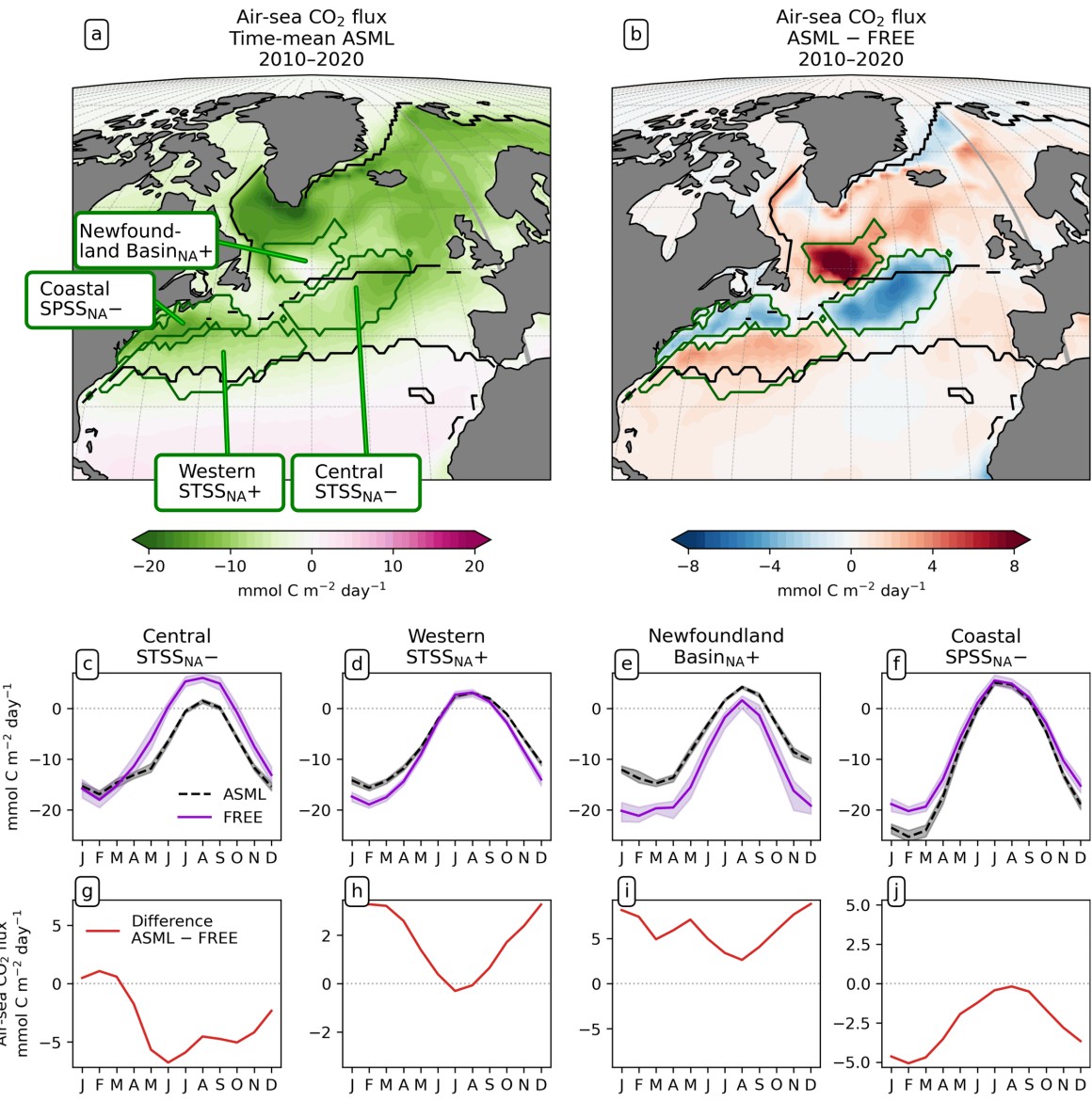

**Figure 7.** Effect of data assimilation on North Atlantic $CO_2$ flux (negative: into the ocean) and its seasonality averaged over the period 2010-2020. (a) Map of mean $CO_2$ flux in ASML. (b) Map of $CO_2$ flux difference $ASML - FREE$. (c - f) Seasonal cycle of air-sea $CO_2$ flux by region. Shading indicates the range of ensemble members in the year 2020. (g - j) Seasonal air-sea $CO_2$ flux difference $ASML - FREE$ by region. Note different scales.

overall result in a spatial pattern in ASML that more closely aligns with the pattern in the observation-based mixed-layer climatology, the modeled mixed layer in the simulations is still overall deeper than in the climatology, leading to less agreement in the Central $STSS_{NA}-$ (Fig. A6). Likely facilitated by higher SST and more available nutrients through deeper mixing in winter and spring, ASML features a higher biological sink of DIC above 190 m (Fig. A16d), more biological carbon export through sinking of detritus at 190 m, more column integrated phytoplankton biomass and surface chlorophyll in spring, which is illustrated by the example of surface chlorophyll difference between ASML and FREE in Fig. 8e. In combination, the higher alkalinity associated with NAC transport and the higher biological sink of DIC result in lowered surface $pCO_2$ and higher oceanic uptake.

**Western $STSS_{NA}+$**    In the Western $STSS_{NA}+$, the DA reduces the $CO_2$ uptake and increases $pCO_2$ mainly during winter, as a direct effect of increased SST (Fig. 8a). The direct thermal effect is dominant over the combined effect of DIC and alkalinity (relative importance of thermal effect in Fig. A12b). The latter have effects comparable in magnitude to SST, but mostly cancel each other out (Fig. 8b-c). The effect of DA on surface properties (SST, SSS, DIC and alkalinity) in the Western $STSS_{NA}+$ is similar to the effect in the Central $STSS_{NA}-$, which indicates a higher influence of subtropical waters in both regions.

**Newfoundland Basin$_{NA}+$**    In the Newfoundland Basin$_{NA}+$, the dominant effect of DA is a reduction of the $CO_2$ uptake and an increase of $pCO_2$ mainly during winter, as a direct effect of increased SST (Fig. 8a). In addition, ASML also features a more stable stratification due to lower density at the surface than FREE (Fig. 8f), which mostly affects DIC at 50-400 m depth through reduced subduction of DIC (Fig. 8g). Furthermore, ASML represents less surface chlorophyll in the Newfoundland Basin$_{NA}+$ (Fig. 8e) as a result of a redistribution of biomass from the surface to 50-400 m depth due to spring mixing (not shown). The downward mixing of biomass results in an increase of the biological sink of DIC above 50 m likely due to more primary production near the surface, and a decrease of the biological sink at 50-400 m likely due to more remineralization at this depth. However, the differences in the biological sink of DIC are compensated by mixing of DIC (profiles not shown). Overall, differences of the regional DIC profile to the observational GLODAP climatology slightly increase (Fig. 8g).

**Coastal $SPSS_{NA}-$**    In the Coastal $SPSS_{NA}-$, $pCO_2$ is reduced and the ocean $CO_2$ uptake is increased in ASML during winter and spring (Fig. 7f and j). The reduction of $pCO_2$ is facilitated by colder SST (Fig. 8a). This might be due to subpolar water masses penetrating further south along the coast in ASML because the location where the current separates from the coast is further south in ASML (velocities in Fig. A2).

In summary, DA affects the $CO_2$ flux in the North Atlantic mainly through changes in SST, combined with changes in horizontal advection of DIC and alkalinity near the NAC. Changes in the vertical mixing of DIC and alkalinity are largely compensated by feedbacks in biogeochemical cycles. Which of these effects is dominant, however, varies from region to region.

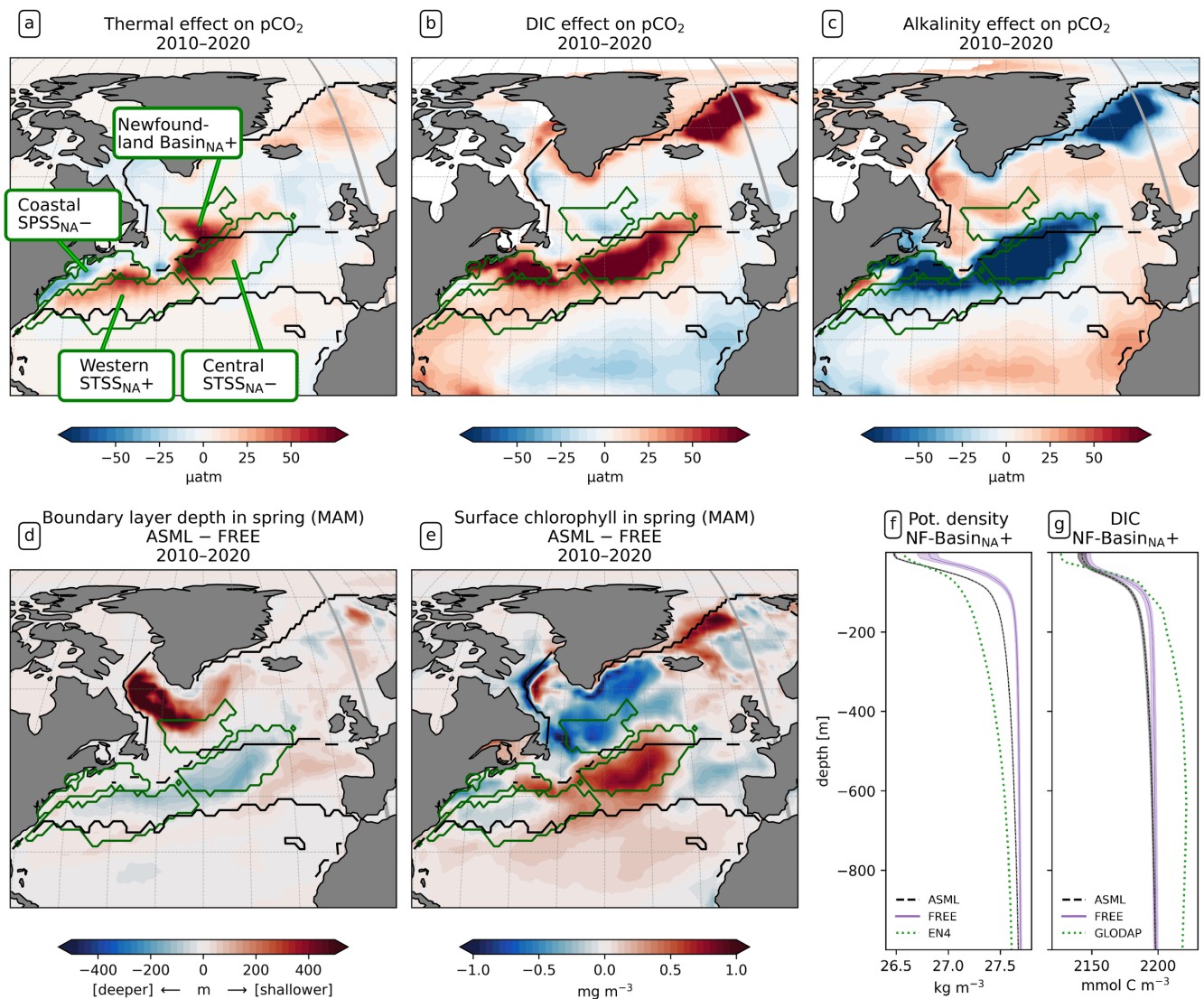

**Figure 8.** Drivers of the effects of data assimilation on $pCO_2$ in the North Atlantic. Panels a, b and c show the effects of SST, surface DIC and alkalinity differences $ASML - FREE$ on surface $pCO_2$. (d) Difference of boundary layer depth ($ASML - FREE$) for spring (MAM) 2010-2020, where positive denotes a shallower boundary layer in ASML. (e) Difference of surface chlorophyll ($ASML - FREE$) for spring (MAM) 2010-2020. (f) Potential density profiles for the Newfoundland Basin$_{NA}$+ region, with FREE (violet line) and ASML (dashed black line) based on daily T and S, and with EN4-OA (dotted green line) based on monthly T and S. (g) DIC profiles for the Newfoundland Basin$_{NA}$+ region, showing FREE (violet line), ASML (dashed black line) from 2010-2020 and climatological DIC from GLODAP. Shading in f and g indicates the range of ensemble members in the year 2020.

### 3.4 Comparison with biogeochemical observations

#### 3.4.1 pCO$_2$ (SOCAT)

To evaluate the modeled air-sea CO$_2$ flux based on observations, surface pCO$_2$ is the most informative variable, as it is closely related to the air-sea CO$_2$ flux. Effects of the DA on the modeled ecosystem and associated carbon fluxes, as well as thermal and dynamical effects that affect the CO$_2$ flux, are all included in pCO$_2$. The global mean of absolute monthly model-observation differences to the available SOCAT pCO$_2$ observations is $27.26\,\mu$atm for FREE. For ASML, the difference is slightly larger with $27.60\,\mu$atm. On global average, pCO$_2$ is higher than in SOCAT by $3.70\,\mu$atm in FREE and $4.59\,\mu$atm in ASML, as regions with positive and negative differences to SOCAT compensate (Fig. 9a). As an illustration of the regional changes through DA, the absolute differences in pCO$_2$ amount to $8.08\,\mu$atm (absolute difference ASML-FREE calculated at every grid point then averaged globally), which is $\pm27\%$ of the mean absolute model-observation difference. A linear offline estimation demonstrates that this change in pCO$_2$ would lead to an absolute change in the air-sea CO$_2$ flux by $1.06\,\mathrm{mmol\,C\,m^{-2}\,day^{-1}}$ on average (Equation 8).

Overall, FREE and ASML show very similar regional pCO$_2$ differences compared to SOCAT (difference of FREE and SOCAT in Fig. 9a; difference of ASML and SOCAT not shown). In the subtropical and tropical Atlantic and the subtropical Pacific, FREE and ASML have higher pCO$_2$ than SOCAT, while in the equatorial Pacific, pCO$_2$ is lower. At high latitudes, FREE and ASML represent mostly lower pCO$_2$ than SOCAT.

In the Southern Ocean, the simulations represent lower pCO$_2$ than SOCAT in the SPSS$_{\text{SO}}$ and ICE$_{\text{SO}}$ biomes in the annual mean (Fig. 9c), which is dominated by summer differences to SOCAT (not shown) when most observations are available. Through the assimilation, pCO$_2$ is slightly increased in summer and mostly reduced in winter (not shown), leading to an overall better agreement with SOCAT (Fig. 9e). In contrast, in the STSS, FREE and ASML represent higher pCO$_2$ than SOCAT, and through the assimilation, the agreement with SOCAT decreases.

In the North Atlantic, the simulations and SOCAT show a similar large-scale pattern, namely that pCO$_2$ is higher in the subtropics (ASML: around $400\,\mu$atm) than in the subpolar regions (ASML: around $280\,\mu$atm). Yet, this latitudinal difference of pCO$_2$ is stronger in the simulations compared to SOCAT, meaning that in the subtropics, pCO$_2$ in the simulations is higher than in SOCAT (Fig. 9d), while it is lower in the subpolar regions. Furthermore, in both simulations there is a pronounced pCO$_2$ surface gradient in the NAC and North Atlantic Subpolar Gyre region, whose position is changed by the assimilation, and which appears to be further northward in SOCAT. Thereby, the assimilation overall leads to a better agreement with SOCAT, in particular through a decrease of pCO$_2$ in the Central STSS$_{\text{NA}}-$, where the average difference is reduced from $26\,\mu$atm (FREE - SOCAT) to $1\,\mu$atm (ASML - SOCAT). However, in the Newfoundland Basin$_{\text{NA}}+$, the average difference is reversed from $-17\,\mu$atm (FREE - SOCAT) into $13\,\mu$atm (ASML - SOCAT), which is associated with a larger absolute discrepancy of ASML and SOCAT.

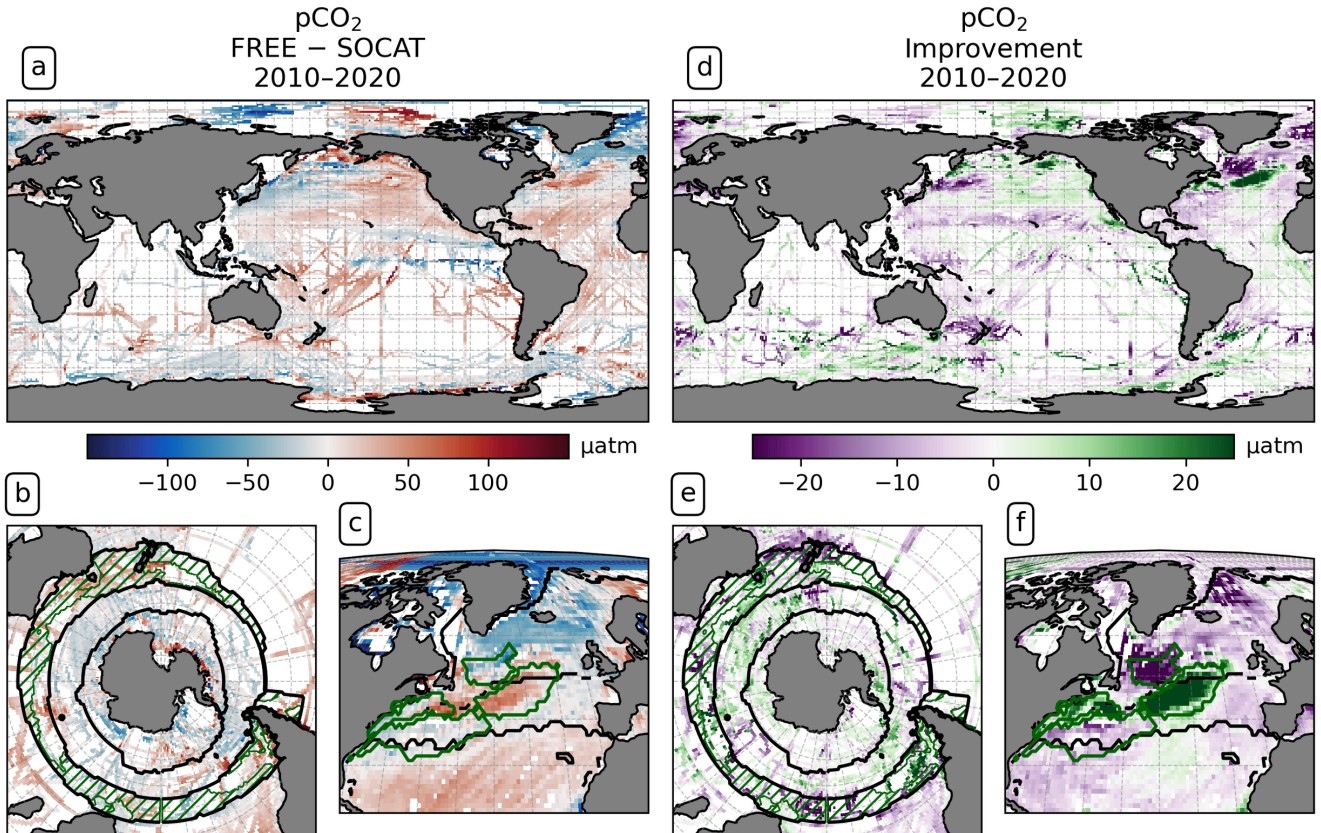

**Figure 9.** Partial pressure of $CO_2$ ($pCO_2$) at the surface averaged over the years 2010-2020. Panels (a-c) show the difference between FREE and SOCAT observations in (a) the global ocean, (b) Southern Ocean and (c) North Atlantic; panels (d-f) show the impact of the assimilation as 'improvement' relative to SOCAT observations computed from monthly mean $pCO_2$ in the same regions. Positive values (green color) denote a reduced difference to SOCAT.

### 3.4.2 DIC and alkalinity (GLODAP)

DIC and alkalinity are two of the most important variables from which $pCO_2$ is derived (Section 3.3). Comparing them with observations provides more insights into the strengths and weaknesses of the modeled carbonate system than a comparison with $pCO_2$ observations alone. The FESOM2.1-REcoM3 simulations represent higher surface DIC than GLODAP bottle observations (Lauvset et al., 2024a, gridded monthly-means) on average (Fig. 10a), with a global mean surface difference FREE-GLODAP of $6.46\,\mathrm{mmol\,C\,m^{-3}}$ for DIC. Although fewer DIC observations are available than $pCO_2$ observations, similarities between the respective model-observations differences for DIC and $pCO_2$ can be recognized. For example, DIC in the model is lower in the tropical and subtropical Atlantic than GLODAP, and higher in the polar Atlantic. This is consistent with SOCAT $pCO_2$ observations in the same areas. The model-observation differences to GLODAP DIC and SOCAT $pCO_2$ are also consistent with each other in the north Pacific. The assimilation induces absolute changes in surface DIC of $6.33\,\mathrm{mmol\,C\,m^{-3}}$

on global average, with regional differences in sign. These changes slightly reduce the mean absolute difference to the surface observations from $32.78 \, \mathrm{mmol\,C\,m^{-3}}$ to $32.15 \, \mathrm{mmol\,C\,m^{-3}}$, and yield a mixed picture of the improvement (Fig. 10b).

While the trend in surface DIC due to anthropogenic input makes it necessary to compare the model with contemporaneous observations at the ocean surface, a comparison with climatological data is meaningful below a depth of approximately 200 m. In fact, the modeled global distribution of DIC at depth is overall similar to that in the GLODAP climatology for both simulations (zonal mean DIC surface to 1000 m depth in Fig. A17). For example, the model results and GLODAP data sets show that DIC is lowest in the isopycnals of the subtropical gyres ($2050 - 2150 \, \mathrm{mmol\,C\,m^{-3}}$; Fig. A17a) and that DIC mostly increases with depth and is higher in the Pacific ($2420 \, \mathrm{mmol\,C\,m^{-3}}$ at 1000 m in the North Pacific) than in the Atlantic ($2320 \, \mathrm{mmol\,C\,m^{-3}}$ below 3000 m in the South Atlantic). Yet, depending on the ocean basin and depth, there can be both negative and positive differences between the simulations and the GLODAP climatology, which are in the order of $20 \, \mathrm{mmol\,C\,m^{-3}}$ (Fig. A17c). On a global average, the assimilation leads to an increase in DIC between 200-600 m depth and a reduction of DIC between the surface and 200 m, with the largest effect in the upper 400 m (Fig. A17b). This leads to an improved agreement with the GLODAP climatology, with the largest global mean improvement at a depth of 400 m ($2.5 \, \mathrm{mmol\,C\,m^{-3}}$; Fig. A17d). Below 1000 m depth, the global mean absolute difference FREE-ASML of DIC and alkalinity is only $1 - 2 \, \mathrm{mmol\,m^{-3}}$ and is therefore substantially smaller than at the surface.

The comparison with GLODAP bottle alkalinity at the surface shows a similar spatial patterns as for DIC (see Fig. 10a and c). The magnitude of the bias is also comparable ($14 \, \mathrm{mmol\,Alk\,m^{-3}}$). The global mean of the absolute difference ASML-FREE of surface alkalinity is $7.72 \, \mathrm{mmol\,Alk\,m^{-3}}$. The assimilation leads to a reduction of the absolute difference of the model alkalinity to GLODAP from $34.34 \, \mathrm{mmol\,Alk\,m^{-3}}$ to $32.60 \, \mathrm{mmol\,Alk\,m^{-3}}$. Since the effects of physics assimilation on alkalinity and DIC are regionally consistent, regions of improved or deteriorated agreement with GLODAP often coincide for both variables (compare Fig. 10b and d). Because changes of DIC and alkalinity have an opposing effect on the $CO_2$ flux, it is likely that their correlation results in compensating effects. A linear estimate shows that the joint effect of DIC and alkalinity changes is responsible for a change in the $CO_2$ flux in the order of $1.22 \, \mathrm{mmol\,C\,m^{-2}day^{-1}}$ on average, and, globally integrated, the assimilation-induced changes in DIC and alkalinity lead to an estimated net increase of the air-sea $CO_2$ flux in the order of $0.50 \, \mathrm{Pg\,C\,yr^{-1}}$ (Equations 4, 3 and 8). However, this linear offline estimate is subject to a large uncertainty because regionally large effects with opposite sign lead to uncertainty in the global mean.

### 3.4.3 Surface chlorophyll (OC-CCI)

The representation of chlorophyll by the model is of interest as a proxy for primary production. Surface chlorophyll reflects the phytoplankton state and biomass, and therefore, effects of the DA on the biological model state can be seen in the total surface chlorophyll concentration. A comparison of the modeled surface chlorophyll with remotely-sensed chlorophyll from OC-CCI reveals that both simulations feature a higher surface chlorophyll concentration than OC-CCI (FREE-OBS in Fig. 11a and c). In FREE, the difference to OC-CCI is $0.02 \, \mathrm{mg\,m^{-3}}$ on global average, with low deviations in the tropics and an enhanced difference north of $30 \, °\mathrm{N}$ ($0.12 \, \mathrm{mg\,m^{-3}}$) and south of $30 \, °\mathrm{S}$ ($0.24 \, \mathrm{mg\,m^{-3}}$). Apart from this, both simulations capture the global distribution of chlorophyll well. The simulations show the seasonal maxima in each hemisphere around one month earlier in

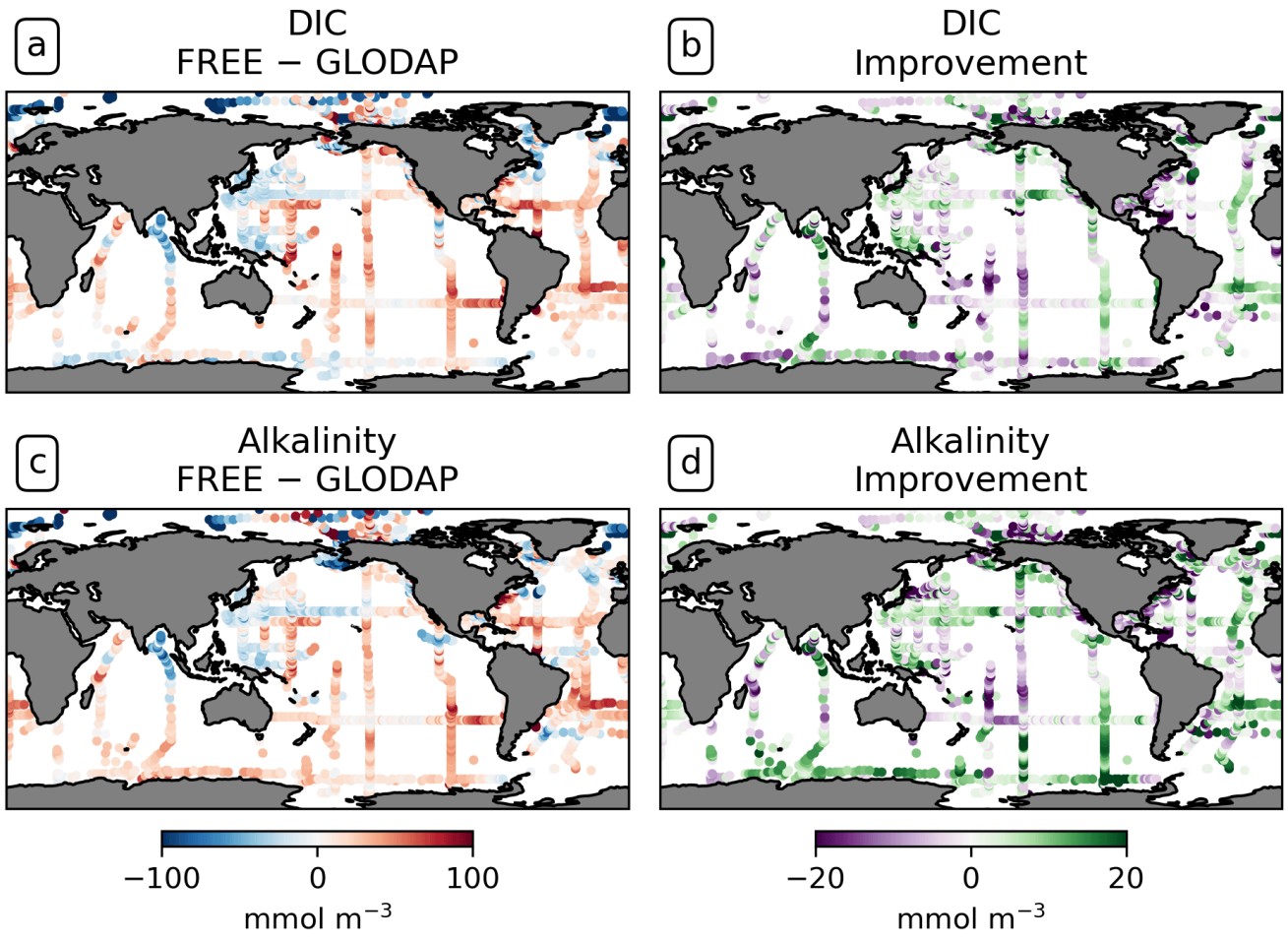

**Figure 10.** Comparison of the model results with surface DIC and alkalinity bottle observations from GLODAP globally over the years 2010 to 2020. (a) Surface DIC differences FREE − GLODAP. (b) Improvement of monthly surface DIC relative to GLODAP. (c and d) For alkalinity.

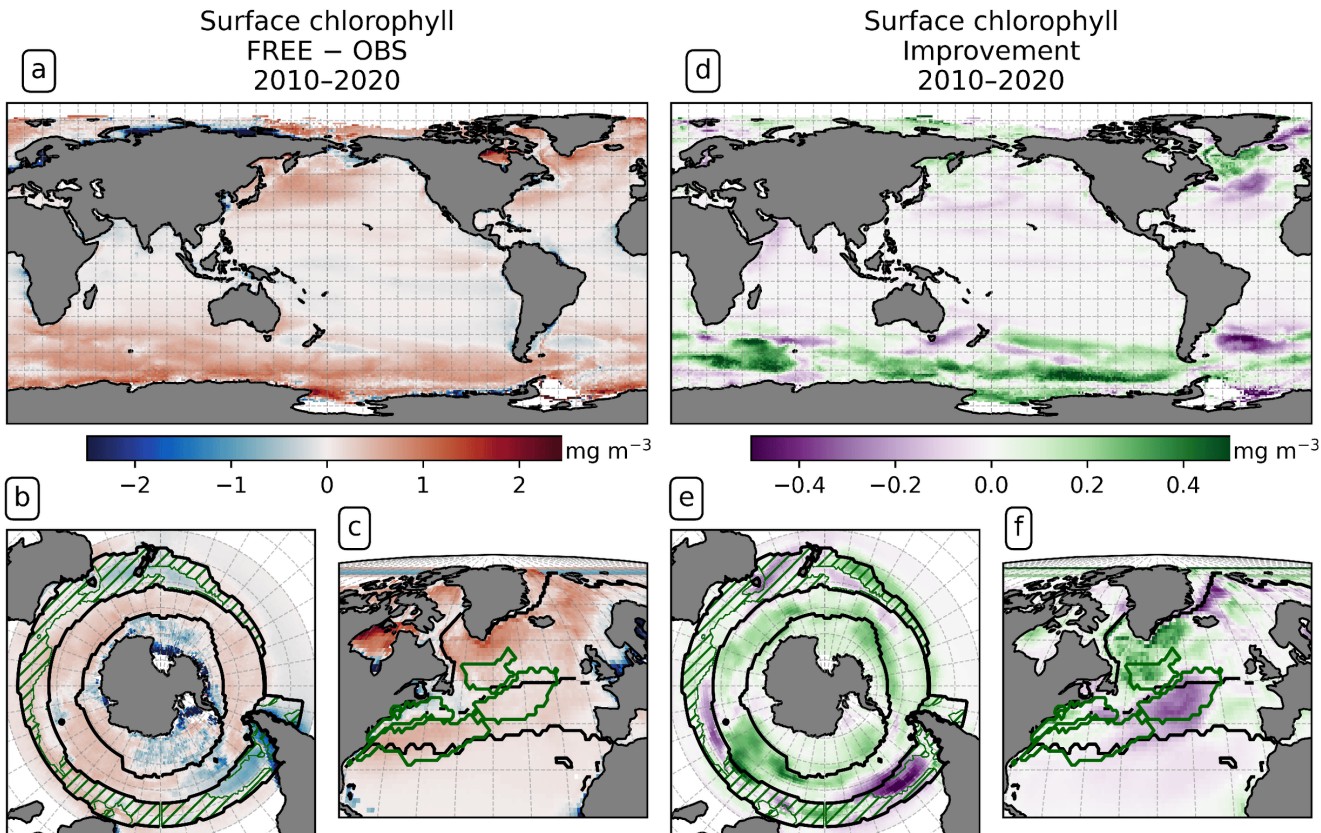

**Figure 11.** Surface chlorophyll for the years 2010-2020: (a-c) difference between FREE and SOCAT observations in (a) the global ocean, (b) the Southern Ocean and (c) the North Atlantic; (d-f) impact of the assimilation as 'improvement' relative to the observations in the same regions. Panels (a, c, d) and (f) compare to monthly OC-CCI observations, panels (b) and (e) refer to the climatology for 1998-2019 by Johnson et al. (2013).

the year (not shown). South of $30\,°$S, FREE is in better agreement with chlorophyll-a from Johnson et al.'s (2013) Southern Ocean specific chlorophyll product (Fig. 11b) than with OC-CCI data (Fig. 11a).

On global average, the assimilation slightly reduces the differences between model and OC-CCI data, from a global mean absolute difference of $0.31\,\mathrm{mg\,m^{-3}}$ to $0.29\,\mathrm{mg\,m^{-3}}$. The assimilation changes the chlorophyll concentration by an absolute value of $0.05\,\mathrm{mg\,m^{-3}}$ on average, which is 15% of the global mean absolute difference to OC-CCI. There are regions in which assimilation leads to a reduction in chlorophyll and thus to better agreement with the satellite products, for example in the North Atlantic Subpolar Gyre and the Southern Ocean $\mathrm{SPSS_{SO}}$ (Fig. 11e and f). In contrast, the model reacts to the DA with an increase in chlorophyll in the North Atlantic Central $\mathrm{STSS_{NA}}-$ and the Argentine Basin, which leads to poorer agreement.

## 4 Discussion

The improvement in temperature and salinity overall leads to a heterogeneous picture in biogeochemistry. While near-surface temperature and salinity fields are improved through DA almost everywhere, the global mean absolute difference of modeled surface $pCO_2$ to SOCAT remains similar in ASML compared to FREE, and this also applies to the model-observation differences for surface chlorophyll, DIC and alkalinity (Section 3.4). Where improvements in one BGC variable occur, these do not necessarily lead to consistent improvement in all BGC variables. For example, the representation of $pCO_2$ improves while that of chlorophyll deteriorates in the North Atlantic Central $STSS_{NA}-$ (Fig. 11f and Fig. 9f). In the Southern Ocean $SPSS_{SO}$, the reduction of modeled surface chlorophyll in spring and the increase of $pCO_2$ in summer lead to a better agreement with $pCO_2$ observations, yet the available observations of DIC and alkalinity do not resolve the regional scales to evaluate the corresponding changes in these variables (Fig. 9, Fig. 10 and Fig. 11f). The uncertainty represented by the ensemble is reduced by the DA, which has the most obvious effect on the directly assimilated fields (SST in Fig. 6d and e and density in Fig. 8f). The ensemble standard deviation of the $CO_2$ flux, where it is large in FREE, is constrained by the DA to globally more uniform and smaller values (Fig. 5c-f, Fig. 7c-f and Fig. A1). Only in the North Pacific, the standard deviation of $CO_2$ fluxes is equally high in ASML and FREE, precisely in a region that also presents a challenge for $pCO_2$ products (compare Fig. A1 and Mayot et al., 2024, Figure 5a). In the rest of the ocean, the reduced uncertainty represented by the ensemble does not necessarily coincide with improved agreement with BGC observations. One possible reason for improvement of model-data mismatch in one variable with worsening in another may lie in inconsistencies between the observational datasets. Another reason may be missing processes in the model and the use of constant BGC model parameters. Those parameters are responsible for linking changes between ecosystem variables and in reality, they vary across space and time depending on species composition in the ecosystem (Mamnun et al., 2023, Chapter 3). Overly simplified links between ecosystem variables can lead to canceling errors, which means that the state of one variable may worsen as a result of improving the other through DA (as in Ford and Barciela, 2017). For example, surface chlorophyll (Fig. 11f) and $pCO_2$ (Fig. 9f) in the central Greenland Sea deteriorate in response to improvements of SST (Fig. 1c), SSS (Fig. 1f) and sea-ice concentration (not shown). This could indicate that the BGC parametrization compensates for flaws in the free running physical model in this region. The parameter mismatch might cause difficulties in modeling the change of BGC variables under the ongoing loss of Arctic sea ice (Chen et al., 2016).

The major effects of physics DA on BGC variables seem to follow changes of SST and are largely uniform over the full period of DA (Section 3.4). Surface chlorophyll changes show a pattern similar to SST changes (Figs. 1 and 11). The modeled phytoplankton growth is temperature-dependent (Gürses et al., 2023). Furthermore, indirect temperature effects on plankton dynamics due to stratification and mixing changes contribute, albeit those can have heterogeneous effects and the correlation of chlorophyll and boundary-layer depth is less clear (not shown). The changes of surface DIC and alkalinity show similar spatial patterns with regional heterogeneity (Section 3.3), again with the major changes being coherent with the changes in SST (Fig. 1). Furthermore, the effects of the assimilation on DIC and on temperature in the upper 1000 m correlate regionally: Cooling through DA at intermediate depth (Fig. 2b) is usually accompanied by higher DIC in ASML (Fig. A17b), while warming through DA near the surface occurs together with reduced DIC in ASML. An overall more stable ocean stratification

in the upper hundreds of meters explains why. On global average, the assimilation leads to lower DIC above 200 m and higher DIC between 200-600 m depth. In regions of substantial DA effects on vertical transport of DIC, as for example in the Central $STSS_{NA}$− or in the $SPSS_{SO}$ (Section 3.3), the modeled biogeochemical cycles adjust dynamically to the altered

vertical transport. The resulting changes in biological sources and sinks of DIC compensate for 20-70% of the changes in vertical transport of DIC (Fig. A13a). In addition to changes in stability and mixing, the assimilation affects the distribution of DIC and alkalinity through local changes in near-surface horizontal transport. As the horizontal distribution of surface DIC, alkalinity and SST is governed by latitudinal gradients and common pathways of transport (Figs. A10 and A15), all of them undergo similar changes as the SST field is modified. An exception to this is in the $STSS_{SO}$, where regional shifts along

contrasting surface gradients of DIC, alkalinity and temperature affect the respective variables differently (Section 3.3). These shifts change the spatial pattern of air-sea $CO_2$ fluxes. With the exception of the Southern Ocean, zonally averaged changes in surface $pCO_2$ are dominated by the combined effects of surface alkalinity and DIC on $pCO_2$ (Fig. A9a). Because alkalinity and DIC are usually modified according to the same pattern through mechanisms acting on both, their effects on $pCO_2$ are anticorrelated (Fig. A9b). The direct thermal effect on $pCO_2$ can still be the largest locally, for example in the North Atlantic

Newfoundland $Basin_{NA}$+ (Fig. A12b). While the DA dynamically induces changes in surface $pCO_2$ everywhere, the strongest effects on the air-sea $CO_2$ flux are at high latitudes, where $pCO_2$ changes are amplified by high wind velocities.

The net effect of DA on the global air-sea $CO_2$ flux varies from year to year between $-0.12 \, \mathrm{Pg\,C\,yr^{-1}}$ and $0.15 \, \mathrm{Pg\,C\,yr^{-1}}$, which is small compared to the changes in regional $CO_2$ fluxes. The global net effect of lateral redistribution of alkalinity and DIC at the ocean surface is a result of compensation between regions where alkalinity and DIC are added and removed.

Similarly, regional SST effects on surface $pCO_2$ mostly balance out globally, because DA primarily induces a correction of regional SST biases, reducing the mean absolute difference to the observations from $0.59 \, ^\circ\mathrm{C}$ to $0.32 \, ^\circ\mathrm{C}$, rather than changing the global mean SST, which differs by only $0.02 \, ^\circ\mathrm{C}$ between FREE and ASML. DA-induced differences in vertical transport of DIC are comparably large south of $50 \, ^\circ\mathrm{S}$, but approximately 95% of them are balanced globally by opposing changes in vertical transport further north (vertical transport of DIC in Fig. A13a). In particular, the effect of DA on subduction of DIC through

vertical advection into the ocean's deeper layers (not shown), which is the rate-limiting step on oceanic uptake of anthropogenic $CO_2$ emissions (DeVries, 2022), appears small, which may be due to an insufficient amount of deep observations. Besides, experiments on longer time scales might be necessary to generate a visible effect of deep circulation changes on the ocean's carbon cycle (Cao et al., 2009), which could however lead to imbalances in the $CO_2$ flux (Lebehot et al., 2019; Kriest et al., 2020; Primeau and Deleersnijder, 2009). Another possible reason why the DA effect on the global $CO_2$ flux in our simulation

is small, is the variable stoichiometry in REcoM. The dynamic biological functioning reduces the sensitivity of critical fields, like DIC, to physical changes (Buchanan et al., 2018). Furthermore, negative feedback effects between surface alkalinity, DIC, atmospheric $pCO_2$ and air-sea fluxes might reduce the overall response (Bunsen et al., 2024).

The overall impact of the DA on the air-sea $CO_2$ flux on a global scale is modest ($0.05 \, \mathrm{Pg\,C\,yr^{-1}}$) compared to the differences between other estimates (e.g., a standard deviation of $0.45 \, \mathrm{Pg\,C\,yr^{-1}}$ of GOBMs in DeVries et al., 2023). The global air-sea

$CO_2$ flux estimates of FREE and ASML fall in the range of previous model estimates and in the range of previous $pCO_2$ products (Fig. 4a and b) for the period 2010-2018, during which comparable estimates are available (DeVries et al., 2023).

We compare here to two other data assimilating BGC model approaches, namely ECCO-Darwin (global; Carroll et al., 2020) and B-SOSE, which is restricted to the Southern Ocean (Verdy and Mazloff, 2017). Both approaches use Linearized Least Squares Optimization data assimilation methods (4D-var/adjoint and Green's function, Wunsch, 1996; Menemenlis et al., 2005). However, the largest difference to our study is probably that they assimilate BGC observations in addition to physical data. Thus, as expected, the effect on $pCO_2$ in our study is smaller (3%) than in ECCO-Darwin and B-SOSE where a reduction in $pCO_2$ model-data misfit of 6% and 64% was reported, respectively (here given as quadratic misfit). The global $CO_2$ flux (2010-2018) is smaller in FESOM2.1-REcoM3 ($-2.73\,\mathrm{Pg\,C\,yr^{-1}}$ in FREE and $-2.78\,\mathrm{Pg\,C\,yr^{-1}}$ in ASML) than in ECCO-Darwin ($-3.13\,\mathrm{Pg\,C\,yr^{-1}}$). The discrepancy between the $CO_2$ flux estimates based on models and $pCO_2$-products is an area of active research and not fully resolved (Friedlingstein et al., 2023; DeVries et al., 2023). On the one hand, model biases in the Atlantic Meridional Overturning Circulation, in Southern Ocean ventilation and possibly biases in the surface ocean carbonate chemistry were suggested as reasons why models might underestimate the global mean $CO_2$ uptake in recent decades (Friedlingstein et al., 2023; Terhaar et al., 2024, 2022). On the other hand, the sparsity of observations is a concern for the $pCO_2$ products. According to one testbed simulation, the $pCO_2$ products reflect the global mean and the seasonal cycle relatively well, while the decadal variability may be overestimated (Gloege et al., 2021). An overestimation of the decadal trend, as suggested by Hauck et al. (2023b), could explain the high estimates of the $pCO_2$ products for the present-day global mean $CO_2$ flux. In contrast, for the North Atlantic, it was argued that $pCO_2$ is comparatively well constrained by observations in the last decade but not in the 1980s, which has an erroneous influence on the long-term trend (Pérez et al., 2024).

The effects of data assimilation on the $CO_2$ flux are most pronounced in the Southern Ocean $STSS_{SO}$ and $SPSS_{SO}$ in winter. Verdy and Mazloff (2017) also found the largest effects of assimilation on the $CO_2$ flux in this region. Although the region is of crucial importance for the global ocean carbon sink, it also has the greatest uncertainty due to the lack of ship-based winter observations (Friedlingstein et al., 2023; Hauck et al., 2020). In the last decade, the number of winter observations has increased due to the introduction of biogeochemical Argo floats (Johnson et al., 2017; Williams et al., 2017), although the float-based $pCO_2$ derived from pH measurements and estimated alkalinity is subject to higher uncertainty compared to direct $pCO_2$ measurements (Williams et al., 2017; Bakker et al., 2016). Machine learning approaches incorporating BGC Argo float observations suggest a stronger winter outgassing around Antarctica, particularly south of $50\,°$S in the $SPSS_{SO}$ and $ICE_{SO}$ biomes, for 2015-2017 (Bushinsky et al., 2019; Gray et al., 2018). This results in a lower estimate of annual Southern Ocean $CO_2$ uptake in the float products. One suggestion in the literature is that model inadequacies in the representation of mixing and upwelling in the Southern Ocean might cause the discrepancy between float products and models (Gray et al., 2018). However, improvements in the modeled ocean physics and changes in mixing through data assimilation do not lead to closer agreement between the FESOM2.1-REcoM3 estimate and the float products (comparison of FESOM2.1-REcoM3, float products and B-SOSE in Fig. A14). In contrast, ASML shows even weaker winter outgassing and stronger summer uptake south of $50\,°$S than FREE, which brings the FESOM2.1-REcoM3 estimate further away from the float products. However, ASML is brought close to B-SOSE in terms of winter outgassing in the Antarctic polar ocean south of $60\,°$S and winter uptake in the $STSS_{SO}$ around $40\,°$S. Additionally, airborne $CO_2$ flux estimates and direct $pCO_2$ measurements stemming from a sail drone have questioned

the estimates of winter outgassing based on the BGC floats, either attributing the high $pCO_2$ values to possible biases in the floats' measuring devices or to anomalously high $pCO_2$ in the years 2015-2016 (Long et al., 2021; Sutton et al., 2021).

## 5 Conclusion

We apply data assimilation of temperature and salinity into a global ocean-biogeochemical model to improve the physical state for the years 2010-2020. The simulation is then assessed with regard to the effects on the biogeochemical variables. The experiments show that the effect of data assimilation (DA) on biogeochemical variables is mostly related to temperature changes. While the air-sea $CO_2$ flux and $pCO_2$ are directly affected by sea surface temperature, the DA also induces indirect changes to $pCO_2$ through dissolved inorganic carbon (DIC) and alkalinity. Globally integrated, these are more relevant for $pCO_2$ than the direct temperature effect. Yet, which of these factors has a dominant effect on $pCO_2$ varies locally. The assimilation leads to regional shifts in areas of $CO_2$ outgassing and uptake. Local effects on the air-sea $CO_2$ flux are particularly large in dynamic regions such as the North Atlantic Current and near the Subantarctic Front, whose pathways are challenging for the model to resolve without DA. The largest effect on the air-sea $CO_2$ flux occurs in the Southern Ocean during winter. In the simulation with assimilation, the uptake south of $50\,°\mathrm{S}$ is increased due to shallower mixing and surface cooling, and the uptake northward of that (40-50 °S) is weakened. In this area of the ocean, the uncertainty in current estimates of $CO_2$ fluxes is particularly high. Overall, the uncertainty inherent to the biogeochemical model appears to be larger than the uncertainties induced through physical biases in the free running model. Locally, the changes in surface $pCO_2$, chlorophyll, alkalinity, and DIC caused by the assimilation range between about 15 and 30% of the mean absolute model-observation difference. Yet, local improvements in one variable do not necessarily come along with improvements across other observed biogeochemical variables. Therefore, globally, physics DA does not generally improve the difference between the model and observations. In total, the effect of physics DA on the global ocean carbon uptake is with $0.05\,\mathrm{Pg\,C\,yr^{-1}}$ small compared to the spread between previous estimates of models, $pCO_2$ products and other DA estimates. While the assimilation of temperature and salinity improves the representation of these two and also of mixed-layer depth, sea-ice concentration and horizontal near-surface velocities, possible errors in the vertical velocities and overturning circulation are not eliminated. Further biogeochemical variables are only indirectly affected. To this end, the additional assimilation of biogeochemical observations is an obvious next step to reduce the uncertainty stemming from the ecosystem model and to improve the model-observation differences for biogeochemical variables.

*Code and data availability.* The code used to perform the free simulation and the data assimilation is available at 10.5281/zenodo.11495274. This code archive additionally contains Jupyter Notebooks to produce the manuscript figures from the model output. The processed model output data underlying the figures of this manuscript are available at 10.5281/zenodo.11495081.

## Appendix A

**Table A1.** List of tracers in REcoM3

| Tracers in REcoM3 |
| --- |
| Dissolved inorganic nitrogen and carbon (DIN, DIC) |
| Dissolved organic nitrogen and carbon (DON, DOC) |
| Alkalinity |
| Oxygen |
| Iron |
| Silicate |
| Intracellular concentrations of nitrogen, carbon, chlorophyll, and calcium in small phytoplankton (PhyN, PhyC, PhyChl, PhyCalc) |
| Intracellular concentrations of nitrogen, carbon, chlorophyll, and silicate in diatoms (DiaN, DiaC, DiaChl, DiaSi) |
| Intracellular concentrations of nitrogen and carbon in each of two zooplankton groups (HetN, HetC, Zoo2N, Zoo2C) |
| Two size classes of detritus for nitrogen, carbon, silicate, and calcium (DetN, DetC, DetSi, DetCalc; and DetZ2N, DetZ2C, DetZ2Si, DetZ2Calc) |

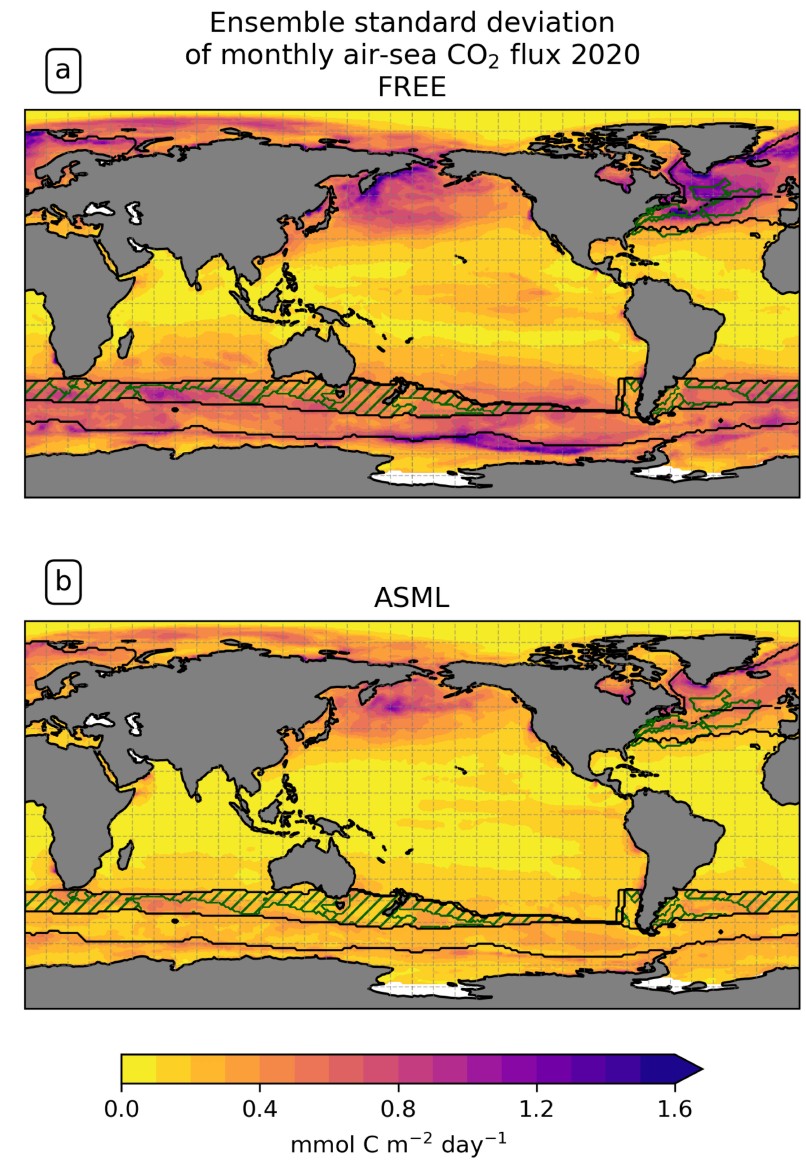

**Figure A1.** Ensemble standard deviation of monthly air-sea $CO_2$ flux in the year 2020 in (a) FREE and (b) ASML.

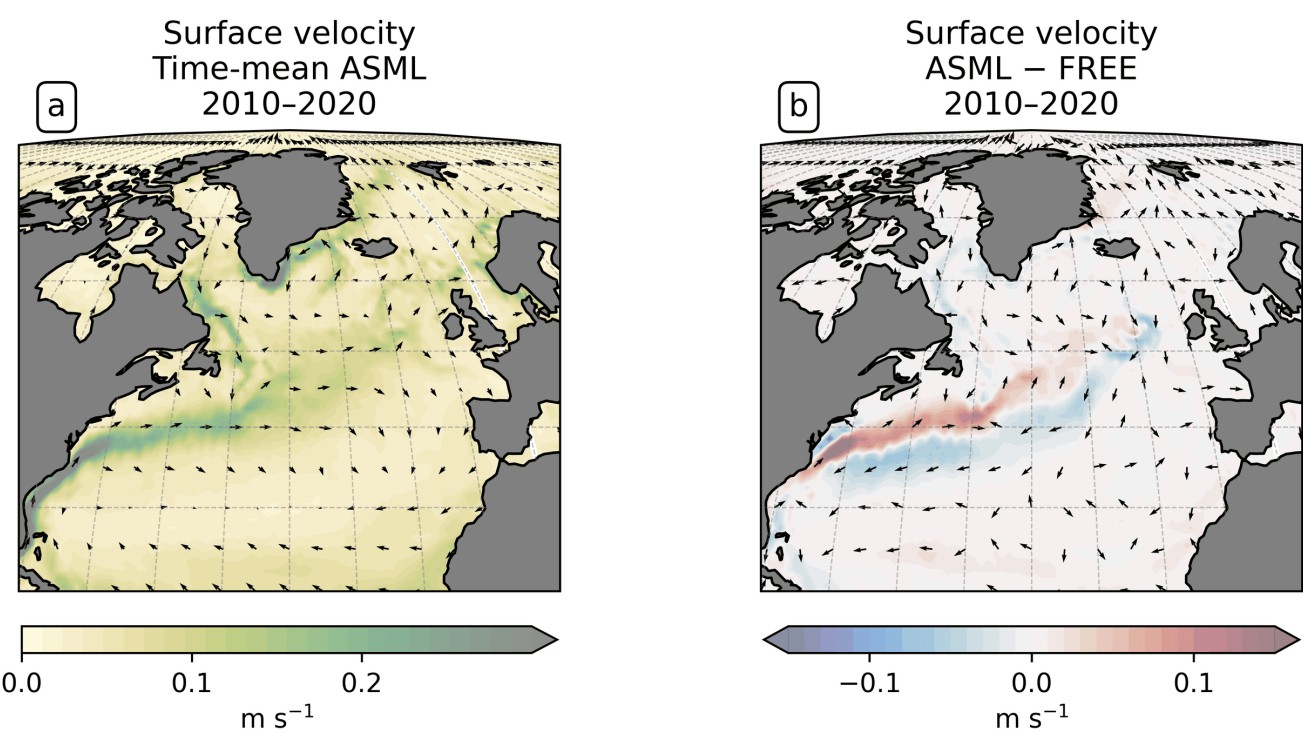

**Figure A2.** North Atlantic surface velocities, (a) time-mean in ASML and (b) difference ASML − FREE.

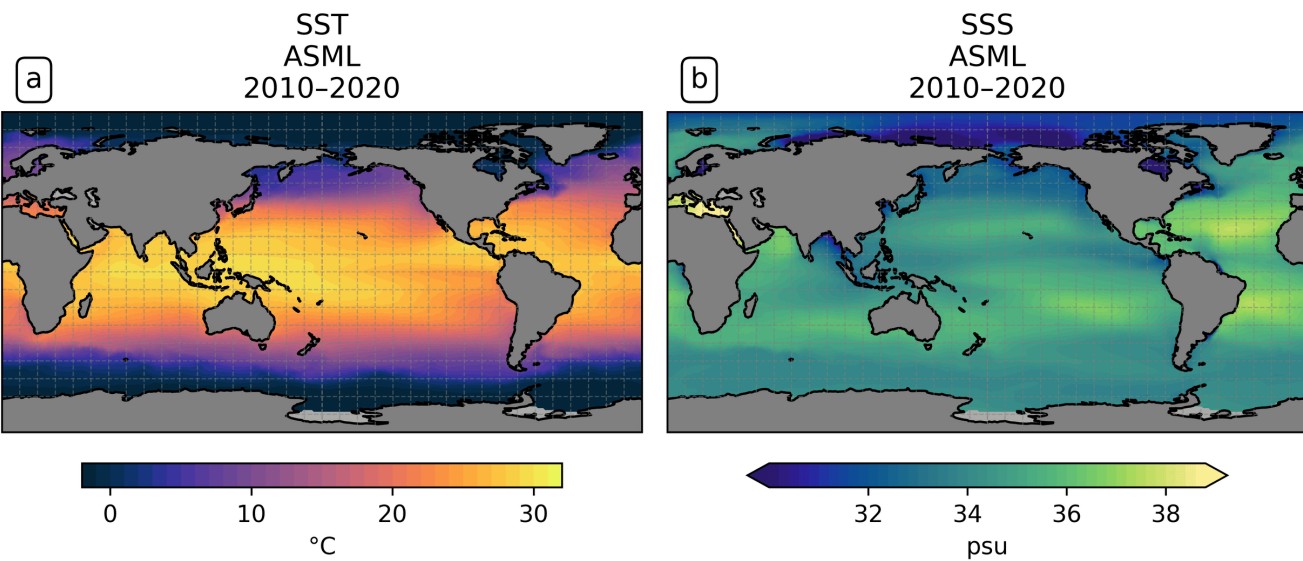

**Figure A3.** Time-mean sea surface (a) temperature and (b) salinity in ASML.

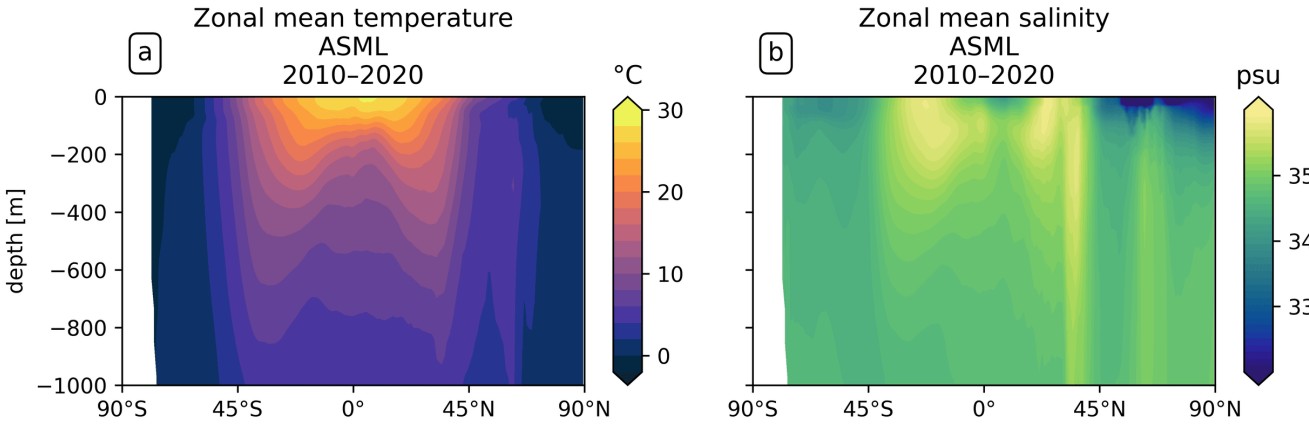

**Figure A4.** Zonally averaged time-mean (a) temperature and (b) salinity in ASML.

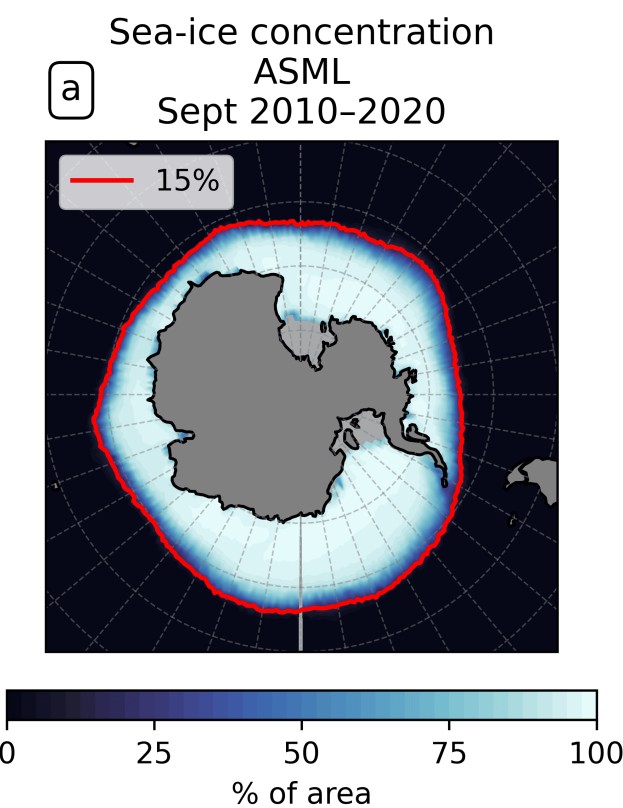

**Figure A5.** September mean Antarctic sea-ice concentration in ASML.

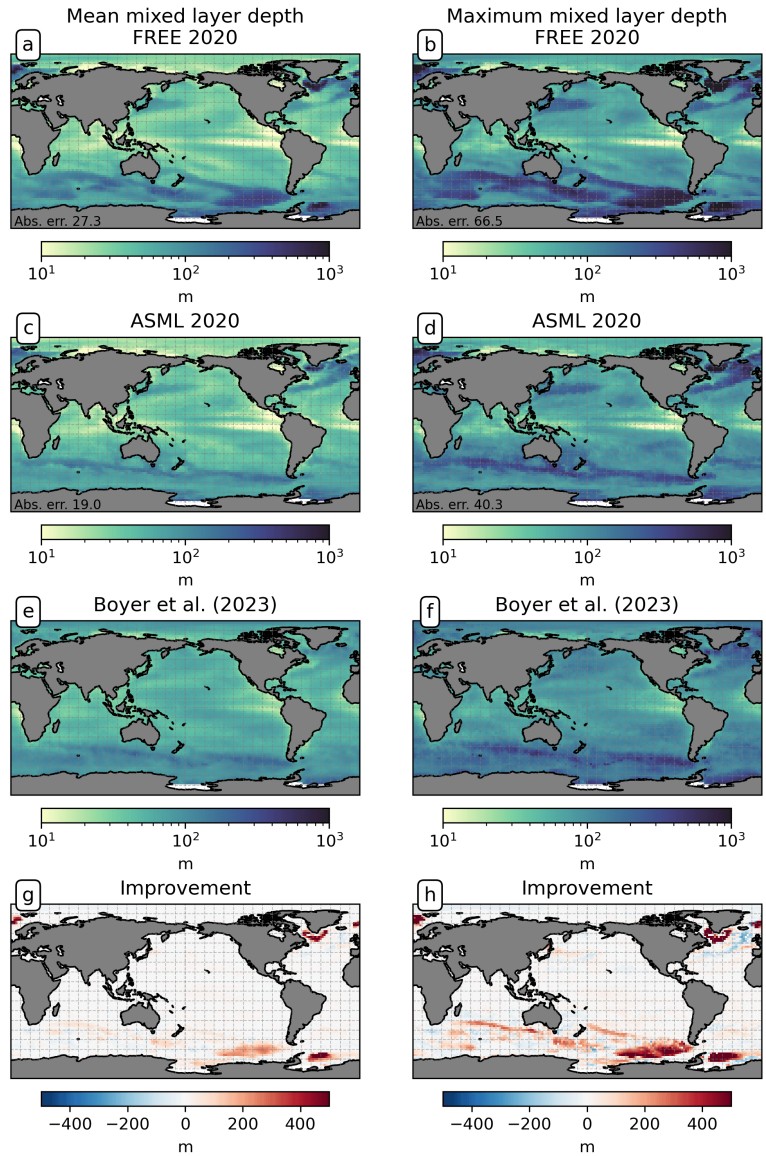

**Figure A6.** Mixed-layer depth (a,b) in FREE and (c,d) ASML in the year 2020, (e,f) de Boyer Montégut et al.'s (2004) profile-based climatology v2023 and (g,h) the improvement through DA relative to the climatology. On the left: time-mean mixed layer, on the right: maximum of monthly-mean mixed layer. For FREE and ASML (a,b,c,d), the mean absolute difference to the climatology is given in the bottom-left corner.

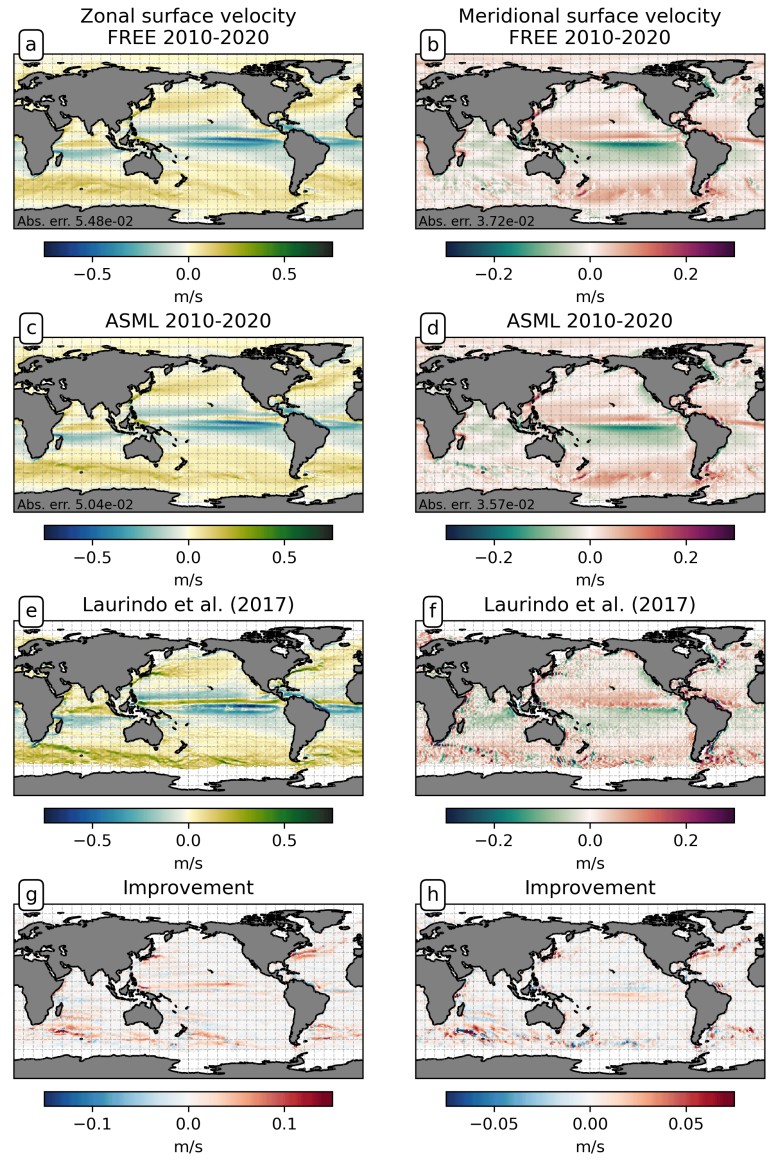

**Figure A7.** Near-surface velocities (a,b) in FREE and (c,d) ASML for the period 2010-2020, (e,f) Laurindo et al.'s (2017) climatology from drifter observations and (g,h) the improvement through DA relative to the climatology. On the left: zonal velocities, on the right: meridional velocities. For FREE and ASML (a,b,c,d), the mean absolute difference to the climatology is given in the bottom-left corner.

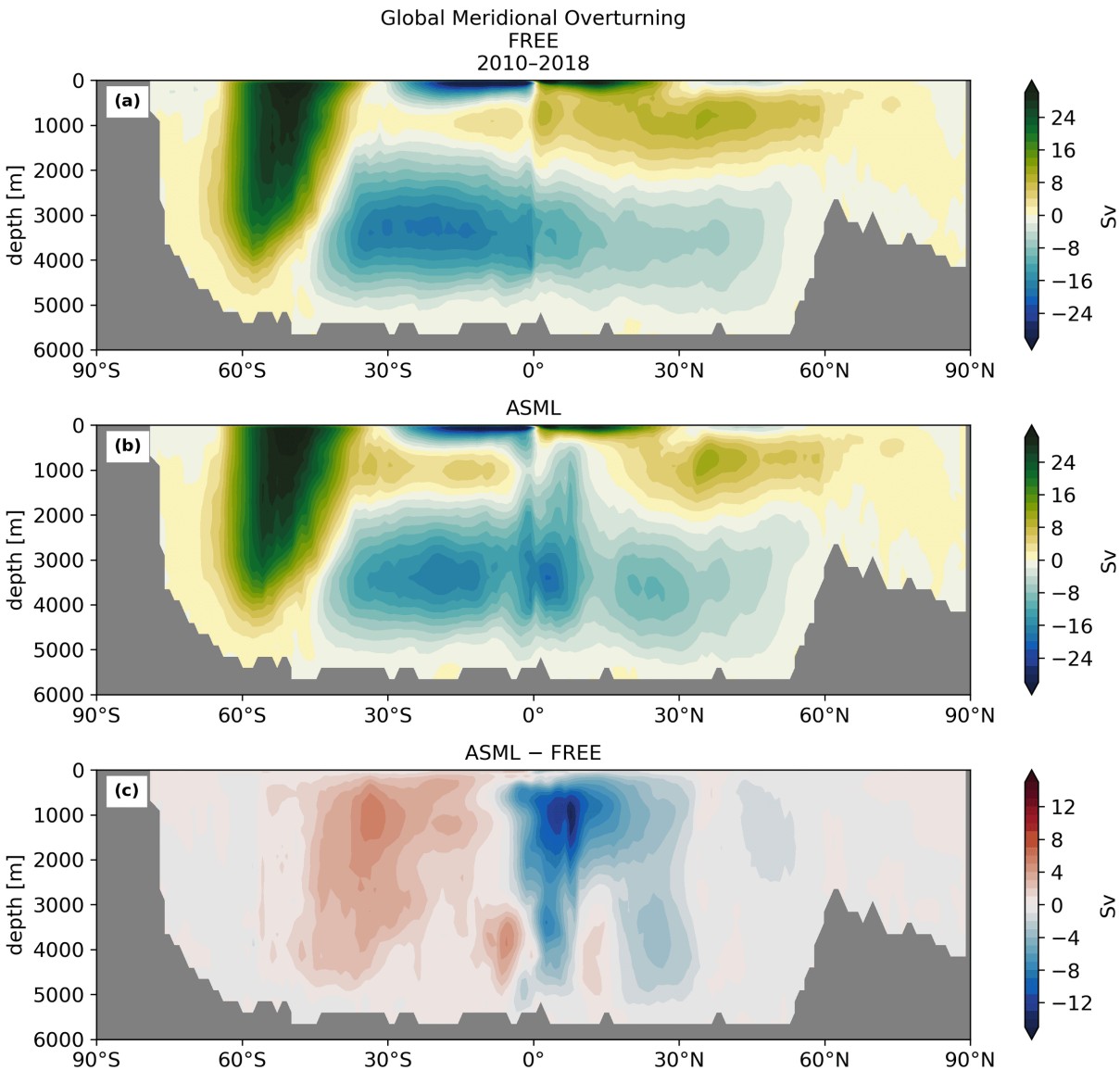

**Figure A8.** Global meridional overturning in (a) FREE, (b) ASML and (c) difference ASML-FREE.

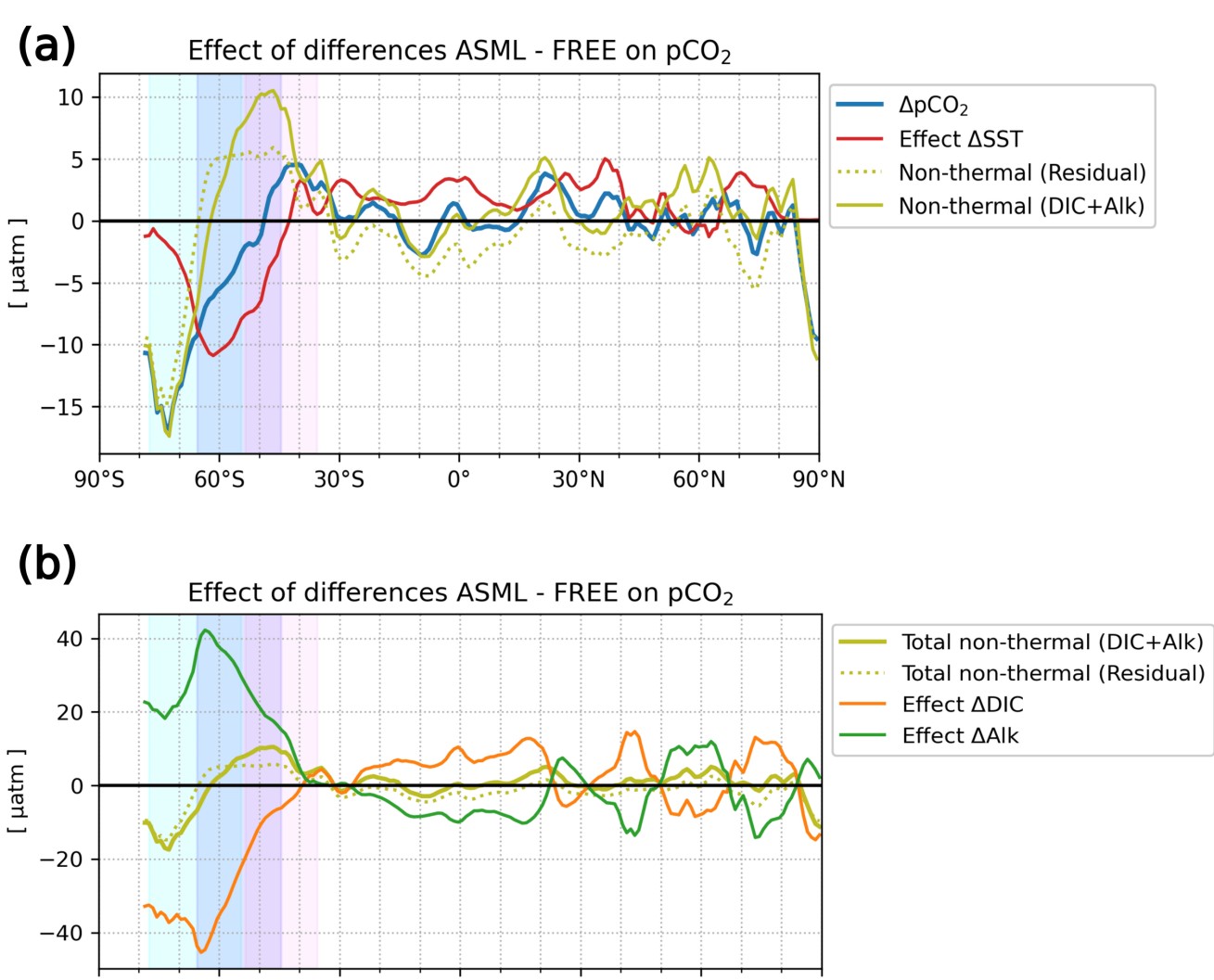

**Figure A9.** The net difference $\mathrm{ASML} - \mathrm{FREE}$ of surface $pCO_2$ by latitude (panel a, blue line), and the offline-approximated effects causing that $pCO_2$ difference for the period 2010-2020: Thermal effect (panel a, red line); non-thermal effect calculated, firstly as the residual i.e. net-minus-thermal (panels a and b, light-green dotted lines), and secondly as the sum of alkalinity and DIC effects (panels a and b, light-green solid lines); and effects of alkalinity and DIC individually (panel b, orange and dark-green lines). The shaded areas in the background indicate the zonal extent of defined biomes in the Southern Ocean: $ICE_{SO}$ in light-blue, $SPSS_{SO}$ in blue and $STSS_{SO}$ in pink. Colors blend where the regions overlap.

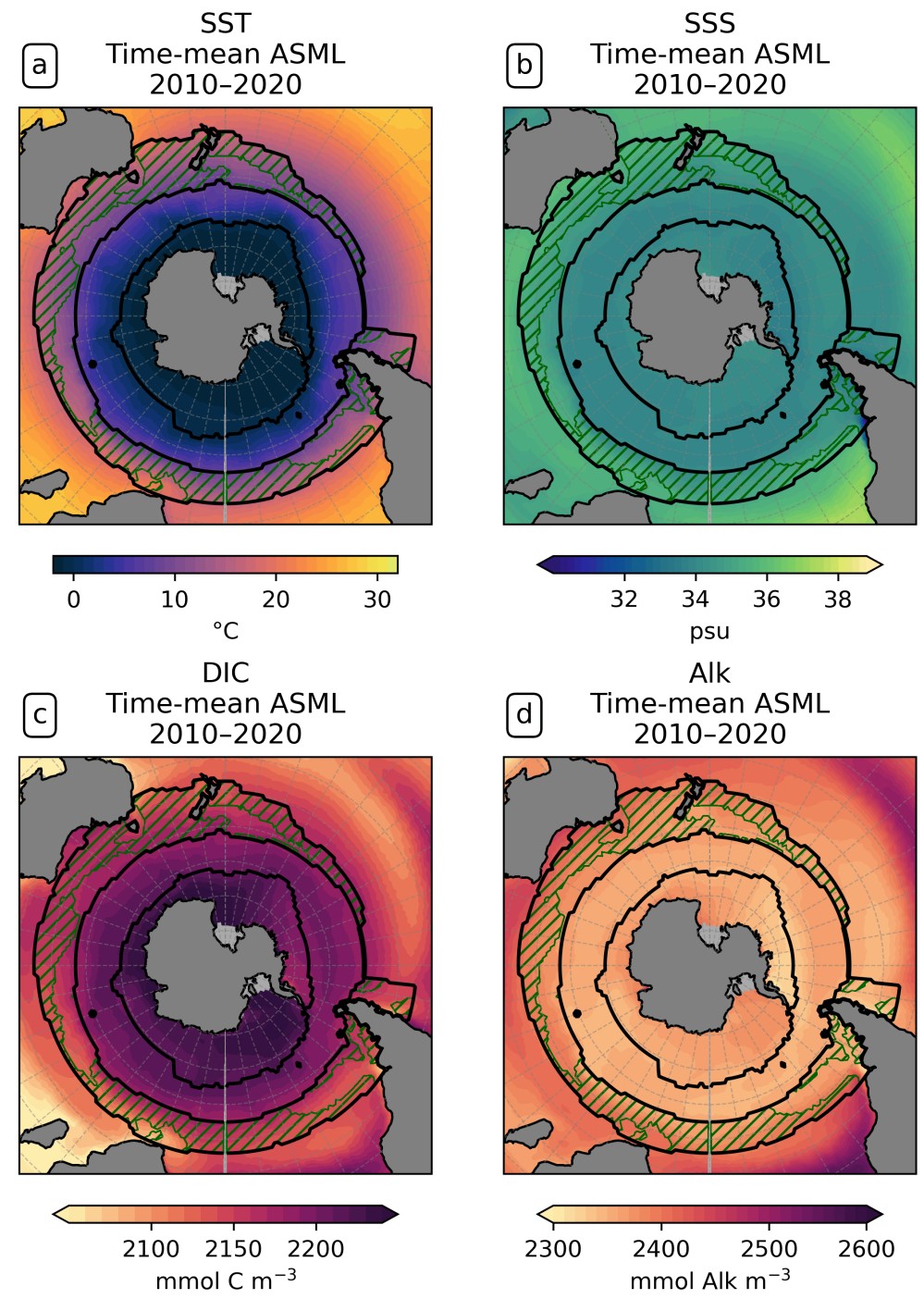

**Figure A10.** Southern Ocean time-mean sea surface (a) temperature, (b) salinity, (c) DIC and (d) alkalinity in ASML.

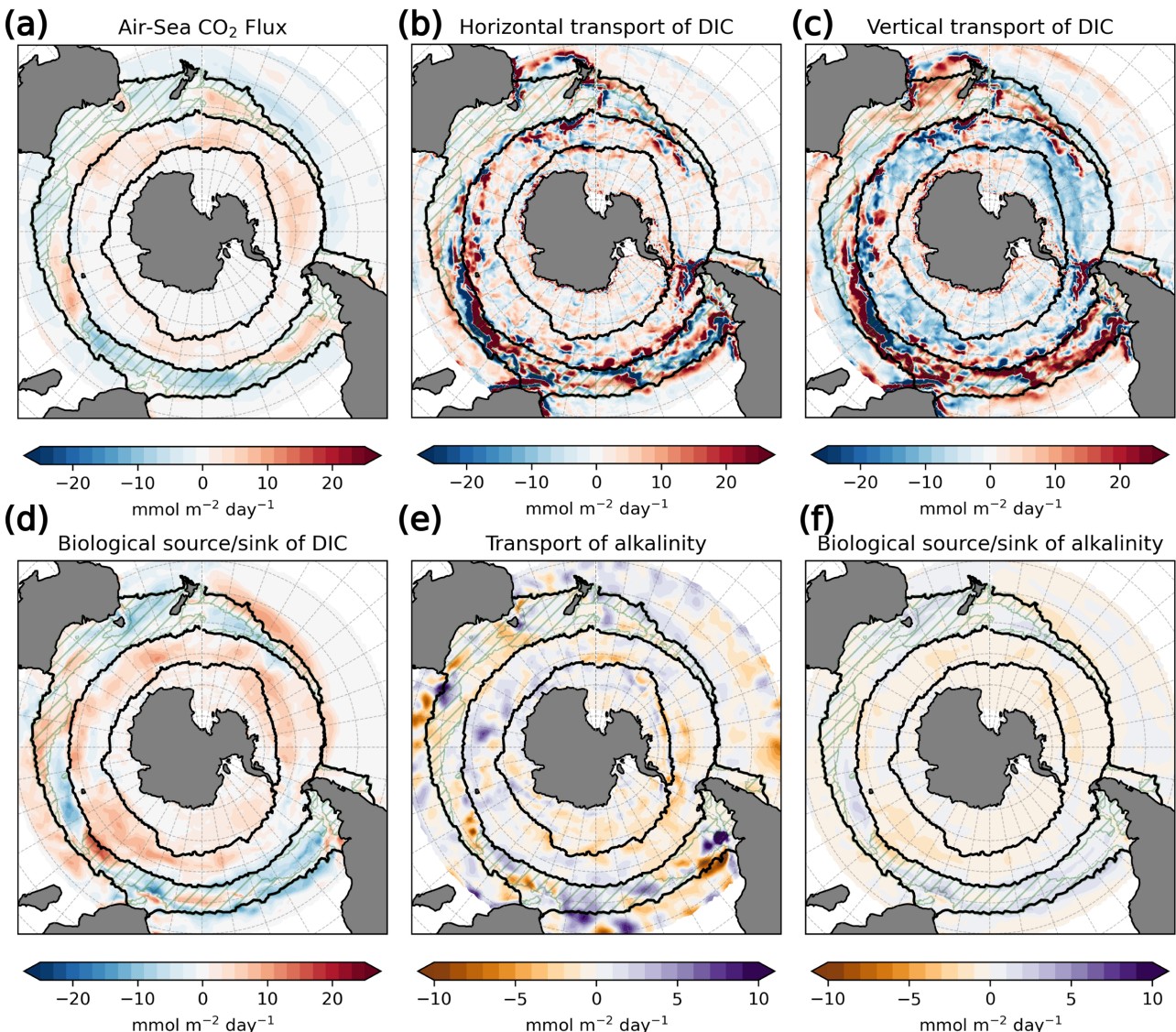

**Figure A11.** The difference $\mathrm{ASML} - \mathrm{FREE}$ of source and sink terms for the ocean's DIC and alkalinity content integrated over 0-190 m in the Southern Ocean in the year 2020. Transport terms include advection and diffusion of DIC and alkalinity. Biological terms for DIC are the sum of: photosynthesis, respiration, remineralization of dissolved organic carbon, and formation and dissolution of calcite. Biological terms for alkalinity are the sum of: nitrogen assimilation and remineralization, and formation and dissolution of calcite.

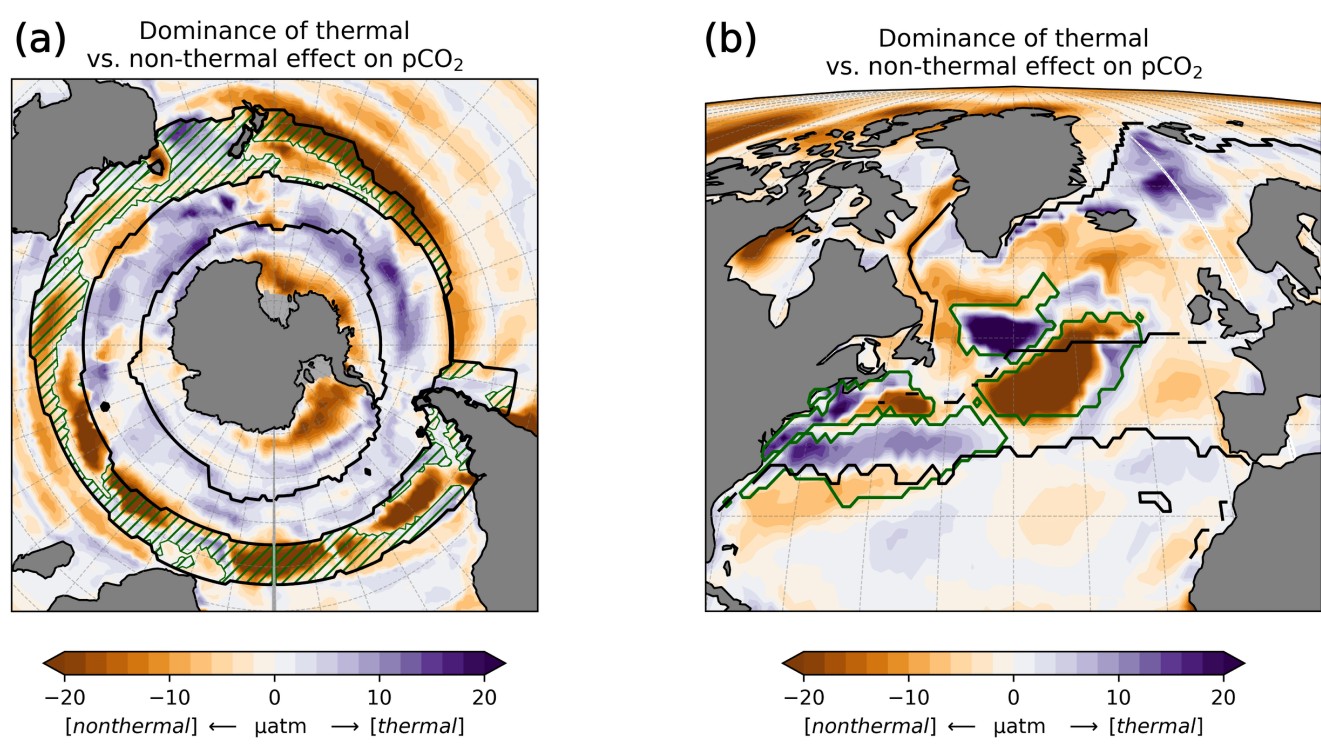

**Figure A12.** Linear offline estimate of the dominance of thermal versus the non-thermal effect through the assimilation on pCO$_2$ in the Southern Ocean and North Atlantic for the period 2010-2020.

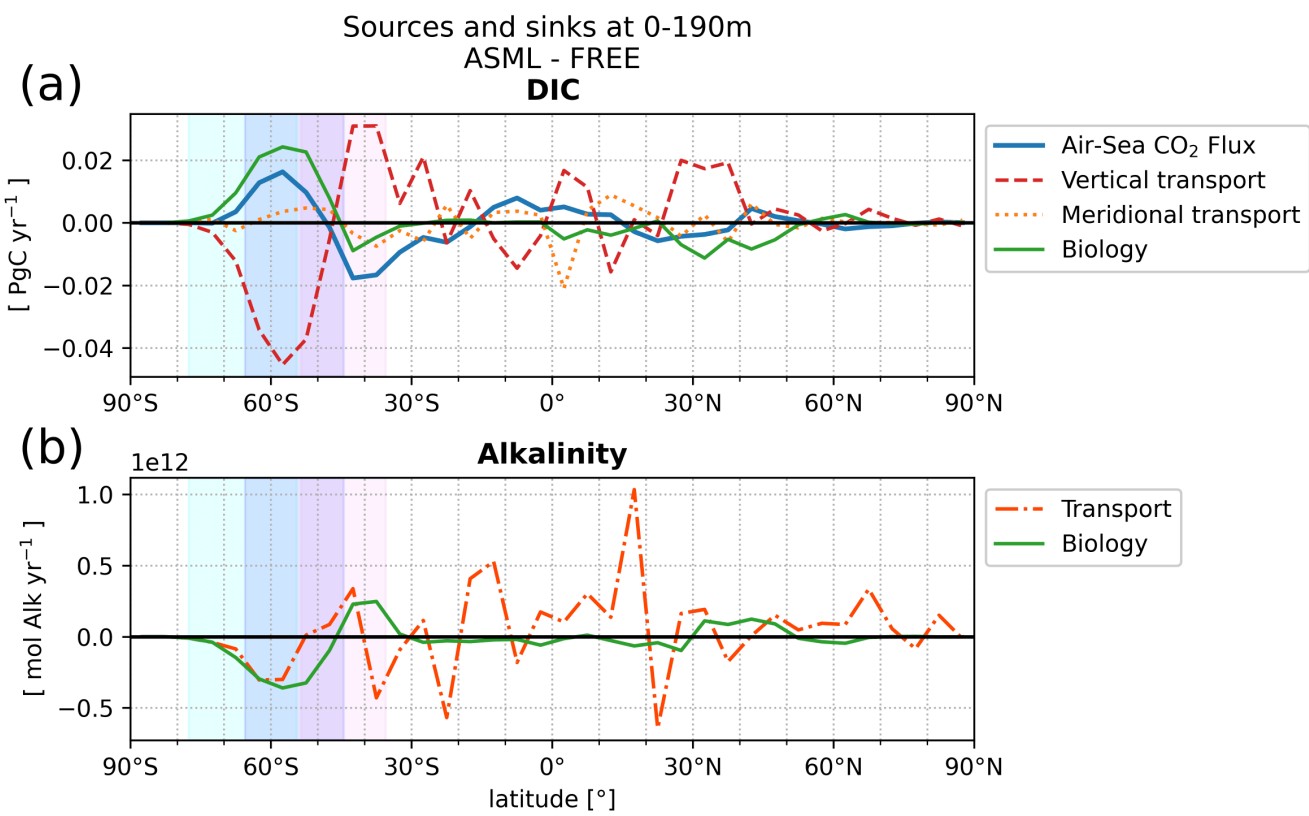

**Figure A13.** The difference ASML − FREE of source and sink terms for the ocean's (a) DIC and (b) alkalinity content integrated over 0-190 m per 1° latitude in the year 2020. Transport terms include advection and diffusion of DIC and alkalinity. Meridional transport is averaged across bins of 5° latitude. In panel b, vertical and horizontal transport are summed up for readability. Biological terms for DIC are the sum of: photosynthesis, respiration, remineralization of dissolved organic carbon, and formation and dissolution of calcite. Biological terms for alkalinity are the sum of: nitrogen assimilation and remineralization, and formation and dissolution of calcite.

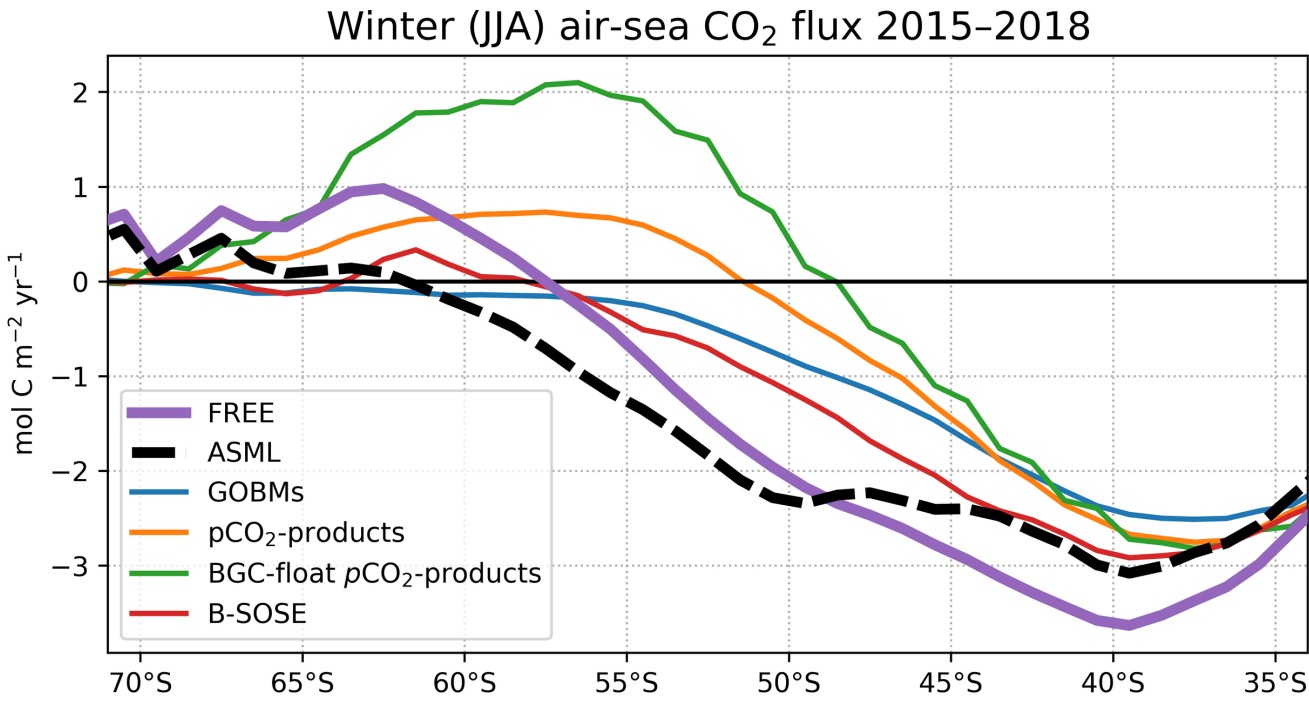

**Figure A14.** Zonally averaged winter (JJA) air-sea CO$_2$ flux (negative: into the ocean) in FREE, ASML and previous estimates (Hauck et al., 2023a; Verdy and Mazloff, 2017).

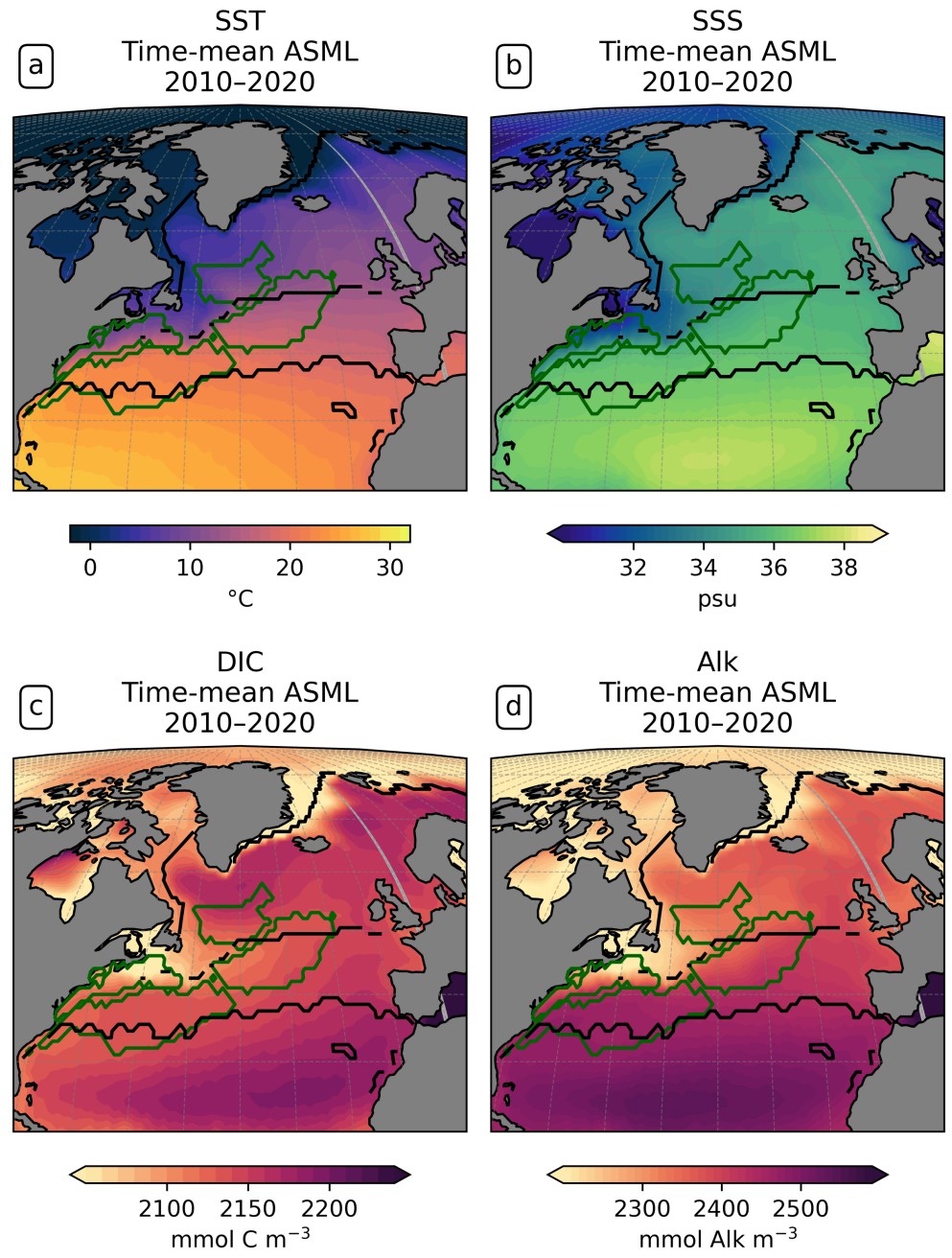

**Figure A15.** North Atlantic time-mean sea surface (a) temperature, (b) salinity, (c) DIC and (d) alkalinity in ASML.

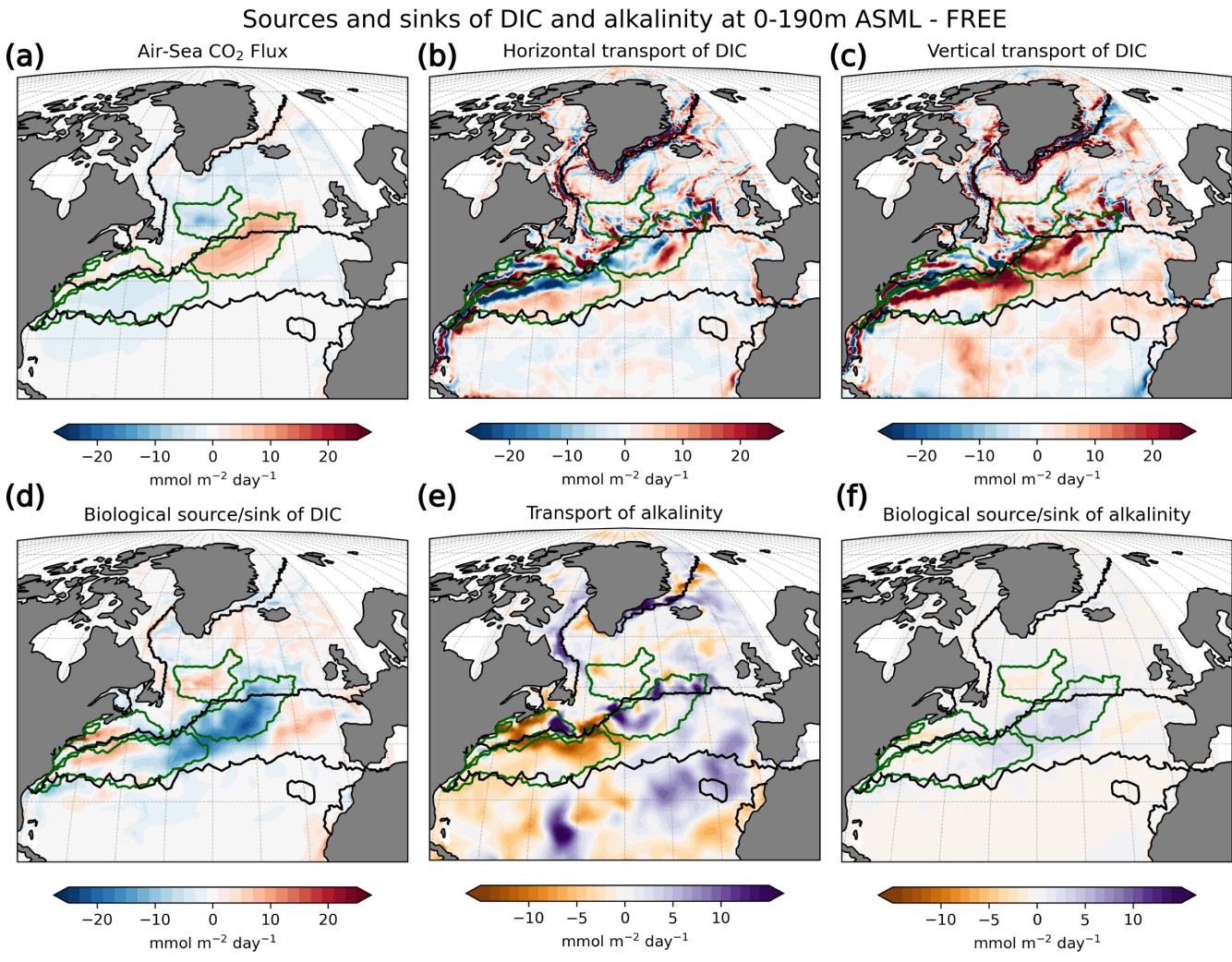

**Figure A16.** The difference $\mathrm{ASML - FREE}$ of source and sink terms for the ocean's DIC and alkalinity content integrated over 0-190 m in the North Atlantic in the year 2020. Transport terms include advection and diffusion of DIC and alkalinity. Biological terms for DIC are the sum of: photosynthesis, respiration, remineralization of dissolved organic carbon, and formation and dissolution of calcite. Biological terms for alkalinity are the sum of: nitrogen assimilation and remineralization, and formation and dissolution of calcite.

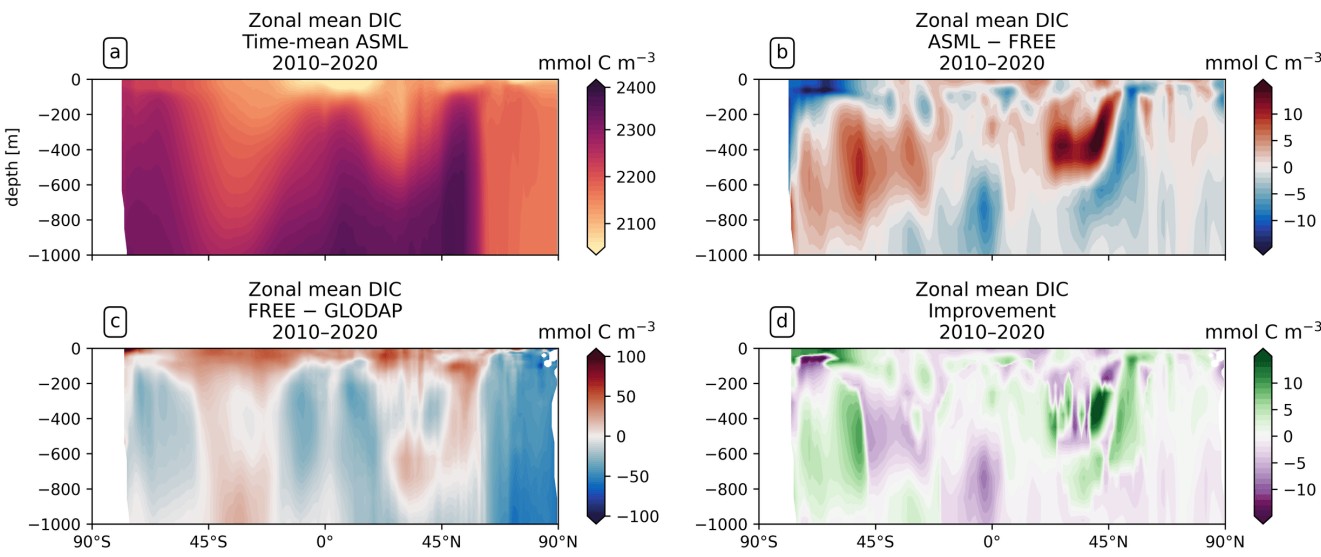

**Figure A17.** Zonally averaged DIC: (a) time-mean in ASML, (b) difference ASML − FREE, (c) difference FREE − OBS based on the GLODAP climatology (Lauvset et al., 2016) and (d) improvement respective to GLODAP.

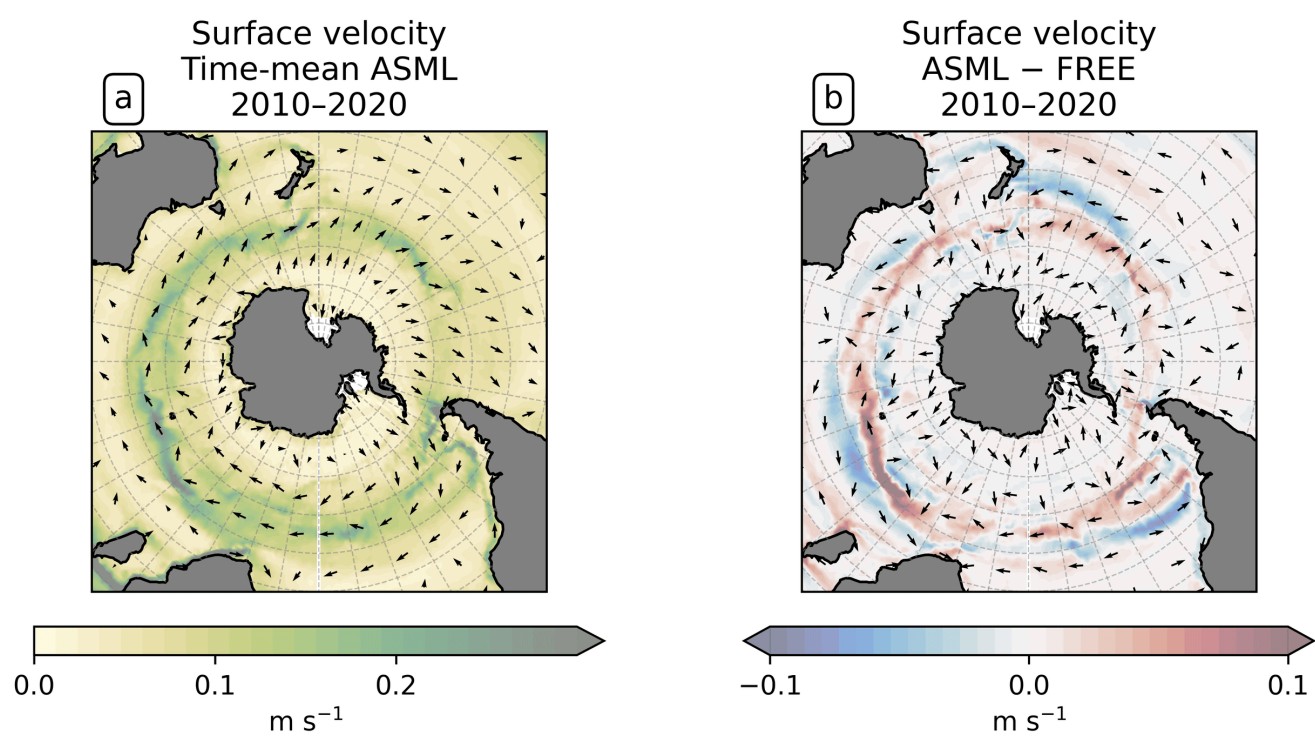

**Figure A18.** Surface velocities in the Southern Ocean, (a) time-mean in ASML and (b) difference ASML − FREE.

*Author contributions.* JH and LN conceptualized the research idea and provided supervision of the work. FB worked on the code for the model binding, for which LN provided supervision, and performed formal analysis of the data and figure production. FB prepared the initial paper draft with conceptional inputs from all authors. All authors contributed to the review and editing of the final manuscript.

*Competing interests.* The authors declare that they have no conflict of interest.

*Acknowledgements.* We acknowledge the Global Carbon Project, which is responsible for the Global Carbon Budget and RECCAP2 and we thank the ocean modeling and $fCO_2$-mapping groups for producing and making available their model and $fCO_2$-product output; in particular Cara Nissen for providing the files. Further, we thank Longjiang Mu who provided code for a PDAF-model binding within the FESOM model family in order to modify it for our study. At last, we acknowledge the use of DeepL Free (DeepL SE, https://www.deepl.com/translator) for translations and of ChatGPT 3.5 (Open AI, https://chat.openai.com) to provide rewording suggestions for the text.

FB has received funding from the AWI INSPIRES programme, and JH from the Helmholtz Young Investigator Group Marine Carbon and Ecosystem Feedbacks in the Earth System (MarESys, Grant VHNG-1301), and from the European Research Council Starting Grant ERC2022-STG OceanPeak (Grant 101077209). The work reflects only the authors' view; the European Commission and their executive agency are not responsible for any use that may be made of the information the work contains.

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
