# Peer review of "Ocean carbon sink assessment via temperature and salinity data assimilation into a global ocean biogeochemistry model"

_EGUsphere, 2024_

## Author Comment (AC1)

The manuscript presents an application of an ensemble-based physical data assimilation technique to a global biogeochemical ocean model, with a focus on the effect of physical data assimilation on climate-relevant carbon estimates. The manuscript is mostly well written and offers some valuable insights on the effects of physical DA, but the text could be improved in places and several aspects of the DA experiments should be examined further.

**general comments**

One aspect that is becoming more important in modeling studies but is seemingly ignored in the current version of the manuscript is the reporting of model uncertainty -- even though ensembles are used to generate the results. The authors mention ranges of estimates when reporting results from other studies. However, in their own analysis, the focus is solely on the ensemble mean, without examining the full model ensemble or reporting any uncertainty estimates. It would be beneficial to explore ensemble-based ranges of estimates and compare them to the improvements brought about by data assimilation. This could lead to interesting questions, such as the extent to which data assimilation constrains estimates and whether the estimates improve in areas where they are more constrained. Additionally, figures like Fig. 4 and the seasonal difference plots could be enhanced by including uncertainty estimates, such as the ensemble standard deviation or the interquartile range.

Thank you for the suggestion. Indeed, a reduction in the uncertainty of the $CO_2$ flux estimate would be a very relevant result in addition to an improved estimate of the mean $CO_2$ flux.

There is, however, one difficulty in the interpretation of the ensemble standard deviation in our method. Because we use a Kalman filter variant, the ensemble standard deviation (STD) of the DA-updated variables (T, S, SSH, u, v) is reduced in ASML. Most of the reduction in ensemble spread occurs over the course of the first year. After that, the STD remains stable, precisely because we tune our ensemble perturbation and ensemble inflation in such a way that the STD of temperature is maintained after the initial phase (Figure R1; yellow and green lines).

It is thus expected that the ensemble standard deviation of $CO_2$ flux decreases as well in ASML, but this is a result of the model and not part of the tuning. Indeed, we find that the STD for the local $CO_2$ fluxes in ASML is reduced to about 75-80% of the STD in FREE after the first year of assimilation (see example in Figure R2; however, this data is not area-weighted). We will add analysis and discussion of the uncertainty estimates in the revised manuscript in a computationally efficient way (rerunning for additional output is needed and may be done for one or more years).

[Figure]

Figure R1: Ensemble standard deviation for 3D-temperature. Note: No volume-weighting applied for the global mean (includes empty cells).

[Figure]

Figure R2: Ensemble standard deviation for the CO2 flux. Note: No area-weighting applied for the global mean.

Additional output with ensemble statistics for the year 2020 suggests that there are again regional differences. For example, in the Newfoundland Basin, which showed a strong effect of DA on $CO_2$ fluxes, the standard deviation was reduced strongly by assimilation, but less in the other regions in the North Atlantic (Figure R3). Discussion of these effects will be added to the revised manuscript.

[Figure]

Figure R3: Ensemble standard deviation of $CO_2$ flux and its reduction for the year 2020 (last year of the simulation).

The manuscript emphasizes carbon storage through physical transport, i.e. "upwelling and subduction of DIC, as well as the physical transport of other biogeochemical tracers" (l 60). However, the role of biological carbon fixation and sinking of particulate organic matter seems underexplored. Given that the model includes both slow and fast sinking detritus variables, a more comprehensive examination of these processes would be valuable. Here, it would help to clarify whether the biological carbon export at 200m (l 379 and following) is primarily due to sinking or physical transport. A closer examination or clearer description of the effects of the DA on the biological drivers of carbon export would help to improve the manuscript.

We would like to note that we're most interested in anthropogenic $CO_2$ uptake, which is primarily physically driven (e.g., Gruber et al., 2023 https://doi.org/10.1038/s43017-022-00381-x). A much closer examination of the biological carbon pump would be interesting, but is beyond the scope of this paper. Yet, on a regional scale, changes in biological export production contribute to the overall carbon balance and thus may have noticeable effects on the regional net $CO_2$ fluxes.

In response to the reviewer's comment, in the revised manuscript, we will address explicitly where the assimilation produces a change in export production. We will provide supplementary maps of carbon export through sinking of detritus at 190m. For example, we find that in the North Atlantic Central STSS, the increase in export production (by up to 4 mmol C m$^{-2}$ day$^{-1}$),

presumably in response to mixed layer deepening and/or increase of SST, is essential to explain the overall effect in the direction of more $CO_2$ uptake (Figure R4).

[Figure]

Figure R4: Effect of assimilation on carbon export through sinking of detritus at 190m.

In contrast, in the Southern Ocean, assimilation-induced changes in the air-sea $CO_2$ flux driven by physics (the effect of SST on $pCO_2$ and transport of DIC and alkalinity) are about twice as large as the response of the biological pump to the assimilation. In particular south of 50°S, the response of the biological pump is more than compensated for. The export of carbon through sinking of detritus decreases presumably because of shallower mixing. However, this is outweighed by a decrease of upward DIC transport and thus more ocean $CO_2$ uptake (Figure R5).

[Figure]

Figure R5: Assimilation-induced changes in the air-sea $CO_2$ flux and in carbon export through sinking of detritus by latitude. Negative denotes a more downward flux, i.e. $CO_2$ flux from air to sea and downward sinking of detritus.

We would also like to clarify that vertical export of organic carbon takes place almost entirely through sinking (the gravitational pump; Boyd et al., 2019: https://doi.org/10.1038/s41586-019-1098-2) rather than through physical transport. To clarify, we have changed these lines to:

(l. 379) "biological carbon export at 200~m through sinking of detritus".
(l. 60) "physical transport of DIC " (deleted)

This is because the carbon biomass itself is very low: In REcoM, the integrated global ocean living carbon biomass is around 2 PgC and the mass of dead organic carbon is of similar magnitude. However, the biological fluxes are very efficient so that globally, sinking of detritus removes about 10 PgC (i.e. a multiple of its own mass) per year from the upper 200m of the ocean. Due to the low concentration of carbon biomass, transport by advection and mixing of organic mass is less important.

In contrast, the concentration of DIC in the ocean is much higher. In REcoM, the ocean globally holds around 38000 PgC in the form of DIC. Therefore, the transport of DIC by advection and mixing plays a decisive role.

The assimilation of physical observations that only directly updates the physical variables can lead to "shocks" in the biogeochemical variables. It would be valuable to know if the authors observed any negative effects of daily physical updates on the biogeochemical state, such as unexpected phytoplankton blooms (for example, caused by a deepening of the mixed layer transporting nutrients, formerly below the mixed layer, to the surface).

We are not aware of any such shocks. This might relate to our overall finding that the modeled carbon fluxes and other inspected variables such as chlorophyll-a, NPP and plankton biomass act almost surprisingly indifferent to substantial differences in the model physics. The most rapid assimilation-induced changes take place in the first few months after the start of the assimilation, yet there was no noticeable shock.

Several aspects of the model setup and data assimilation process could benefit from further explanation or discussion. For instance, the restoration of surface salinity towards climatology may interfere with the assimilation of salinity data. It would be informative to know if the authors have experimented with switching off the nudging when or where salinity data is being assimilated, and how well the salinity climatology aligns with the assimilated data.

The main effects of SSS assimilation and salinity restoring are to reduce SSS globally. In addition, there are certain regions of model bias, such as the Amazonas river inflow area and the North Atlantic Current, where both methods are consistent with each other. While there are gaps in the SSS-CCI data near the poles, the salinity restoring towards climatology is with global coverage. Experiments with and without salinity restoring show that without restoring, sea surface salinity in FREE drifts by approximately +0.05 psu during the first year after switching it off. In ASML, the difference between switching salinity restoring off or on is smaller (less than 0.01 psu globally), because the assimilation compensates for the lack of restoring. In ASML, global SSS is reduced by approximately 0.15 or 0.2 psu, respectively, after one year, which shows that the assimilation has a stronger effect than the restoring. The best agreement with SSS-CCI observations is achieved when assimilation and salinity restoring are used simultaneously.

In summary, we added to the manuscript: "Additional experiments with and without salinity restoring towards climatology show that the best agreement with the SSS-CCI observations is achieved by simultaneously using assimilation and restoring. Hereby, a benefit of additional restoring is the global coverage of the SSS climatology."

Similarly, the exclusion of temperature observations from the DA when the model-observation difference exceeds 2.4°C could use a better explanation, as this seems to hinder assimilation where it might be most needed.

By excluding these observations, the aim is to prevent strong and sudden corrections from making the model unstable, especially in the initial phase. Instead, a 'gentler' correction is made by assimilating neighboring points. Because we use a gap-filled SST observational product, observations are continuously available in the neighboring domains. After the initial phase, about 7% of SST observations are excluded because of the temperature-threshold regularly. However, the data assimilation still has a strong effect in areas where these large model-observation discrepancies are found (North Atlantic Current, near Japan and in some places of the Southern Ocean).

We added this information to the manuscript.

To improve readability, particularly for readers less familiar with data assimilation techniques and carbon modeling, brief explanations of key concepts and modeling choices would be beneficial. These would include descriptions of the term used to perturb atmospheric forcing, the role of ensemble inflation, and the rationale behind the choice of γ_DIC and γ_Alk in Equations 4 and 5 (see also my specific comments below). Currently, the manuscript often uses references to other studies to motivate implementation details, and an additional sentence here and there could help the reader to better understand these details without having to go through other papers.

We agree and have made text additions in the places that you mentioned.

In places, the structure of the manuscript can be improved to enhance clarity and flow. Sections 4.2 and 4.3 are quite lengthy and could be subdivided based on location (Southern Ocean, Atlantic) and the different data products used in the comparisons. Section 3, which contains results from the two ensemble simulations, could be merged with Section 4 to create a more cohesive results section.

Thank you, we rearranged the sections and section titles accordingly.

Overall the figures look very good and are helpful, I only have a minor suggestion here: it might be more informative to report ASML-OBS instead of ASML-FREE in Figures 1-3. This would provide a clearer picture of the model error following data assimilation. Also, some of the figures, such as Figure 7, have lots of whitespace that could be reduced.

We have chosen ASML - FREE throughout the manuscript because it allows us to visualize comparatively small changes in some of the biogeochemical variables. On the one hand, for temperature and salinity, ASML-OBS provides a clear picture of the model error after data assimilation (see SST, Figure R6). On the other hand, for the biogeochemical variables, FREE-OBS and ASML-OBS are visually too similar to recognize the differences (see chlorophyll, Figure R7). Showing ASML-FREE for all variables allows one to recognize correlations between the effects of DA on different variables.

[Figure]

Figure R6: FREE-OBS and AMSL-OBS for SST, useful to illustrate the model error before and after assimilation.

[Figure]

Figure R7: FREE-OBS and AMSL-OBS for chlorophyll, not useful because the effect of the assimilation is almost unrecognizable.

**specific comments**

L 8: "the mean CO2 uptake increases by 0.18 Pg C yr−1": Add "regionally" here to make it explicit that this increase is not a resulting global estimate.

Thank you. Done.

L 40: "the model mean": It would be helpful to the reader to add a few words about the kind of models that were considered here.

On this, added: "the mean of GOBMs included in the Global Carbon Project"

L 65: "DIC" was used before the abbreviation is introduced here (l 59). The earlier sentence actually makes a quite similar point about subduction of DIC and also mentions upwelling, perhaps this could be made more concise.

Thanks, we merged both sentences into one.

L 65: "It was shown that assimilating ocean physics at the initial state of a model simulation has a stronger and more positive impact on the modeled carbon cycle than assimilating the BGC initial state": Is this due to the lack of BGC observations mentioned earlier, the importance of physical processes for carbon export, or a large physical model error that cannot be decreased through BGC DA?

This study (Fransner et al., 2020) relates the strong and positive effect of assimilating ocean physics to the strong control ocean physics exerts on the biogeochemical variability on interannual to decadal time scales (rather than low availability of BGC observations or strong physical model errors).

The next sentence brings up the question of which processes are most important. Maybe a few candidates could be named and briefly discussed here before going into the details of the DA algorithm.

We name some candidates in the revised text:

"This raises the question which mechanisms produce the response of the \ce{CO2} flux in physics DA approaches. Is it the transport of DIC and alkalinity through physical advection, mixing and in particular upwelling of carbon-rich waters, as the model velocities and diffusities are changed by the assimilation? How much is \ce{pCO2} changed directly through its temperature-dependence? Does the biological pump respond to the assimilation of physics? How large are these effects, and when and where do they occur?"

L 70: "continuously assimilating ocean-physics for eleven years": A bit more detail could be useful here as well: What does assimilating ocean physics entail, what observations are being used for the DA here?

Thank you, added: "We continuously assimilate temperature and salinity observations for eleven years and update the modeled temperature, salinity, horizontal velocities and sea surface height."

Detailed info can be found in the methods section.

L 89: "The model allows for a variable mesh resolution": What is a typical coarse and fine resolution used in the model grid?

Please see section "Simulation set-up", which we have now moved here:
 "The mesh resolution is nominally 1 degree, ranging between 120km and 20km with enhanced resolution in the equatorial belt and north of $\SI{50}{\degree N}$  (126858 surface nodes)."

L 93: A salinity flux of 0.1m/day? Please describe this better.

Thanks for asking. This number was a typo. We corrected the number and added description:

"The surface salinity is restored towards the World Ocean Atlas climatology through a fictional surface flux with a velocity of 50m /300 days according to the equation:

$(S_{\mathrm{clim}}-S_{\mathrm{model}}) / \mathrm{layer\_width} * \mathrm{velocity}$"

For the example of a salinity bias of 0.5 psu and with the surface layer width being 5m, this would yield a correction of 0.016 psu per day.

L 96: "DIC" is introduced again, a quick search shows 7 introductions of "DIC", also counting captions.

Thanks, we only kept it in the Introduction and Conclusion.

L 117: "observations are weighted by distance": This is not a precise statement that could confuse some readers, express more clearly that the ensemble estimated correlation between a model grid point and an observation is down-weighted using a distance-based metric. Is vertical localization applied as well?

Thank you for the more clear wording suggestion, we used it. And we added that there is no vertical localization.

L 124: It would be useful to add equation numbers to all equations, even those that are not referenced in the text, so that they can be more easily referenced in other texts, such as this one.

Done.

Eq L 124: Why does a larger ensemble amplify rand? It does not seem that intuitive to have larger perturbations in a larger ensemble.

The incomplete definition of 'rand' in the manuscript has led to an obvious misunderstanding: In fact, there are no larger perturbations in a larger ensemble. The factor (N_ens-1) compensates that the values of 'rand' become smaller with increasing ensemble size. The values for rand are generated by Second-Order Exact Sampling from a trajectory of atmospheric forcing fields, a method introduced by Pham et al., see e.g.:
https://doi.org/10.1175/1520-0493(2001)129<1194:SMFSDA>2.0.CO;2
and briefly explained here:
https://pdaf.awi.de/trac/wiki/EnsembleGeneration

To clarify, in the updated manuscript we add a few sentences on the generation of the initial perturbation and the stochastic element 'rand'.

L 153: "model values are computed as the average of the grid points of the triangle enclosing the observation because the number of observations is fewer than model grid points": Averaging is required to interpolate the model solution at the observation locations, why is this dependent on the number of observations?

Thanks for pointing out how this can lead to confusion. In fact, we simply meant: If observations of a variable are spatially highly resolved, they are interpolated to the model grid (as for SST and SSS). If observations of a variable are sparse, it is the other way round and the model solution is interpolated to the observation locations (as for the profile data).

As this was unnecessarily confusing, we have now removed the last part of the sentence.

L 157: This information about the model grid is missing from Section 2.1 where the model grid is described for the first time. It would also be useful to describe the atmospheric forcing before describing the perturbation to it (Section 2.2.1).

Thank you, we have moved the section.

L 171 "the river flux adjustment (...) is applied to the pCO2 products. ...": It is not entirely clear what this means, the focus here is just the CO2 flux associated with the oceans, I presume? The next sentence provides some more information but it seems to imply that the RECCAP2 CO2 flux is not being used for comparison, when previous sentences stated that it was. Some clearer language would be useful here.

In response to both reviewers pointing this out, we have rephrased:

*"We present \ce{CO2} flux estimates for the period 2010-2020, that are compared to the 'Regional Carbon Cycle Assessment and Processes 2' (RECCAP2) global air-sea \ce{CO2} flux estimates \citep{devries2023}. To make the RECCAP2 estimates comparable with our estimate stemming from a model without river carbon input, we apply a river flux adjustment \citep{friedlingstein2023,regnier2022} to the RECCAP \ce{pCO2} products. Thus, we quantify the anthropogenic perturbation of the ocean carbon sink \citep[as $\mathrm{S_{OCEAN}}$ in the Global Carbon Budget][]{friedlingstein2023,hauck2020}, and not the contemporary net air-sea \ce{CO2} flux with outgassing of river carbon into the atmosphere (as in RECCAP2)."*

L 183: Should the US East Coast be considered subpolar, are all regions characterized by seasonal stratification, or does SPSS stand for something different here? A alternative choice of region names may be suitable and would avoid confusion with the region names in the Southern Ocean.

In the revised text we point out that, according to the definition of Fay and McKinley, the STSS, SPSS and ICE biomes exist analogously in both hemispheres (https://doi.org/10.5194/essd-6-273-2014). Therefore, there is an SPSS biome in the North Atlantic, of which we discuss only specific parts (e.g. the East Coast SPSS region).

The Fay and McKinley biomes are used widely in the ocean carbon cycle community (see e.g. RECCAP papers, https://reccap2-ocean.github.io/publications/).

To avoid confusion with the regions names in the North Atlantic (NA) and Southern Ocean (SO), we will add subscripts the biomes, e.g. STSS$_{SO}$+ and East Coast SPSS$_{NA}$

L 185: Please explain "NAC".

North Atlantic Current, done.

Eq 1 and 2: Is there an easy to communicate motivation for the choice of γ_DIC and γ_Alk ?

In order to assess the dynamic DA effects on surface \ce{pCO2}, it is useful to distinguish between different variables that constitute the change in \ce{pCO2}. Oceanic \ce{pCO2} varies mainly with temperature, DIC and alkalinity. Thus, we decompose changes in \ce{pCO2} into their contributions from changes in SST (SST), surface DIC and alkalinity (Alk). For that, we apply the following approximations of \citet{sarmiento2006} and \citet{takahashi1993}:

[ equations ]

Here, differences between ASML and FREE are denoted by $\Delta$; else, the average of ASML and FREE is used for the computation. The sensitivities $\gamma_{\mathrm{DIC}}$ and $\gamma_{\mathrm{Alk}}$ describe how \ce{pCO2} varies with changes in one variable while keeping the other variables constant. For the sensitivities, we use an approximation derived from the solution chemistry of carbon dioxide in seawater following \citet{sarmiento2006}:

[ equations ]

Eq 1, 2 and 3: Previously Delta denoted the difference between ASML and FREE, is this still the case here? If so, are the regular terms (e.g. DIC in Eq 1 or the terms in γ_DIC) from the FREE experiment? This should be mentioned in the description.

Yes, delta is the difference between ASML and FREE and the regular terms are calculated from the average of the two - this has been added above.

L 220: Why not mention EN4-OA earlier when the other data products are introduced?

Okay, done.

L 250: "at greater depth than 500 m, where the model's subsurface temperature": The "subsurface" can be deleted here.

Okay, done.

L 266: Please explain what a 15%-line is.

"The maximum extent of sea-ice in September, here defined as the area where the sea-ice concentration is more than 15\%, is smaller in FREE than OSI-SAF, which is demonstrated by the 15\%-line surrounding that area for FREE and OSI-SAF (\cref{fig:sic}a);"

L 301: "In the more northern part of the STSS, which we call the STSS+, the CO2 uptake is reduced ...": The text here could be considered misleading because STSS+ is not defined as the northern part of the STSS, but as the part of the STSS with a positive CO2 flux difference. I would prefer a change in formulation that avoids this ambiguity, for example: "The part of the STSS characterized by a positive CO2 flux difference between ASML and FREE, which we call the STSS+ and in which the CO2 uptake is reduced, forms an outer (northern) ring around the STSS region." The same comment applies to STSS+ a few lines below.

Thank you for the suggested wording! We made use of it.

L 373: "the effect of the DA is towards increased uptake of CO2 during boreal summer and autumn in ASML (Fig. 6g). This prevents summer outgassing": The increased summer uptake prevents summer outgassing, isn't this just describing the same effect? I would suggest rewording this sentence.

"In the Central STSS, the effect of the DA is overall towards an increased uptake of \ce{CO2} from May to November (\cref{fig:CO2_NA}g). In particular, this prevents outgassing in high summer and even reverses the flux direction for some months, so that there is uptake in ASML almost all year round (\cref{fig:CO2_NA}c)."

L 411: "(difference of FREE and SOCAT in (Fig. 9a); difference of ASML and SOCAT not shown)": The figure label claims that ASML - SOCAT is shown.

Thank you for noting this. Figure data and labels have been updated to show FREE - SOCAT, as indicated in the text.

---

## Author Comment (AC2)

The manuscript describes a study assimilating temperature and salinity observations into a global physics-biogeochemistry ocean model, with the aim of improving the modelled air-sea $CO_2$ flux. The assimilation brought the model temperature and salinity closer to the assimilated observations, and had a mixed impact on the carbon variables and wider biogeochemistry. The global mean change was small, but could be regionally significant, with the mechanisms explored.

The experiments are well conceived, and the manuscript generally well written and well presented. I just have some comments where aspects could use clarifying or expanding on.

L51: "Data assimilation (DA) has been employed …" This paragraph doesn't need to be comprehensive, but could be modified and expanded a little to more fully represent the available literature. Valsala and Maksyutov (2010, https://doi.org/10.1111/j.1600-0889.2010.00495.x) ran a global assimilation for 1996-2004; not multidecadal but almost as long as the present study. The paragraph states "In each of these studies, an Adjoint or Green's Function DA approach is used", but the Gerber et al. (2009) study referenced used an EnKF – another non-adjoint/Green's function example is While et al. (2012, https://doi.org/10.1029/2010JC006815) who used a sequential analysis correction scheme to assimilate pCO2. The paragraph opens by talking about "DA studies of the air-sea CO2 flux" in general terms, only semi-clarifying later that it's focussing on studies which directly assimilated pCO2 data. There have also been other studies which, like the present one, looked at the impact of assimilating other variables on the air-sea CO2 flux, e.g. Ciavatta et al. (2016; https://doi.org/10.1002/2015JC011496) and other papers from that group, and Ford and Barciela (2017, https://doi.org/10.1016/j.rse.2017.03.040).

Thank you for the references to the literature, great! We acknowledge these.

L65: "It was shown that assimilating ocean physics at the initial state of a model simulation has a stronger and more positive impact on the modeled carbon cycle than assimilating the BGC initial state (Fransner et al., 2020)." In no way diminishing the motivation for this current study – which is undoubtedly important for the reasons stated in Fransner et al. (2020) and others – it could be clarified that this was a single model study and may or may not hold in general. The relative importance of physics vs biogeochemistry initialisation on different variables and time scales remains an open question – see e.g. the discussion in Section 4.4 of Lebehot et al. (2019, https://doi.org/10.1029/2019GB006186) and indeed the ultimate conclusions of this current manuscript.

Yes, this definitely needs to be put into context. Thank you again for the literature references.

L67: "Therefore the question arises which processes are most important when altered physics change CO2 fluxes in DA approaches." I think I understand the meaning of this sentence, but it could be reworded for clarity.

"This raises the question which mechanisms are responsible for the response of the \ce{CO2} flux in physics DA approaches."

L68: "to improve" – a better wording could be "to aim to improve"?

Yes, we changed this.

L75-79: The issues discussed by Park et al. (2018) and others, mentioned later in the manuscript, could be introduced at this point.

Okay, we moved this here.

L103: "Alkalinity is restored by a fictional surface flux of 10m/yr." Is there a reference for this, or was it introduced in this study?

We follow the set-up of Gurses et al. (2023). This alkalinity restoring has been used by Hauck et al. (2013) and Schourup-Kristensen (2014) as well.

Gurses: doi.org/10.5194/gmd-16-4883-2023
Hauck: doi.org/10.1002/2013GB004600
Schourup-Kristensen: doi.org/10.5194/gmd-7-2769-2014

L121: "After each assimilation step, corrections are applied to the analysis state to ensure the consistency of model physics." Can you give an indication of whether these corrections need to be applied regularly or just occasionally?

While the correction is necessary at each step for about 10\% of SSH updates and \SI{1e-3}{}\% of temperature values, the correction of salinity is never needed.

L148: How is the weekly-resolution SSS used in the daily assimilation?

To clarify, we have rephrased: "ESA-CCI contains daily data at a spatial resolution of 50~km, albeit not capturing temporal variability below weekly."

ESA-CCI: doi.org/10.1029/2021JC017676

We use the daily ESA-CCI data for the daily assimilation steps. It is not necessary for the observations to contain the day-to-day variability, as the data assimilation has a comparatively slow effect: For example, it takes several months of assimilation to achieve the maximum feasible correction of a large-scale model bias.

L153: "model values are computed as the average of the grid points of the triangle enclosing" – what's done in the vertical?

"To assimilate the profiles, the observations are assigned to the respective model layers (depth range) in the vertical."  - added to the manuscript.

L171: "For the comparison …" – this paragraph would benefit from a clearer explanation of what adjustments have been made to what products and why, including the model estimates from this study (which presumably have no river carbon inputs?).

In response to both reviewers pointing this out, we have rephrased:

"We present \ce{CO2} flux estimates for the period 2010-2020, that are compared to the 'Regional Carbon Cycle Assessment and Processes 2' (RECCAP2) global air-sea \ce{CO2} flux estimates \citep{devries2023}. To make the RECCAP2 estimates comparable with our estimate stemming from a model without river carbon input, we apply a river flux adjustment \citep{friedlingstein2023,regnier2022} to the RECCAP \ce{pCO2} products. Thus, we quantify the anthropogenic perturbation of the ocean carbon sink \citep[as $\mathrm{S_{OCEAN}}$ in the Global Carbon Budget][]{friedlingstein2023,hauck2020}, and not the contemporary net air-sea \ce{CO2} flux with outgassing of river carbon into the atmosphere (as in RECCAP2)."

L206: "we define the improvement as" – I'm in two minds whether calling the statistic "improvement" is good as it's clear and intuitive, or if it should be more objective and phrased as "reduction in mean absolute difference" or something equally dry. On balance I'm happy how it is, given it's clearly defined, but will keep this comment here for completeness. It can be a little odd when positive and negative improvement gets discussed (e.g. L254, L258).

The term 'improvement' was used before (see e.g. Losa et al., 2012: https://doi.org/10.1016/j.jmarsys.2012.07.008, with positive and negative improvements in Figure 1 and 2).

L220: "EN4-OA" – this is a reasonable product to use for comparison, but my understanding is that it includes no observations beyond the assimilated data, just interpolation between data points. So calling it "partly-independent" or "non-assimilated" (L244) may be misleading. Furthermore, it could have been introduced in the previous section.

Yes, we refrain from this wording because EN4-OA and EN4 are not independent. We have also moved the introduction of EN4-OA to the previous section.

L228: "in particularly" – in particular

Thank you, done.

L240: "particularly much" – "particularly"

Thank you, done.

L241: "Albeit negative side effects of temperature assimilation" – how is it judged that the temperature assimilation is responsible?

"Tests with the assimilation of temperature alone show negative side-effects of temperature assimilation on SSS in some locations. In the final set-up with combined assimilation, negative effects on SSS are found in 9\% of the observed area." - added to the manuscript.

Fig. 1 and others: My instinct would be to plot ASML – OBS rather than ASML – FREE. However, I've argued about this with coauthors on papers before, and appreciate others

strongly feel ASML – OBS is the better choice. So I'm merely flagging it as something to consider, I can see the argument both ways.

We have chosen ASML - FREE throughout the manuscript because it allows us to visualize comparatively small changes in some of the biogeochemical variables. On the one hand, for temperature and salinity, ASML-OBS provides a clear picture of the model error after data assimilation. On the other hand, for the biogeochemical variables, FREE-OBS and ASML-OBS are visually too similar to recognize the differences. Showing ASML-FREE for all variables allows us to recognize similarities between the effects of DA on different variables.

L275: "see Appendix Text A1 for further discussion". Appendix Text A1 is a single short paragraph, I don't understand why it's in an appendix. It would be better in the main manuscript, either here or in the Discussion section.

We have moved it here.

L276: "Thus, it can be assumed that the velocities in the upper part of the ocean are also well represented." I don't think you can make this assumption, certainly not for vertical velocities. See e.g. Raghukumar et al. (2015, https://doi.org/10.1016/j.pocean.2015.01.004) and Gasparin et al. (2021, https://doi.org/10.1016/j.ocemod.2021.101768). The data assimilation will continually update the observed variables to better match the observations, without necessarily leading to improvements in non-observed variables such as velocities – although of course that's the aim. The current study certainly doesn't seem to have the issues with vertical velocities the above studies do, but without providing assessment of the wider circulation there's no guarantee it's improved.

We agree that there is no guarantee that it's improved and have therefore rephrased: "This can be interpreted as an indication that the velocities in the upper part of the ocean are also well represented."

The advantage of referring to T and S observations is that these are directly comparable, which is not the case for velocities (that are partly parametrized in FESOM). Furthermore, the modeled boundary layer cannot be directly compared to a classically defined mixed layer either.

L280: "4 Results" – Section 3, "Effect of DA on ocean physics" is also results. Perhaps Section 4 should be "Effect of DA on ocean biogeochemistry".

Yes, we have adjusted the section titles.

L282: "The ocean absorbs 2.78 Pg C dec$^{-1}$" – is this the correct unit? From Fig. 4a, it looks to be absorbing 2.78 Pg C yr$^{-1}$ on average over the decade.

Thank you. Indeed, this was a typo and is now fixed.

L290: "air-sea $CO_2$ flux (negative: into the ocean)" – if negative's into the ocean shouldn't it be "sea-air $CO_2$ flux"?

While the direction of air-sea CO2 flux is not uniformly defined in the literature, the term 'air-sea' is commonly used for both for some reason, see e.g. Global Carbon Budget (Friedlingstein et al., 2023): 'air-sea flux' is positive into the ocean; and Roobaert et al. (2023): 'air-sea exchange' is negative into the ocean.

L301: While STSS+ is broadly the northern bit and STSS- southern, it's a bit more nuanced than that and that should be reflected in the text.

We have rephrased this (with credits to the other reviewer's suggestions):

*"The part of the STSS characterized by a positive \ce{CO2} flux difference between ASML and FREE, which we call the STSS+ and in which the \ce{CO2} uptake is reduced through the assimilation, roughly forms an outer (northern) ring around the STSS region."*

*"In contrast, the part of the STSS characterized by a negative \ce{CO2} flux difference between ASML and FREE, which we call the STSS- and in which the \ce{CO2} uptake is increased through the assimilation, is fragmented and roughly consists of segments of an inner (southern) ring."*

Fig. 5: Add to the caption that the lines in a and b denote the regions, and the hashing (striping?) denotes STSS+.

Added this.

L462: "a pCO2-independent proxy for primary production" – I'm not sure "pCO2-independent" is needed here, I don't quite understand what's meant.

We agree that it is not needed here and have deleted this word.

Originally, we meant to point out that there is no direct relationship of chlorophyll and $pCO_2$ through carbonate chemistry - unlike for all other variables (T, S, DIC and Alk) that are included in the observation comparisons.

L480: "as the modelled phytoplankton growth is temperature-dependent" – how sure are you the change is due to the direct temperature dependence rather than the indirect influence of stratification and mixing changes?

We cannot separate these effects and have therefore rephrased the text:

"Surface chlorophyll changes follow SST changes (\cref{fig:chl} and \cref{fig:SST_glob}). As the modeled phytoplankton growth is temperature-dependent \citep{gurses2023}, the similarity of spatial patterns indicates a direct temperature effect. In addition, indirect temperature effects on plankton dynamics due to stratification and mixing changes may contribute, but the link between sea surface temperature and mixing is not straight-forward (not shown)."

As the link between sea surface temperature and mixing is not straight-forward, the temperature-dependence of growth is a more likely candidate to explain the similar spatial patterns of SST and chlorophyll changes (Figure R1).

[Figure]

Figure R1: Spatial patterns of the difference ASML-FREE for surface chlorophyll, SST and boundary layer depth.

L515: "There are two other data assimilating BGC model approaches" – there are many other data assimilating BGC model approaches! Perhaps a more accurate phrasing might be: "We compare here to two other data assimilating BGC model approaches …"

Thank you for the rephrasing suggestion, we used it.

L524: "suggesting that a flawed representation of ocean physics as an argument for the models underestimating the CO2 flux trend is unlikely" – I broadly agree, though it may depend on how well the wider circulation is represented.
L559: "suggests that the physical processes are already well represented in FREE" – again I broadly agree, but there may still be pertinent limitations, especially depending on the time and space scale.
L565: "the adjustment of the ocean's carbon cycle to changes in the circulation" – true, though it's also possible that this might itself introduce biases in the carbon chemistry. See e.g. Lebehot et al. (2019, https://doi.org/10.1029/2019GB006186).

We agree with the reviewer's last three points and will mention these limitations in our discussion.

---

## Author Response (AR1)

**Answers to the reviewers' comments**

We would like to thank the editor and both reviewers for their feedback and many helpful suggestions to improve the text, for pointing precisely to small errors, and for their contributions to the discussion.

In response to the reviewers' comments, we have made the following major changes:

- We reproduced the last year of our simulations to provide additional output on the uncertainty represented by the ensemble
- We saved additional diagnostics (also for the last year of the simulation) on the sources and sinks of DIC and alkalinity through biological processes and advective and diffusive transport. These allow us to better understand and quantify where biological or physical processes affect $pCO_2$, and where biological and physical processes compensate
- We expanded the introduction
- We restructured the method section and added explanations for technical terms
- We restructured the result section and worked on a more precise wording, also including the new information based on the additional output

Otherwise, we updated the order of authors. The author's contributions are still the same.

In the following, we respond to the reviewers' comments point-by-point. Note that when text was modified and/or new text included, we use extracts from the LaTeX differences template. This highlights deleted text in crossed red text and new text in blue, underlined).

**RC 01**

1 **Reviewer's comment:**

The manuscript presents an application of an ensemble-based physical data assimilation technique to a global biogeochemical ocean model, with a focus on the effect of physical data assimilation on climate-relevant carbon estimates. The manuscript is mostly well written and offers some valuable insights on the effects of physical DA, but the text could be improved in places and several aspects of the DA experiments should be examined further.

One aspect that is becoming more important in modeling studies but is seemingly ignored in the current version of the manuscript is the reporting of model uncertainty -- even though ensembles are used to generate the results. The authors mention ranges of estimates when reporting results from other studies. However, in their own analysis, the focus is solely on the ensemble mean, without examining the full model ensemble or reporting any uncertainty estimates. It would be beneficial to explore ensemble-based ranges of estimates and compare them to the improvements brought about by data assimilation. This could lead to interesting questions, such as the

extent to which data assimilation constrains estimates and whether the estimates improve in areas where they are more constrained. Additionally, figures like Fig. 4 and the seasonal difference plots could be enhanced by including uncertainty estimates, such as the ensemble standard deviation or the interquartile range.

Answer:

Thank you for the suggestion. Indeed, a reduction in the uncertainty of the $CO_2$ flux estimate would be a very relevant result in addition to an improved estimate of the mean $CO_2$ flux.

However, the standard deviation in the Kalman filter methodology does not directly translate into an uncertainty estimate. Here, the ensemble standard deviation (STD) of the variables affected during the assimilation step (T, S, SSH, u, v) is reduced. In ASML, most of the reduction in ensemble spread occurs over the course of the first year. After that, the STD remains stable, precisely because we tune our ensemble perturbation and ensemble inflation in such a way that the STD of temperature is maintained after the initial phase (Figure R1; yellow and green lines).

[Figure]

Figure R1: Ensemble standard deviation for 3D-temperature. Note: No volume-weighting applied for the global mean (includes empty cells).

It is thus expected that the ensemble standard deviation of $CO_2$ flux decreases as well in ASML, but this is a result of the model and not part of the tuning. Indeed, we find that the STD for the local $CO_2$ fluxes in ASML is reduced to about 75-80% of the STD in FREE after the first year of assimilation (see example in Figure R2; however, this data is not area-weighted).

[Figure]

Figure R2: Ensemble standard deviation for the CO2 flux. Note: No area-weighting applied for the global mean.

We have added information on the uncertainty estimates in the revised manuscript (sub-subsection 2.3.1, subsection 3.2 and section 4). Rerunning the simulations was required for additional ensemble member output, and to save computing, we did this only for the year 2020.

In the manuscripts, this reads:

*2.3.2 Assimilation method and implementation*

*(Line 220 in manuscript)*

> the forgetting factor is set to either 0.99 or 1.0 depending on the ensemble standard deviation of temperature. The ensemble standard deviation of the local instantaneous air-sea $CO_2$ fluxes that results from the perturbation of physical fields is larger than that of the global $CO_2$ flux, with a mean standard deviation of $0.32 \, \mathrm{mmol \, m^{-2} \, day^{-1}}$ for monthly means of local fluxes compared to a standard deviation of $0.0068 \, \mathrm{mmol \, m^{-2} \, day^{-1}}$ ($0.01 \, \mathrm{Pg \, C \, yr^{-1}}$) for the annual global flux in FREE in the year
> 225   2020. The largest ensemble standard deviation is generated in the Southern Ocean, the North Atlantic and the North Pacific

> (map in Fig. A1a), which corresponds to regions of high uncertainty in existing $CO_2$ flux estimates (Pérez et al., 2024; Hauck et al., 2023a; Mayot et al., 2024). However, the modelled standard deviation should not be understood as the true uncertainty of the model, but as a value dependent on tuning (Evensen, 2003).

*3.2 Effect of DA on global CO2 flux*

> 375   $0.08 \, \mathrm{Pg \, C \, yr^{-1}}$ in ASML (not significantly different according to F-test). Through DA, the ensemble standard deviation of the global $CO_2$ flux in 2020 is reduced from $1.0 \times 10^{-2} \, \mathrm{Pg \, C \, yr^{-1}}$ in FREE to $0.7 \times 10^{-2} \, \mathrm{Pg \, C \, yr^{-1}}$ in ASML.

*4. Discussion*

ing changes in these variables (Fig. 9, Fig. 10 and Fig. 11f). The uncertainty represented by the ensemble is reduced by the DA, which has the most obvious effect on the directly assimilated fields (SST in Fig. 6d and e and density in Fig. 8f). The ensemble standard deviation of the $CO_2$ flux, where it is large in FREE, is constrained by the DA to globally more uniform

610   and smaller values (Fig. 5c-f, Fig. 7c-f and Fig. A1). Only in the North Pacific, the standard deviation of $CO_2$ fluxes is equally high in ASML and FREE, precisely in a region that also presents a challenge for $pCO_2$ products (compare Fig. A1 and Mayot et al., 2024, Figure 5a). In the rest of the ocean, the reduced uncertainty represented by the ensemble does not necessarily

coincide with improved agreement with BGC observations.

The respective figures (and captions) have been updated to show the range of ensemble members through semi-transparent shading.

Figure 5:

[Figure]

Figure 6:

[Figure]

[Figure]

Figure 7:

[Figure]

Figure 8:

[Figure]

We have not marked the range of ensemble members in Figure 4 because, for area-integrated fluxes globally and zonally, the uncertainty is so small that it cannot be seen.

We provide Appendix Figure A1:

[Figure]

[Figure]

2 **Reviewer's comment:**

The manuscript emphasizes carbon storage through physical transport, i.e. "upwelling and subduction of DIC, as well as the physical transport of other biogeochemical tracers" (l 60). However, the role of biological carbon fixation and sinking of particulate organic matter seems underexplored. Given that the model includes both slow and fast sinking detritus variables, a more comprehensive examination of these processes would be valuable. Here, it would help to clarify whether the biological carbon export at 200m (l 379 and following) is primarily due to sinking or physical transport. A closer examination or clearer description of the effects of the DA on the biological drivers of carbon export would help to improve the manuscript.

Answer:

We would like to note that we're most interested in anthropogenic $CO_2$ uptake, which is primarily physically driven (e.g., Gruber et al., 2023 https://doi.org/10.1038/s43017-022-00381-x). Yet, on a regional scale, changes in the biological carbon sink contribute to the overall carbon balance and thus may have noticeable effects on the regional net $CO_2$ fluxes. A much closer examination of the biological carbon pump would be interesting, but is beyond the scope of this paper.

In response to the reviewer's comment, in the revised manuscript, we have analyzed additional output for the year 2020, as indicated below, namely:

biological net sources or sinks of DIC and alkalinity through combined biological processes:

- For DIC, the net biological term is the sum of photosynthesis, respiration, remineralization of dissolved organic carbon, and formation and dissolution of calcite (Gürses et al., 2023, equation A6).
- For alkalinity, the net biological term is the sum of nitrogen assimilation and remineralization, and formation and dissolution of calcite (Gürses et al., 2023, equation A7).

For these, differences ASML-FREE (integrated over 0-190m) are shown in Appendix Figure A15:

[Figure]

Appendix Figure A12:

[Figure]

Appendix Figure A10 (green line in a and b):

[Figure]

Sources and sinks at 0-190m
ASML - FREE

(a) DIC

(b) Alkalinity

In the depth range 0-190m, the biological source/sink term for DIC is negative (-7.5 PgC yr⁻¹ globally in FREE for the year 2020). It describes the net transformation of DIC into organic carbon, and therefore only contains the part of biologically fixed carbon that is not remineralised within this depth range again. Thus, while a small amount of this term might add to an increase or decrease of biomass at the same depth range on annual time-scales, most of it is transported to below 190m depth through sinking of detritus (-5.3 PgC yr⁻¹; the gravitational pump; Boyd et al., 2019: https://doi.org/10.1038/s41586-019-1098-2), and some of it is transported to below 190m depth through advection and diffusion of organic material (-1.4 PgC yr⁻¹).

Wherever we have found that DA has a considerable effect on the biological source/sink term in a certain region, we have indicated this in the manuscript (see Track Changes document). This reads:

*Section 3.2 Effect of DA on regional CO₂ fluxes and their drivers*

[revised manuscript text omitted]

**3 Reviewer's comment:**

The assimilation of physical observations that only directly updates the physical variables can lead to "shocks" in the biogeochemical variables. It would be valuable to know if the authors observed any negative effects of daily physical updates on the biogeochemical state, such as unexpected phytoplankton blooms (for example, caused by a deepening of the mixed layer transporting nutrients, formerly below the mixed layer, to the surface).

Answer:

We are not aware of any such shocks. This might relate to our overall finding that the modeled carbon fluxes and other inspected variables such as chlorophyll-a, NPP and plankton biomass act almost surprisingly indifferent to substantial differences in the model physics. The most rapid assimilation-induced changes take place in the first few months after the start of the assimilation, yet there was no noticeable shock.

**4 Reviewer's comment:**

Several aspects of the model setup and data assimilation process could benefit from further explanation or discussion. For instance, the restoration of surface salinity towards climatology may interfere with the assimilation of salinity data. It would be informative to know if the authors have experimented with switching off the nudging when or where salinity data is being assimilated, and how well the salinity climatology aligns with the assimilated data.

Answer:

The main effects of SSS assimilation and salinity restoring are to reduce the simulated SSS globally. In addition, there are certain regions of model bias, such as the Amazon river inflow area and the North Atlantic Current, where both methods are consistent with each other. While there are gaps in the SSS-CCI data near the poles, the

salinity restoring towards climatology is with global coverage. Experiments with and without salinity restoring show that without it, in FREE, sea surface salinity drifts by approximately +0.05 psu during the first year after switching it off. In ASML, the difference between switching salinity restoring off or on is smaller (less than 0.01 psu globally), because the assimilation compensates for the lack of restoring. In ASML, global SSS is reduced by approximately 0.15 or 0.2 psu compared to FREE, respectively, after one year, which shows that the assimilation has a stronger effect than the restoring. The best agreement with SSS-CCI observations is achieved when assimilation and salinity restoring are used simultaneously.

In summary, we added the following to the manuscript, 3.1 Effect of DA on ocean physics:

> The assimilation also improves the agreement with the assimilated SSS observations. Additional experiments with and without salinity restoring towards climatology show that the best agreement with the SSS-CCI observations is achieved by
> 375    simultaneously using assimilation and restoring. A benefit of the additional use of restoring is the global coverage of the SSS climatology. FREE shows a global SSS bias (0.49 psu, Fig. 1d). The assimilation leads to a global surface freshening (Fig. 1e).

**5 **Reviewer's comment:**

Similarly, the exclusion of temperature observations from the DA when the model-observation difference exceeds 2.4°C could use a better explanation, as this seems to hinder assimilation where it might be most needed.

Answer:

By excluding these observations, the aim is to prevent strong and sudden corrections from making the model unstable, especially in the initial phase. Instead, a 'gentler' correction is made by assimilating neighboring points. Because we use a gap-filled SST observational product, observations are continuously available in the neighboring domains. We have added some text to reflect this to the manuscript, on SST assimilation:

> than the nominal resolution of the model grid. An observation error standard deviation of 0.8°C is prescribed for the DA following Nerger et al. (2020). Observations are excluded in the DA process if the difference between the model and observation exceeds three times the observation error standard deviation, thus 2.4°C, and at grid points with sea ice in the model, as in Tang et al. (2020) and Mu et al. (2022). This exclusion keeps the model stable despite large differences between model and
> 180    observations at these sites, in particular as water temperature and salinity develop differently under sea ice than under the influence of the atmosphere (Tang et al., 2020). Instead, a 'gentler' correction is made by assimilating neighboring points. After the initial phase, about 7% of SST observations are excluded because of the 2.4°C-threshold. Nevertheless, the data assimilation still has a strong effect in areas where these large model-observation discrepancies are typically found (North Atlantic, Japan and Southern Ocean).

**6 **Reviewer's comment:**

To improve readability, particularly for readers less familiar with data assimilation techniques and carbon modeling, brief explanations of key concepts and modeling choices would be beneficial. These would include descriptions of the term used to perturb atmospheric forcing, the role of ensemble inflation, and the rationale behind the choice of y_DIC and y_Alk in Equations 4 and 5 (see also my specific comments below). Currently, the manuscript often uses references to other studies to motivate implementation details, and an additional sentence here and there could help the reader to better understand these details without having to go through other papers.

In places, the structure of the manuscript can be improved to enhance clarity and flow. Sections 4.2 and 4.3 are quite lengthy and could be subdivided based on location (Southern Ocean, Atlantic) and the different data products used in the comparisons. Section 3, which contains results from the two ensemble simulations, could be merged with Section 4 to create a more cohesive results section.

Answer:

Thank you for the suggestion. We have rearranged the sections and section titles accordingly. The structure of the revised manuscript is now as follows:

To add structure to Sections 3.2.1. and 3.2.2, we use bold font to state which region is described in the following paragraph, e.g.:

470    **STSS**$_{SO}$    In the northernmost biome of the Southern Ocean, the subtropical seasonally stratified biome (STSS,

$_{SO}$), the mean oceanic $CO_2$ uptake is comparably high (Fig. 5a).  The uptake is largest in austral winter

**7 Reviewer's comment:**

Overall the figures look very good and are helpful, I only have a minor suggestion here: it might be more informative to report ASML-OBS instead of ASML-FREE in Figures 1-3. This would provide a clearer picture of the model error following data assimilation. Also, some of the figures, such as Figure 7, have lots of whitespace that could be reduced.

Answer:

Indeed, for temperature and salinity, ASML–OBS provides a clear picture of the model error after data assimilation (see SST, Figure R6).

[Figure]

**Figure R6**: FREE-OBS and AMSL-OBS for SST, useful to illustrate the model error before and after assimilation

However, for the biogeochemical variables, FREE–OBS and ASML–OBS are visually too similar to recognize the differences (see chlorophyll, Figure R7).

[Figure]

**Figure R7**: FREE-OBS and AMSL-OBS for chlorophyll, the effect of the assimilation is almost invisible

Therefore, we have chosen to show ASML–FREE because it allows us to visualize comparatively small changes in the biogeochemical variables. Showing ASML–FREE for all variables throughout the manuscript allows one to recognize correlations between the effects of DA on different variables.

**8 **Reviewer's comment:**

L 8: "the mean CO2 uptake increases by 0.18 Pg C yr−1": Add "regionally" here to make it explicit that this increase is not a resulting global estimate.

Answer:

Thank you for the detailed comments here and below. Implemented here:

> 10 Ocean during winter. South of $50\,°$S, winter $CO_2$ outgassing is reduced and thus the  regional $CO_2$ uptake increases by $0.18\,\mathrm{Pg\,C\,yr^{-1}}$ through the assimilation. Other particularly strong regional effects on the air-sea $CO_2$ flux are located in the

**9 **Reviewer's comment:**

L 40: "the model mean": It would be helpful to the reader to add a few words about the kind of models that were considered here.

Answer:

Done here:

> atmospheric oxygen data and atmospheric inversions (Friedlingstein et al., 2023). For the years 2010-2020, $pCO_2$ products included in the Global Carbon Project suggest a mean oceanic sink of $3.0 \pm 0.4\,\mathrm{Pg\,C\,yr^{-1}}$, while the  mean
>
> 45  of Global Carbon Project GOBMs is $2.5 \pm 0.4\,\mathrm{Pg\,C\,yr^{-1}}$  (data provided by Friedlingstein et al., 2023). Trends over the same time period are $0.7\,\mathrm{Pg\,C\,yr^{-1}\,dec^{-1}}$ and $0.3\,\mathrm{Pg\,C\,yr^{-1}\,dec^{-1}}$, respectively

**10 Reviewer's comment:**

L 65: "DIC" was used before the abbreviation is introduced here (l 59). The earlier sentence actually makes a quite similar point about subduction of DIC and also mentions upwelling, perhaps this could be made more concise.

Answer:

Rearranged to merge the two sentences that make similar points into one sentence, introducing DIC at its first use, now reads:

> While previous studies indicate that the available BGC observations, when assimilated in isolation, are too sparse to constrain the modeled carbon cycle (Verdy and Mazloff, 2017; Spring et al., 2021), the assimilation of physical variables is expected to have a significant indirect effect on the modeled
>
> 75   air-sea $CO_2$ fluxes (Bernardello et al., 2024). This is because the uptake of atmospheric $CO_2$ depends ultimately on the modeled physical carbon transport between the surface, the mixed layer and the deep ocean in the form of dissolved inorganic carbon (DIC) through mixing, upwelling and subduction (Doney et al., 2004). According to current knowledge, ocean physics is the dominant driver

**11 Reviewer's comment:**

L 65: "It was shown that assimilating ocean physics at the initial state of a model simulation has a stronger and more positive impact on the modeled carbon cycle than assimilating the BGC initial state": Is this due to the lack of BGC observations mentioned earlier, the importance of physical processes for carbon export, or a large physical model error that cannot be decreased through BGC DA?

Answer:

Fransner et al., 2020 relate the strong and positive effect of assimilating ocean physics to the strong control ocean physics exerts on the biogeochemical variability on interannual to decadal time scales (rather than low availability of BGC observations or strong physical model errors). Thus, we have added:

According to current knowledge, ocean physics is the dominant driver of interannual variability of the global air-sea $CO_2$ flux and also responsible for stagnation and acceleration of the $CO_2$ uptake on decadal scales (Doney et al., 2009; Keppler and Lands

. Related to the strong control that physics exert on the interannual variability of air-sea  $CO_2$ fluxes, it was shown

85 in one idealized study that assimilating ocean physics at the initial state of a model simulation has a stronger and more positive impact on the modeled carbon cycle on interannual time-scales than assimilating the BGC initial state (Fransner et al., 2020).

**12 Reviewer's comment:**

The next sentence brings up the question of which processes are most important. Maybe a few candidates could be named and briefly discussed here before going into the details of the DA algorithm.

Answer:

Naming and discussing a few candidates here:

physics DA. The question therefore arises  to what extent an ecosystem model coupled to a data-assimilated physical model also represents a more realistic biogeochemistry.  , and which mechanisms are responsible for the response of the $CO_2$ flux in physics

110 DA approaches. One possible driver is the physical transport of DIC and alkalinity because velocities and diffusivity are changed by the DA, affecting in particular the upwelling of carbon-rich waters and subduction, which is important to capture the ocean storage of anthropogenic carbon (Davila et al., 2022). Furthermore, physics DA may change $pCO_2$ directly through its temperature-dependence, an effect emphasized by Verdy and Mazloff (2017). Additionally, the modelled biological pump might be altered, for example through the temperature-dependency of phytoplankton growth or through effects of stratification

115 on nutrient availability.

**13 Reviewer's comment:**

L 70: "continuously assimilating ocean-physics for eleven years": A bit more detail could be useful here as well: What does assimilating ocean physics entail, what observations are being used for the DA here?

Answer:

More details added to the introduction:

> of ocean physics a prerequisite for a realistic simulation of the contemporary $CO_2$ flux. We here use ensemble-based data
> assimilation of ocean physics into a global ocean biogeochemistry model aiming to improve the modeled air-sea $CO_2$ flux
> 90   for the years 2010-2020. For
>  this, we continuously assimilate temperature and salinity
> observations from remote-sensing at the surface and from in-situ profile measurements for eleven years

> and update
> 95   the modelled temperature, salinity, horizontal velocities and sea surface height, using an ensemble Kalman filter variant
> (Nerger et al., 2012).

**14 Reviewer's comment:**

L 89: "The model allows for a variable mesh resolution": What is a typical coarse and fine resolution used in the model grid?

Answer:

We have now moved Section "Simulation set-up" up here, clarifying:

> ### 2.2 Simulation set-up
>
> The model setup for both simulations closely follows Gürses et al. (2023). The mesh resolution is nominally 1 degree, ranging
> between 120 km and 20 km with enhanced resolution in the equatorial belt and north of $50\,°N$ (126858 surface nodes). It has 47
> vertical layers with thickness ranging from 5 m at the surface to 250 m in the deep ocean, as described by Scholz et al. (2019, COF

**15 Reviewer's comment:**

L 93: A salinity flux of 0.1m/day? Please describe this better.

Answer:

Thanks for asking, in fact, this number was a typo. We corrected the number and added Eq. (1) to clarify:

> . The surface salinity (SSS) is restored towards the
> World Ocean Atlas climatology through a fictional surface flux with $v_{SSS} = 50\,m/300\,days$ according to equation 1 and as in
> Gürses et al. (2023):
>
> $$(SSS_{clim} - SSS_{model}) * v_{SSS} * (h_{surf})^{-1} \qquad (1)$$
>
> 135   with surface-layer width $h_{surf}$. A detailed description of FESOM2.1 and a model assess-

For the example of a salinity bias of 0.5 psu and with the surface-layer width being around 5m (more or less depending on sea surface height etc.), this would yield a correction of approx. 0.016 psu per day.

**Reviewer's comment:**

L 96: "DIC" is introduced again, a quick search shows 7 introductions of "DIC", also counting captions.

Answer:

Thanks, we only kept the introduction of "DIC" once in the Introduction, and once more in the Conclusion.

**Reviewer's comment:**

L 117: "observations are weighted by distance": This is not a precise statement that could confuse some readers, express more clearly that the ensemble estimated correlation between a model grid point and an observation is down-weighted using a distance-based metric. Is vertical localization applied as well?

Answer:

The localization acts in the horizontal only. We have phrased more precisely:

> With localization of the LESTKF,  the observation error is increased for an increasing horizontal distance between an observation and a model grid point, which weighs down the influence of a more distant observation. This avoids that the model is influenced by observations at distant locations
>
> 210    through spurious ensemble estimated correlations. We use a localization radius of 200 km and choose a 5th-order polynomial

**Reviewer's comment:**

Eq. L 124: It would be useful to add equation numbers to all equations, even those that are not referenced in the text, so that they can be more easily referenced in other texts, such as this one.

Why does a larger ensemble amplify rand? It does not seem that intuitive to have larger perturbations in a larger ensemble.

Answer:

We added equation numbers to all equations.

The incomplete definition of 'rand' in the initial manuscript has led to an obvious misunderstanding: In fact, there are no larger perturbations in a larger ensemble. The factor ($N_{ens}$-1) compensates that the values of 'rand',

defined as elements of a stochastic matrix which sum up to 1, become smaller with increasing ensemble size because the matrix becomes larger. In detail, the values for rand are generated by Second-Order Exact Sampling from a trajectory of atmospheric forcing fields, a method introduced by Pham et al., see e.g.:
https://doi.org/10.1175/1520-0493(2001)129<1194:SMFSDA>2.0.CO;2
and briefly explained here:
https://pdaf.awi.de/trac/wiki/EnsembleGeneration

To clarify, in the updated manuscript we added a few sentences on the generation of the initial perturbation, and we have now redefined the stochastic element (still called 'rand'), so that it already includes the factor (N_ens-1) that initially caused confusion.
* * *
To maintain ensemble spread, we apply a perturbed atmospheric forcing with an autoregressive perturbation $(\text{perturb}_n)$ $(\text{perturb}_{e,n})$ at every model time step (n) to each ensemble member (e), with:

$$\text{perturb}_{n+1} = (1 - \text{arc}) * \text{perturb}_n + \text{arc} * s * (N_{ens} - 1) * \text{rand}$$

230

$$\text{perturb}_{e,n+1} = (1 - \text{arc}) * \text{perturb}_{e,n} + \text{arc} * s * \text{rand}_e \qquad (2)$$

where rand is a stochastic element that is based on a covariance matrix derived, again generated by second-order exact sampling from a 72-days-long period trajectory of atmospheric forcing ; the fields that captures patterns of day-to-day atmospheric variability. The autoregression coefficient (arc) is can be used to tune how quickly the perturbation changes and is set to
235 the inverse number of model steps per day; and. $s$ is a scaling factor for each perturbed atmospheric forcing field. For spe-
* * *
 **Reviewer's comment:**

L 153: "model values are computed as the average of the grid points of the triangle enclosing the observation because the number of observations is fewer than model grid points": Averaging is required to interpolate the model solution at the observation locations, why is this dependent on the number of observations?

Answer:

Thanks for pointing out how this can lead to confusion. In fact, we simply meant:

1. If observations are spatially highly resolved, they are interpolated to the model grid (as for SST and SSS).
2. If observations are available only at a few points, it is the other way round and the model solution is interpolated to the observation locations (as for the profile data).

Because this was unnecessarily confusing, we have left it out. The text now reads:

190    The assimilated temperature and salinity profiles are taken from the EN.4.2.2 data set (Good et al., 2013). The EN4 dataset contains quality-controlled profiles from various in-situ ocean profiling instruments. To assimilate the profiles, the observations

are assigned to the respective model layers (depth range) in the vertical. In the horizontal, the model values are computed as the average of the grid points of the triangle enclosing the observation. The observation error standard deviation is set to $0.8\,°C$ for temperature and to 0.5 psu for salinity, as in Tang et al. (2020).

**20 Reviewer's comment:**

L 157: This information about the model grid is missing from Section 2.1 where the model grid is described for the first time. It would also be useful to describe the atmospheric forcing before describing the perturbation to it (Section 2.2.1).

Answer:

Rearranged to:

2.1 Model FESOM-REcoM
2.2 Simulation set-up *(here, we describe the grid and atmospheric forcing)*
2.3 Data Assimilation
2.3.1 Assimilated observations
2.3.2 Assimilation method and implementation *(here, we describe the perturbation to the atmospheric forcing)*

**21 Reviewer's comment:**

L 171 "the river flux adjustment (...) is applied to the pCO2 products. ...": It is not entirely clear what this means, the focus here is just the CO2 flux associated with the oceans, I presume? The next sentence provides some more information but it seems to imply that the RECCAP2 CO2 flux is not being used for comparison, when previous sentences stated that it was. Some clearer language would be useful here.

Our model and other GOBMs do not account for the natural river flux, which is (simplified):

1. rivers carry organic carbon into the ocean
2. as a consequence, carbon, once remineralized, outgasses from the ocean into the atmosphere
3. fixation of atmospheric $CO_2$ by terrestrial and freshwater ecosystems, and export via rivers ( $\rightarrow$ 1.)

The river flux adjustment (https://www.nature.com/articles/s41586-021-04339-9) serves to make GOBM estimates of the air-sea $CO_2$ flux comparable with other estimates, which, in contrast, do account for the river flux.

To clarify, we have rephrased:

 We present $CO_2$ flux estimates for the period 2010-2020, that are

290    compared to the 'Regional Carbon Cycle Assessment and Processes 2' (RECCAP2) global air-sea $CO_2$ flux estimates (DeVries et al., 2023).  The RECCAP2  p$CO_2$ products account for oceanic outgassing of river carbon into the atmosphere. To make them comparable with our estimate stemming from a model without river carbon input, we apply a river flux adjustment (Friedlingstein et al., 2023; Regnier et al., 2022)  to the RECCAP2 p$CO_2$ products. Thus, we quantify the anthropogenic perturbation of

295    the ocean carbon sink  (as $S_{OCEAN}$ in the Global Carbon Budget Friedlingstein et al., 2023; Hauck et al., 2020), and not the contemporary net air-sea $CO_2$ flux with outgassing of river carbon (as in the original RECCAP2 p$CO_2$ products).

**22* Reviewer's comment:**

L 183: Should the US East Coast be considered subpolar, are all regions characterized by seasonal stratification, or does SPSS stand for something different here? A alternative choice of region names may be suitable and would avoid confusion with the region names in the Southern Ocean.

Answer:

According to the definition of Fay and McKinley, the STSS, SPSS and ICE biomes exist analogously in both hemispheres (https://doi.org/10.5194/essd-6-273-2014). Therefore, there is an SPSS and STSS biome in the Southern Ocean and in the North Atlantic, of which we discuss only specific parts (e.g. the Coastal SPSS).

The Fay and McKinley biomes are used widely in the ocean carbon cycle community (see e.g. RECCAP papers, https://reccap2-ocean.github.io/publications/).

To avoid confusion with the regions names in the North Atlantic (NA) and Southern Ocean (SO), we have added subscripts to the names, e.g. $STSS_{SO}$+ and Coastal $SPSS_{NA}$−. The "+" and "−" symbols denote the sign of the

effect by which each region is defined. In the revised manuscript, this reads:

> To study the effect of DA on the $CO_2$ flux, we define regions where the effect is pronounced and where different mechanisms
>
> 295  are active. , based on the biomes defined by Fay and McKinley (2014). These are, going polewards from the subtropics in each hemisphere, the Subtropical Seasonally Stratified Biome (STSS), the Subpolar Seasonally Stratified Biome (SPSS) and the Sea-Ice Biome (ICE). In the Southern Ocean ($_{SO}$), within the $STSS_{SO}$, we differentiate between the area where the assimilation leads to a more positive air-sea $CO_2$ flux (positive: out of the ocean), referred to as $STSS_{SO}+$ and the area where the assimilation leads to a more negative air-sea
>
> 300  flux, the  $STSS_{SO}-$ (Fig. 5a and b). In the North Atlantic ($_{NA}$), we consider four coherent regions within the $STSS_{NA}$ and $SPSS_{NA}$, defined by the time-mean difference of the air-sea $CO_2$ fluxes in ASML and FREE ($\Delta F_{CO_2}$, ). The Central $STSS_{NA}-$ and Western $STSS_{NA}+$ are located in the central North Atlantic $STSS_{NA}$ biome and are confined by $\Delta F_{CO_2} < -1\,\mathrm{mmol\,C\,day}^{-1}\mathrm{m}^{-2}$ and $\Delta F_{CO_2} > 1\,\mathrm{mmol\,C\,day}^{1}\mathrm{m}^{-2}$, respectively (see Fig. 7b). The Newfoundland Basin and Coastal $SPSS_{NA}-$ are part of the $SPSS_{NA}$. The former is located east of Newfoundland and south of
>
> 305  Greenland, and is confined by $\Delta F_{CO_2} > 3\,\mathrm{mmol\,C\,day}^{-1}\mathrm{m}^{-2}$; and the latter is located off the North American coast and confined by $\Delta F_{CO_2} < -1\,\mathrm{mmol\,C\,day}^{-1}\mathrm{m}^{-2}$. The Central $STSS_{NA}-$ and Western $STSS_{NA}+$ lie on the warm side of the North Atlantic Current (NAC), and the Newfoundland Basin and Coastal $SPSS_{NA}-$ lie on the cold side of the NAC, which is evident from the modeled surface velocity field (Fig. A2a).

**23 Reviewer's comment:**

L 185: Please explain "NAC".

Answer:

Defined in Line 307:

> North Atlantic Current (NAC),

**24 Reviewer's comment:**

Eq 1 and 2: Is there an easy to communicate motivation for the choice of γ_DIC and γ_Alk ?

Answer:

We describe the motivation here:

and surface chlorophyll physical and biogeochemical fields. In order to assess the drivers of dynamic DA effects on surface pCO$_2$, it is useful to distinguish between different variables that constitute the change in pCO$_2$. Oceanic pCO$_2$ varies mainly with temperature, DIC and alkalinity. Thus, we decompose changes in pCO$_2$ are decomposed after the simulation into their contributions from changes in SST(SST), surface DIC (DIC) and and surface alkalinity (Alk)following the linear. For that, we

315     apply the following approximations of Sarmiento and Gruber (2006) and Takahashi et al. (1993):

[ equations ]

the computation. The sensitivities $\gamma_{DIC}$ and $\gamma_{Alk}$ describe how pCO$_2$ varies with changes in one variable while keeping all

325    other variables constant. For the sensitivities, we use an approximation derived from seawater carbonate chemistry following Sarmiento and Gruber (2006):

[ equations ]

In the appendix, we illustrate that the net pCO$_2$ difference (ASML – FREE; blue line in panel a) can approximately be explained by the sum of these three terms. Figure A9:

[Figure]

Here, the non-thermal effect is calculated, firstly, as the sum of alkalinity and DIC effects, and secondly as the residual (i.e. "net ΔpCO₂ minus thermal").

25 **Reviewer's comment:**

Eq 1, 2 and 3: Previously Delta denoted the difference between ASML and FREE, is this still the case here? If so, are the regular terms (e.g. DIC in Eq 1 or the terms in γ_DIC) from the FREE experiment? This should be mentioned in the description.

Answer:

Yes, delta is the difference between ASML and FREE and the regular terms are calculated from the average of the two simulations - this has been added here (Line 321):

>  Here, differences between ASML and FREE are denoted by $\Delta$; else, the average of ASML and FREE is used for the computation. The sensitivities $\gamma_{DIC}$ and $\gamma_{Alk}$ describe how $pCO_2$ varies with changes in one variable while keeping the other

**26 Reviewer's comment:**

L 220: Why not mention EN4-OA earlier when the other data products are introduced?

Answer:

Makes sense, we have rearranged this. Firstly, all observational products that are assimilated are introduced in Section 2.3.1. Secondly, all observations used for validation are introduced in Section 2.4, here:

[revised manuscript text omitted]

**30 Reviewer's comment:**

L 373: "the effect of the DA is towards increased uptake of CO2 during boreal summer and autumn in ASML (Fig. 6g). This prevents summer outgassing": The increased summer uptake prevents summer outgassing, isn't this just describing the same effect? I would suggest rewording this sentence.

Answer:

Reworded to emphasize the seasonal difference between uptake and outgassing:

Central STSS$_{NA}$−    In the Central STSS$_{NA}$−, the effect of the DA is  overall towards a more negative flux of $CO_2$ from May to November (Fig. 7g).  Thus, spring and autumn $CO_2$ uptake are increased and summer outgassing is prevented in ASML (Fig. 7c). The reason for  decreased surface $pCO_2$ is higher

**31 Reviewer's comment:**

L 411: "(difference of FREE and SOCAT in (Fig. 9a); difference of ASML and SOCAT not shown)": The figure label claims that ASML - SOCAT is shown.

Thank you for noting this. Figure data and labels have been updated to show FREE - SOCAT, as indicated in the text. Figure 9:

[Figure]

pCO₂ FREE - SOCAT 2010-2020

pCO₂ Improvement 2010-2020

**RC 02**

32 **Reviewer's comment:**

The manuscript describes a study assimilating temperature and salinity observations into a global physics-biogeochemistry ocean model, with the aim of improving the modelled air-sea $CO_2$ flux. The assimilation brought the model temperature and salinity closer to the assimilated observations, and had a mixed impact on the carbon variables and wider biogeochemistry. The global mean change was small, but could be regionally significant, with the mechanisms explored.

The experiments are well conceived, and the manuscript generally well written and well presented. I just have some comments where aspects could use clarifying or expanding on.

L51: "Data assimilation (DA) has been employed …" This paragraph doesn't need to be comprehensive, but could be modified and expanded a little to more fully represent the available literature. Valsala and Maksyutov (2010, https://doi.org/10.1111/j.1600-0889.2010.00495.x) ran a global assimilation for 1996-2004; not multidecadal but almost as long as the present study. The paragraph states "In each of these studies, an Adjoint or Green's Function DA approach is used", but the Gerber et al. (2009) study referenced used an EnKF – another non-adjoint/Green's function example is While et al. (2012, https://doi.org/10.1029/2010JC006815) who used a

sequential analysis correction scheme to assimilate pCO2. The paragraph opens by talking about "DA studies of the air-sea CO2 flux" in general terms, only semi-clarifying later that it's focussing on studies which directly assimilated pCO2 data. There have also been other studies which, like the present one, looked at the impact of assimilating other variables on the air-sea CO2 flux, e.g. Ciavatta et al. (2016; https://doi.org/10.1002/2015JC011496) and other papers from that group, and Ford and Barciela (2017, https://doi.org/10.1016/j.rse.2017.03.040).

Thank you for the references to the literature, great! We acknowledge these. The expanded paragraph reads:

> *Data assimilation (DA) can be employed to address the emerging discrepancies between $pCO_2$-products and models (Carroll et al., 2020). Several studies assimilating ocean surface $pCO_2$ have focused on specific regions (e.g., a baseline state of air-sea $CO_2$ fluxes in the Southern Ocean; Verdy and Mazloff, 2017), few years (e.g., optimized biogeochemical initial fields for the period 2009-2011 in Brix et al., 2015) or the climatological mean state (e.g., corrections of large-scale $pCO_2$ model biases in While et al., 2012). These studies capture well the assimilated $pCO_2$ observations, while obeying physical laws and biogeochemical (BGC) equations. Data assimilation also provides a better understanding of various components of the ocean carbon cycle, such as the transport of anthropogenic $CO_2$ in the ocean (e.g., a reconstruction of anthropogenic carbon storage since 1770 in Gerber et al., 2009), regional and interannual variability of the air-sea $CO_2$ flux (e.g., global reanalysis in Ford and Barciela, 2017; Carroll et al., 2020; Valsala and Maksyutov, 2010), the biological carbon pump (e.g., carbon export at a nutrient-rich and nutrient-poor site and estimation of BGC parameters related to air-sea $CO_2$ fluxes in Sursham, 2018; Hemmings et al., 2008) and specific ecosystems (e.g., the North West European Shelf ecosystem in Ciavatta et al., 2016, 2018). So far, however, there is no data assimilation product that provides a long-term, annually updated estimate of global ocean $CO_2$ uptake.*

**33 Reviewer's comment:**

L65: "It was shown that assimilating ocean physics at the initial state of a model simulation has a stronger and more positive impact on the modeled carbon cycle than assimilating the BGC initial state (Fransner et al., 2020)." In no way diminishing the motivation for this current study – which is undoubtedly important for the reasons stated in Fransner et al. (2020) and others – it could be clarified that this was a single model study and may or may not hold in general. The relative importance of physics vs biogeochemistry initialisation on different variables and time scales remains an open question – see e.g. the discussion in Section 4.4 of Lebehot et al. (2019, https://doi.org/10.1029/2019GB006186) and indeed the ultimate conclusions of this current manuscript.

Answer:

Thank you for providing the literature, which we have included:

>  $CO_2$ fluxes, it was shown
>
> 85  in one idealized study that assimilating ocean physics at the initial state of a model simulation has a stronger and more positive
>
> impact on the modeled carbon cycle on interannual time-scales than assimilating the BGC initial state (Fransner et al., 2020).
>
>
>
>  However, the relative importance of uncertainties in physical and biogeochemical fields generally remains an open research
>
> question (e.g. Séférian et al., 2014; Li et al., 2016; Lebehot et al., 2019). Therefore, we here use ensemble-based data assimila-

**34 Reviewer's comment:**

L67: "Therefore the question arises which processes are most important when altered physics change CO2 fluxes in DA approaches." I think I understand the meaning of this sentence, but it could be reworded for clarity.

Answer:

Reworded for clarity to:

> physics DA. The question therefore arises  to what extent an ecosystem model coupled to a data-assimilated physical
>
> model also represents a more realistic biogeochemistry
>
> , and which mechanisms drive the response of the $CO_2$ flux in physics DA
>
> 110  approaches. One possible driver is the physical transport of DIC and alkalinity because velocities and diffusivity are changed

**35 Reviewer's comment:**

L68: "to improve" – a better wording could be "to aim to improve"?

Answer:

Included in Line 90:

> aiming to improve the modeled air-sea $CO_2$ flux

**36 Reviewer's comment:**

L75-79: The issues discussed by Park et al. (2018) and others, mentioned later in the manuscript, could be introduced at this point.

Answer:

We describe these issues now in the Introduction, instead of later in the manuscript:

>  Several difficulties are associated with physics
> 100 DA into GOBMs. A common issue is erroneous equatorial upwelling leading to unrealistically high biological productivity in the tropics (Park et al., 2018; Gasparin et al., 2021; Raghukumar et al., 2015). Furthermore, any coupled ecosystem model is adapted to its associated physical model with its strengths and weaknesses through carefully selected parameter values  and a spin-up to near-equilibrium. Accordingly, the modeled carbon cy-
> 105 cle may react very sensitive to deviations from  the physical state that is typical for this model (Kriest et al., 2020; Spring et al., 2021). Potentially, this leads to biases in the carbon cycle through  physics DA. The question therefore arises  to what extent an ecosystem model coupled to a data-assimilated physical

**37 Reviewer's comment:**

L103: "Alkalinity is restored by a fictional surface flux of 10m/yr." Is there a reference for this, or was it introduced in this study?

Answer:

We follow the set-up of Gurses et al. (2023). This alkalinity restoring has been used by Hauck et al. (2013) and Schourup-Kristensen (2014) as well.

Gurses: doi.org/10.5194/gmd-16-4883-2023
Hauck: doi.org/10.1002/2013GB004600
Schourup-Kristensen: doi.org/10.5194/gmd-7-2769-2014

Citations added in Line 147:

> pute $pCO_2$ and air-sea $CO_2$ flux, employing the gas-exchange parameterization of Wanninkhof (2014). Alkalinity is restored
> by a fictional surface flux of  $10 \, \text{m} \, \text{yr}^{-1}$ (as in Hauck et al., 2013; Schourup-Kristensen et al., 2014; Gürses et al., 2023)

**38 Reviewer's comment:**

L121: "After each assimilation step, corrections are applied to the analysis state to ensure the consistency of model physics." Can you give an indication of whether these corrections need to be applied regularly or just occasionally?

Answer:

This has been clarified here:

temperature, salinity, horizontal velocities and sea surface height. After each assimilation step, corrections are applied to the analysis state to ensure the consistency of model physics: Salinity is set to a minimum value of zero and temperature to a minimum value of $-2\,°C$, if  the value is otherwise below. The increment of sea surface height (SSH)  is limited to two standard deviations of the ensemble. While in the simulation the correction was necessary for about 10% of SSH updates and $0.01\,‰$ of temperature values at each step, the correction of salinity was never required. The analysis step is

**39 Reviewer's comment:**

L148: How is the weekly-resolution SSS used in the daily assimilation?

Answer:

SSS data is provided daily. To clarify, see Line 186:

The assimilated SSS data is taken from the European Space Agency (ESA) Sea Surface Salinity Climate Change Initiative (CCI) v03.21 data set (Boutin et al., 2021). ESA-CCI contains daily data at a spatial resolution of 50 km, albeit not capturing temporal variability below weekly. The ESA-CCI observations are averaged to the FESOM2.1 model grid. We prescribe a

The daily sampling of data resolving weekly variability is described in Boutin (2021):
doi.org/10.1029/2021JC017676

It is not necessary that the observations capture the day-to-day variability, as the data assimilation has a comparatively slow effect: For example, it takes several months of assimilation to achieve the maximum feasible correction of a large-scale model bias.

**40 Reviewer's comment:**

L153: "model values are computed as the average of the grid points of the triangle enclosing" – what's done in the vertical?

Answer:

See Line 191:

The assimilated temperature and salinity profiles are taken from the EN.4.2.2 data set (Good et al., 2013). The EN4 dataset contains quality-controlled profiles from various in-situ ocean profiling instruments. To assimilate the profiles, the observations are assigned to the respective model layers (depth range) in the vertical. In the horizontal, the model values are computed as

41 **Reviewer's comment:**

L171: "For the comparison …" – this paragraph would benefit from a clearer explanation of what adjustments have been made to what products and why, including the model estimates from this study (which presumably have no river carbon inputs?).

Answer:

As both reviewers have asked for a clearer explanation, please see our answer to Reviewer's comment 21.

**42 Reviewer's comment:**

L206: "we define the improvement as" – I'm in two minds whether calling the statistic "improvement" is good as it's clear and intuitive, or if it should be more objective and phrased as "reduction in mean absolute difference" or something equally dry. On balance I'm happy how it is, given it's clearly defined, but will keep this comment here for completeness. It can be a little odd when positive and negative improvement gets discussed (e.g. L254, L258).

Answer:

The term 'improvement' was used before (see e.g. Losa et al., 2012:
https://doi.org/10.1016/j.jmarsys.2012.07.008, with positive and negative improvements in Figure 1 and 2).

**43 Reviewer's comment:**

L220: "EN4-OA" – this is a reasonable product to use for comparison, but my understanding is that it includes no observations beyond the assimilated data, just interpolation between data points. So calling it "partly-independent" or "non-assimilated" (L244) may be misleading. Furthermore, it could have been introduced in the previous section.

Answer:

Thanks, we have adjusted the wording, saying that EN4-OA is an objective analysis ingesting the assimilated EN4 profile data. We have also changed the text structure so that all comparison datasets are described in one place. Please see our answer to Reviewer's comment 26.

**44 Reviewer's comment:**

L228: "in particularly" – in particular

Answer:

Thanks, Line 366:

> FREE shows regional SST biases in  particular near strong currents or in eddy-rich regions,

Thanks, Line 381:

> particularly  in the North Atlantic Central STSS$_{NA}$ –

Answer:

We know from experiments during the test phase, assimilating only one variable at the time for a shorter period. Line 382:

> Southern Ocean STSS$_{SO}$ (Fig. 1f).  Tests with the assimilation of temperature alone show negative side-effects of temperature assimilation on SSS in some locations (not shown). In the final set-up with combined assimilation, negative effects on SSS are found in 9% of the observed area. Globally, the mean absolute difference is reduced

Answer:

We have chosen ASML - FREE because it allows us to visualize comparatively small changes in some of the biogeochemical variables. Please see our answer to Reviewer's comment 7.

48 **Reviewer's comment:**

L275: "see Appendix Text A1 for further discussion". Appendix Text A1 is a single short paragraph, I don't understand why it's in an appendix. It would be better in the main manuscript, either here or in the Discussion section.

Answer:

This paragraph has been expanded and is now included in the main manuscript (Section 3.1 Effect of DA on ocean physics):

locities(see Appendix Text A1 for further discussion). The . Throughout the assimilation period, spurious, spatially limited and often deep overturning structures emerge, evolve through several months or years, and disappear in the tropical Indian, Pacific and Atlantic basin (not shown). Thereby, the surface overturning cell sometimes breaks apart where it should extend over the equator, exposing the bottom cell to the surface (Fig. A8b). Transport in the North Atlantic at $26.5°N$, an indicator
430 for the strength of the Atlantic Meridional Overturning Circulation, is between 8-9 Sv in FREE. In ASML, during the first two years of assimilation, transport at $26.5°N$ decreases to below 3 Sv and, during the following years, recovers to 7-8 Sv (2016-2020). One possible cause is the effect of data assimilation on the eddy parameterisation (Gent and Mcwilliams, 1990). The parameterised eddy activity is relevant for the dynamics in the deep ocean, and corrupting it may have a negative impact on the large-scale oceanic circulation, as described in Sidorenko (2004, Chapter 5.5 onwards) for a previous version of the ocean
435 model FESOM.

49 **Reviewer's comment:**

L276: "Thus, it can be assumed that the velocities in the upper part of the ocean are also well represented." I don't think you can make this assumption, certainly not for vertical velocities. See e.g. Raghukumar et al. (2015, https://doi.org/10.1016/j.pocean.2015.01.004) and Gasparin et al. (2021, https://doi.org/10.1016/j.ocemod.2021.101768). The data assimilation will continually update the observed variables to better match the observations, without necessarily leading to improvements in non-observed variables such as velocities – although of course that's the aim. The current study certainly doesn't seem to have the issues with vertical velocities the above studies do, but without providing assessment of the wider circulation there's no guarantee it's improved.

Answer:

We agree that there is no guarantee that it's improved and have therefore rephrased: "This can be interpreted *as an indication* that the velocities in the upper part of the ocean are also well represented."

This indication becomes more reliable, though, through additional evaluation of horizontal surface velocities and mixed-layer depth:

420  The boundary-layer depth and mixed-layer depth are mostly reduced through DA. In particular, deep water formation events characterised by a mixed-layer depth of more than 1000 m or 500 m occur less frequently in ASML (not shown). This improves the agreement with the profile-observation based mixed-layer climatology of de Boyer Montégut et al. (2004), reducing the mean absolute difference to the climatology from 27 m to 19 m (comparison of mixer-layer depth in Fig. A6). In addition, the absolute difference of near-surface horizontal

velocities to the drifter-observation based climatology of Laurindo et al. (2017) is reduced by about 10% through DA (comparison of surface velocities in Fig. A7). The biological productivity near the equator is stable in ASML and FREE, indicating that FESOM-REcoM does not suffer from the erroneous upwelling known from previous DA studies (Park et al., 2018). The

…

440 In summary, the ASML temperature and salinity fields  from the surface to several hundred meters below, and mixed-layer depth are in good agreement with  observations, and the agreement of horizontal near-surface velocities with observations is improved. This can be interpreted as an indication that the velocity field in the upper part of the ocean  is also well represented. Although the spurious effects on deep ocean circulation should be further addressed in future work, we are confident that the DA provides an improved physical state in the upper ocean, which serves as an improved basis to estimate the air-sea $CO_2$ flux.

We show the comparison of mixed-layer depth and horizontal velocities in the Appendix.

**Mixed layer in Figure A6**

[Figure]

**Horizontal surface velocities in Figure A7**

[Figure]

50 **Reviewer's comment:**

L280: "4 Results" – Section 3, "Effect of DA on ocean physics" is also results. Perhaps Section 4 should be "Effect of DA on ocean biogeochemistry".

Answer:

We have adjusted the section titles based on your suggestion. For the structure of the revised manuscript with all sections and subsections, please see the table of contents in our answer to Reviewer's comment 6.

**51 Reviewer's comment:**

L282: "The ocean absorbs 2.78 Pg C dec$^{-1}$" – is this the correct unit? From Fig. 4a, it looks to be absorbing 2.78 Pg C yr$^{-1}$ on average over the decade.

Answer:

Thank you for having taken a closer look. Indeed, this was a typo and is now fixed in this and several other places, e.g.:

445    The ocean absorbs  2.78 Pg C yr$^{-1}$ in ASML and  2.83 Pg C yr$^{-1}$ in FREE during 2010-2020 (Fig. 4b), thus the assimilation decreases the global mean oceanic $CO_2$ uptake by  0.05 Pg C yr$^{-1}$.

**52 Reviewer's comment:**

L290: "air-sea CO2 flux (negative: into the ocean)" – if negative's into the ocean shouldn't it be "sea-air CO2 flux"?

Answer:

While the direction of air-sea $CO_2$ flux is not uniformly defined in the literature, the term 'air-sea' is commonly used for both for some reason, see e.g. Global Carbon Budget (Friedlingstein et al., 2023): 'air-sea flux' is positive into the ocean; and Roobaert et al. (2023): 'air-sea exchange' is negative into the ocean.

By defining outgassing as positive, the direction of $CO_2$ flux corresponds to the $pCO_2$ effect: Higher oceanic $pCO_2$ values result in a more positive flux.

**53 Reviewer's comment:**

L301: While STSS+ is broadly the northern bit and STSS- southern, it's a bit more nuanced than that and that should be reflected in the text.

Answer:

We have rephrased this (giving credits to the other reviewer's suggestions). Line 479:

$STSS_{SO}$+, roughly forms an outer northerly ring around the $STSS_{SO}$ biome (hatched area in Fig. 5a and b).

and Line 527:

SO characterized by a negative $CO_2$ flux difference between ASML and FREE, which we call the $STSS_{SO}-$, is a fragmented region and roughly consists of segments of an inner southerly ring (non-hatched area in Fig. 5a and b). In addition, reduced

**54 Reviewer's comment:**

Fig. 5: Add to the caption that the lines in a and b denote the regions, and the hashing (striping?) denotes STSS+.

Answer:

Added to the captions of figures 5 and 7:

**Figure 5.** Effect of data assimilation on Southern Ocean $CO_2$ flux and its seasonality averaged over the period 2010-2020. Negative numbers indicate a flux into the ocean. Additionally, lines in a and b denote the regions, and the green hatching denotes the $STSS_{SO}+$. (a) Map of

**55 Reviewer's comment:**

L462: "a pCO2-independent proxy for primary production" – I'm not sure "pCO2-independent" is needed here, I don't quite understand what's meant.

Answer:

We agree that it is not needed here. Line 726:

The representation of chlorophyll by the model is of interest as a  proxy for primary production.

Originally, we meant to point out that there is no direct relationship of chlorophyll and $pCO_2$ through the carbonate chemistry of seawater - unlike for all other variables (T, S, DIC and Alk) that are included in the observation comparisons.

**56 Reviewer's comment:**

L480: "as the modelled phytoplankton growth is temperature-dependent" – how sure are you the change is due to the direct temperature dependence rather than the indirect influence of stratification and mixing changes?

Answer:

We cannot separate these effects and have therefore rephrased the text:

> The major effects of physics DA on BGC variables seem to be related to changes of SST and are largely uniform over the full period of DA (Section 3.3). Surface chlorophyll changes follow SST changes  Figs. 1 and 11). The modeled phytoplankton growth is temperature-dependent (Gürses et al., 2023). Furthermore, indirect temperature effects
> 770 on plankton dynamics due to stratification and mixing changes contribute, albeit those can have heterogeneous effects and the correlation of chlorophyll and boundary-layer depth is less clear (not shown). The changes of surface DIC and alkalinity

As the link between sea surface temperature and mixing is not straight-forward, the temperature-dependence of growth is a more likely candidate to explain the similar spatial patterns of SST and chlorophyll changes (Figure R8).

[Figure]

**Figure R8:** Spatial patterns of the difference ASML-FREE for surface chlorophyll, SST and boundary layer depth.

**57 Reviewer's comment:**

L515: "There are two other data assimilating BGC model approaches" – there are many other data assimilating BGC model approaches! Perhaps a more accurate phrasing might be: "We compare here to two other data assimilating BGC model approaches …"

Thank you for the rephrasing suggestion, we used it (Line 833):

>  We compare here to two other data assimilating BGC model approaches, namely ECCO-Darwin (global; Carroll et al., 2020)  and B-SOSE, which is restricted to the Southern Ocean (Verdy and Mazloff, 2017). Both approaches use

**58 Reviewer's comment:**

L524: "suggesting that a flawed representation of ocean physics as an argument for the models underestimating the CO2 flux trend is unlikely" – I broadly agree, though it may depend on how well the wider circulation is represented.

L559: "suggests that the physical processes are already well represented in FREE" – again I broadly agree, but there may still be pertinent limitations, especially depending on the time and space scale.

Answer:

We agree with the reviewer that there are limitations. The revised Discussion no longer contains these statements (at least not verbatim). Furthermore, because "the free running model already represents temperature and salinity *rather well*" is a subjective assessment, we have also reworded the abstract:

> 5 over the period 2010-2020 to study the effect on the air-sea $CO_2$ flux and other biogeochemical variables.  The assimilation nearly halves the model-observation differences in sea surface temperature and salinity, with modest effects on the modeled ecosystem and $CO_2$ fluxes. The  main effects on the air-sea $CO_2$ flux  occur on small scales in highly dynamic regions, which pose challenges to ocean models. The largest imprint of assimilation is in the Southern

**59 Reviewer's comment:**

L565: "the adjustment of the ocean's carbon cycle to changes in the circulation" – true, though it's also possible that this might itself introduce biases in the carbon chemistry. See e.g. Lebehot et al. (2019, https://doi.org/10.1029/2019GB006186).

Answer:

We acknowledge that changes in the circulation may lead to imbalances of the ocean's carbon cycle, in particular during the adjustment phase, which may however take hundreds of years. The corresponding paragraph now reads:

> than changing the global mean SST, which differs by only 0.02°C between FREE and ASML. DA-induced differences in vertical transport of DIC are comparably large south of 50°S, but approximately 95% of them are balanced globally by opposing changes in vertical transport further north (vertical transport of DIC in Fig. A13a). In particular, the effect of DA on subduction of DIC through vertical advection into the ocean's deeper layers (not shown), which is the rate-limiting step on
> 820 oceanic uptake of anthropogenic $CO_2$ emissions (DeVries, 2022), appears small, which may be due to an insufficient amount of deep observations. Besides, experiments on longer time scales might be necessary to generate a visible effect of deep circulation changes on the ocean's carbon cycle (Cao et al., 2009), which could however lead to imbalances in the $CO_2$ flux (Lebehot et al., 2019; Kriest et al., 2020; Primeau and Deleersnijder, 2009). Another possible reason why the DA effect on the

---

## Author Response (AR2)

**Answers to the reviewers' comments**

We would like to thank the reviewers for their positive feedback and detailed comments. We have made minor changes to the manuscript based on their comments.

We have also corrected the data in Figure 8f because during the 1$^{st}$ revision, a confusion occurred as to which regional profile belongs into which figure panel. This does not change the results described in the text, and the data in this panel now matches with the initially submitted version again.

**Report #1**

**General comments:**

The authors have made considerable efforts to improve the manuscript and to address my previous comments. The resulting manuscript is a much better read and includes a thorough model analysis and evaluation. At this point, I have only a few minor comments and one suggested addition to the introduction or discussion.

Thank you for your kind feedback.

A point that has not been mentioned in the manuscript and perhaps a nice counterpoint to the idea that physical DA just disrupts a carefully calibrated coupled model is that the physical DA often reveals errors in the BGC parametrization. I am referring to statements such as "The question therefore arises to what extent an ecosystem model coupled to a data-assimilated physical model also represents a more realistic biogeochemistry..." (l 88). Studies, such as Löptien and Dietze (2019; DOI: 10.5194/bg-16-1865-2019), demonstrate that compensating for physical model errors through biogeochemical parameter estimation can lead to issues in forecasting climate-relevant metrics, despite reducing errors during the estimation period. So, if the physical DA has a large negative impact on biogeochemical estimates, this may point to problems in the biogeochemical model that may influence model forecasts even without DA. For example, if a physical ocean model underestimates coastal upwelling, maybe due to its coarse resolution, the biogeochemical may have an elevated maximum growth rate for phytoplankton to compensate for the reduced nutrient supply. If a climate forecast increases the nutrient supply to the euphotic zone, perhaps via increased wind stress, the response of the biogeochemical model may be unrealistically high. Physical data assimilation could help to reveal issues like this one, though solving the issue would subsequently require a parameter estimation experiment. Here, I am not suggesting any revision, but the authors could include this point in the manuscript.

We have included this point in the Introduction and Discussion (what was added in red):

Line 85: 'Furthermore, any coupled ecosystem model is adapted to its associated physical model with its strengths and weaknesses through carefully selected parameter values and a spin-up to near-equilibrium. Accordingly, the modeled carbon cycle may be  sensitive to deviations from the physical state that is typical for this model (Kriest et al., 2020; Spring et al., 2021). Potentially, this leads to biases in the carbon cycle through physics DA. Such effects highlight where physical model errors are compensated for by BGC parameters, and thereby DA may reveal critical areas for potentially unrealistic BGC model behavior in projections in a changing climate (Löptien and Dietze, 2019).'

Line 625: 'For example, surface chlorophyll (Fig. 11f) and $pCO_2$ (Fig. 9f) in the central Greenland Sea deteriorate in response to improvements of SST (Fig. 1c), SSS (Fig. 1f) and sea-ice concentration (not shown). This could indicate that the BGC parametrization compensates for flaws in the free running physical model in this region. The parameter mismatch might cause difficulties in modeling the change of BGC variables under the ongoing loss of Arctic sea ice (Chen et al., 2016).'

Specific Comments:

L 7: "The main effects": I would suggest adding "of the assimilation", the next sentence could then start with "Its", dropping the "of assimilation" there.
L 46: Why the "However" here? Isn't this another example where undersampling leads to issues?
L 55: "few years": I would suggest using "short time periods".
L 58: "Data assimilation also provides...": I think it is a bit more accurate to state that "Data assimilation can also be utilized to provide...".
L 69: "the uptake of atmospheric CO2 depends ultimately on the modeled physical carbon transport": What does "depends ultimately" mean here? Sinking of particles and perhaps even vertical migration of zooplankton can lead to carbon export. I would suggest using "in large parts" or similar. Also, why use "modeled" in this sentence?
L 196: I think it is more intuitive to use % for both fractions here. Why write "at each step" here, these are surely average fractions?
L 208: Subscript e appears to be missing for "rand".
L 221: "The ensemble standard deviation of the local instantaneous air-sea CO2 fluxes that results from the perturbation": I would suggest starting a new paragraph here.
L 240: I would suggest making it a bit more explicit that subscripts will denote the hemispheres.
Eq 5: What about the units here? There is still a °C in the exp.
L 245: For a consistent naming scheme, I would suggest adding the NA subscript to the "Newfoundland Basin+" region. I at first didn't think it was referring to a region and wasn't sure what to make of the +.
L 338: I had to read it twice to notice that surface values are stabilized while the DA continues to correct subsurface values. I would suggest using "subsurface" instead of "3D" in this sentence to make that more explicit.

Fig 6 and 8: If this is not too difficult, I would encourage the authors to add region labels to panel a in both figures. I found myself jumping between the text, Fig 8 (to look up the result) and Fig 7 (to look up the region name again).

L 623: "The major effects of physics DA on BGC variables seem to be related to changes of SST and are largely uniform over the full period of DA": While I know what is meant here, this statement could be misunderstood by readers. Changes in pCO2 are mostly affected by changes in DIC and alkalinity (which is clearly stated a few lines later). I would suggest rephrasing, so that readers don't get the impression surface pCO2 is mostly directly modified through changes in SST.

Thanks for the suggestions. We have implemented each of the above points, either omitting the words that were rather causing confusion or making minor additions where they were asked for (see Track Changes document).

L 219: "The strongest inflation ($\rho$ = 0.95) is applied during the first two weeks of the DA process.": Is this when the DA increments are strongest because state estimates are furthest from the observations?

That's right. We now added this as an explanation.

L 115: "surface-layer width": Is this the same as height?

That's also right, and we have changed the term to "layer thickness", as it's also called elsewhere in text.

L 173: Are the in-situ observations removed using a similar threshold as for the SST data?

No, we added "without excluding observations".

L 78: "data assimilation of ocean physics into a global ocean biogeochemistry model": Though data assimilation experts will know what is meant, the wording is a bit unclear. I would suggest rephrasing "data assimilation of physical observations into a coupled physical-biogeochemistry global ocean model".

Thanks, we changed this to: 'data assimilation of physical observations into a global ocean general circulation model coupled to a biogeochemistry model'.

L 103: What distinguishes diffusion and mixing here?

Repetitive and we have now deleted "mixing".

L 241: The descriptions that define the regions STSS_SO+ just says "the area where the assimilation leads to a more negative air-sea flux". I assume this is based on the difference between ASML and FREE. If so, please mention this and introduce ∆FCO2 at that point already

and not 3 lines below it. If not, please improve the description. Overall, I would suggest moving the description and definition of ΔFCO2 towards the top, as it is important for defining the regions used in the study.

That's right, and the paragraph now reads:

To study the effect of DA on the $CO_2$ flux, we define regions where the time-mean air-sea $CO_2$ flux difference ASML−FREE ($\Delta FCO_2$) is pronounced, based on the biome definition of Fay and McKinley (2014). Originally, these are, going polewards from the subtropics in each hemisphere, the Subtropical Seasonally Stratified Biome (STSS), the Subpolar Seasonally Stratified Biome (SPSS) and the Sea-Ice Biome (ICE). In the Southern Ocean (denoted by subscript $_{SO}$) within the STSS$_{SO}$, we differentiate between the area where $\Delta FCO_2$ is positive (the assimilation leads to a flux change directed out of the ocean) referred to as region 'STSS$_{SO}$+' and the area where $\Delta FCO_2$ is negative referred to as region 'STSS$_{SO}$−'. All Southern Ocean regions are outlined in Fig. 5a.

L 297: "On a global average, the SST in FREE is 0.14°C colder than the observations (...) In total, the global mean absolute difference of SST to the observations is reduced from 0.59°C to 0.32°C." I know there are many lines between this and re-reading it I see that one is based on absolute values, but I would suggest further adding a comparable number here. As a reader I was waiting to see by how much the 0.14°C bias was improved.

We have changed the order of these sentences, and added: "Thereby, the global mean model-observation difference is reduced from -0.14°C to -0.12°C, and from 0.59°C to 0.32°C in absolute terms."

**Report #2**

General comments:

The authors have done an excellent job of responding to the reviewer comments, and the paper is improved as a result.

We thank you for the positive feedback.

Specific comments:

My only extremely minor comment on reading through is:
Line 349-350: "deep water formation events characterised by a mixed-layer depth of more than 1000 m or 500 m occur less frequently in ASML (not shown)." Is there a significance of "more than 1000 m or 500 m", or could this just read "more than 500 m"?

We changed this to 1000m as it's used e.g. here:
https://os.copernicus.org/articles/13/609/2017/os-13-609-2017.html

---

## Author Response (AR3)

Thank you for the good review process and your positive response.

When preparing the files for article production, I checked the Figure Content Guidelines again. I made small corrections, such as the hyphen and the en dash, the degree sign for coordinates on the tick labels, in Figure 11 the numbering of the panels and the missing continent fill colour.